# Observation of 2D-magnesium-intercalated gallium nitride superlattices

Jia Wang[1,2 ✉], Wentao Cai[2], Weifang Lu[3,6], Shun Lu[4], Emi Kano[2], Verdad C. Agulto[5], Biplab Sarkar[2,7], Hirotaka Watanabe[2], Nobuyuki Ikarashi[2], Toshiyuki Iwamoto[5], Makoto Nakajima[5], Yoshio Honda[1,2] & Hiroshi Amano[1,2 ✉]

Since the demonstration of p-type gallium nitride (GaN) through doping with substitutional magnesium (Mg) atoms[1,2], rapid and comprehensive developments, such as blue light-emitting diodes, have considerably shaped our modern lives and contributed to a more carbon-neutral society[3–5]. However, the details of the interplay between GaN and Mg have remained largely unknown[6–11]. Here we observe that Mg-intercalated GaN superlattices can form spontaneously by annealing a metallic Mg film on GaN at atmospheric pressure. To our knowledge, this marks the first instance of a two-dimensional metal intercalated into a bulk semiconductor, with each Mg monolayer being intricately inserted between several monolayers of hexagonal GaN. Characterized as an interstitial intercalation, this process induces substantial uniaxial compressive strain perpendicular to the interstitial layers. Consequently, the GaN layers in the Mg-intercalated GaN superlattices exhibit an exceptional elastic strain exceeding −10% (equivalent to a stress of more than 20 GPa), among the highest recorded for thin-film materials[12]. The strain alters the electronic band structure and greatly enhances hole transport along the compression direction. Furthermore, the Mg sheets induce a unique periodic transition in GaN polarity, generating polarization-field-induced net charges. These characteristics offer fresh insights into semiconductor doping and conductivity enhancement, as well as into elastic strain engineering of nanomaterials and metal–semiconductor superlattices[13].

The interplay between gallium nitride (GaN) and magnesium (Mg) dates back to the first demonstration of p-type GaN through the substitutional incorporation and activation of Mg atoms (zero-dimensional (0D) Mg doping)[1], a breakthrough that heralded the era of white light-emitting diodes[4]. However, the low hole mobility, whether arising from 0D Mg doping[1] or polarization-induced hole gas[14], remains a fundamental limit in group III/nitride semiconductors. Analogous to strained silicon[15,16], applying strain to modify the band structure of GaN has been identified as a strategy to enhance carrier mobility[17–19]. However, achieving and maintaining high elastic strain in GaN to demonstrate this enhancement has been difficult.

In another domain, intercalation is an important nanotechnology for the fabrication of artificial layered structures and has found a wide range of applications, such as exfoliation of van der Waals materials[20], energy storage[21–23], superconductors[24,25] and tuning of thermal conductivity[26]. In general, van der Waals materials are chosen as the host for the intercalation process because they enable the easy insertion of foreign atom, ion and molecule sheets without causing excessive strain. Similarly, it is considered extremely difficult to intercalate atomic sheets into single crystals with strong ionic and covalent bonds, such as wide-bandgap semiconductor materials.

## 2D-Mg$_i$-intercalated GaN superlattices

Here we report an unusual phenomenon: the spontaneous intercalation of monoatomic Mg sheets into hexagonal GaN, forming a superlattice structure, a process known as 2D-Mg doping.

Our progressively magnified high-angle annular dark-field scanning transmission electron microscopy (HAADF-STEM) images (Fig. 1a–d) reveal the intricate details of the Mg-intercalated GaN superlattices (MiGs) structure. Typically, a single continuous sheet of Mg intercalant has a diameter of several tens of nanometres, with 5–10 layers of GaN observed between each pair of Mg sheets (Fig. 1c). Atomically resolved integrated differential phase contrast (iDPC)-STEM imaging (Fig. 1d) enables the visualization of Ga, N and Mg atoms in a single-imaging condition[27], confirming that the intercalant sheets comprise a single atomic layer. Energy-dispersive X-ray spectroscopy (EDS) and elemental mappings confirm that this monolayer is composed entirely of Mg (Fig. 1f,g). A schematic of this region is shown in Fig. 1e. The interstitial occupancy of the Mg layer results in an ABCAB registry with the adjacent hexagonal GaN layers following an ABAB stacking sequence. Each Mg atom, positioned at the C site, is surrounded by six N atoms, occupying an octahedral interstitial site (Fig. 1e). This confirms that

[1]Institute for Advanced Research, Nagoya University, Nagoya, Japan. [2]Institute of Materials and Systems for Sustainability, Nagoya University, Nagoya, Japan. [3]Department of Materials Science and Engineering, Meijo University, Nagoya, Japan. [4]Department of Electronics Engineering, Nagoya University, Nagoya, Japan. [5]Institute of Laser Engineering, Osaka University, Osaka, Japan. [6]Present address: Department of Physics, Future Display Institute in Xiamen, Xiamen University, Xiamen, China. [7]Present address: Department of Electronics and Communication Engineering, Indian Institute of Technology Roorkee, Uttarakhand, India. ✉e-mail: wang@nagoya-u.jp; amano@nuee.nagoya-u.ac.jp

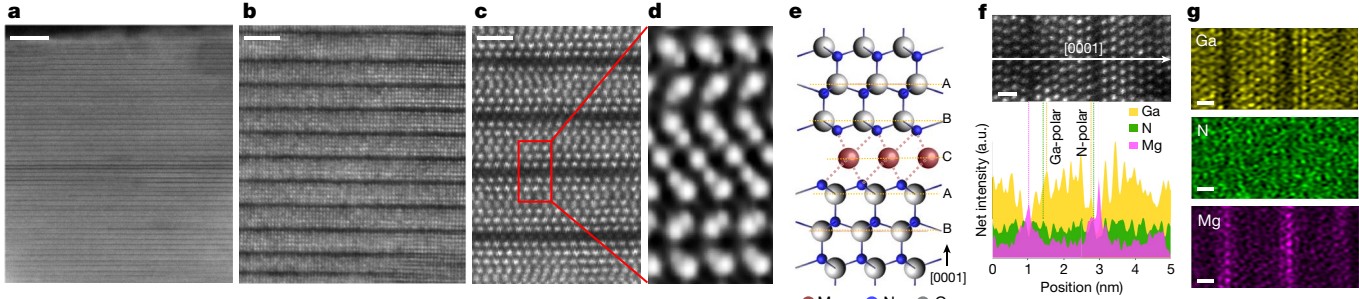

**Fig. 1 | Mg-intercalated GaN superlattices. a–c**, Cross-sectional HAADF-STEM images with progressively magnified views reveal the typical structure of 2D-Mg-intercalated GaN superlattices. The dark lines indicate monoatomic 2D-Mg$_i$ sheets and are perpendicular to the *c* axis (the [0001] direction). **d**, An iDPC-STEM image (magnified view of the region in the red box in **c**) showing the repeating unit structure of the superlattice in which the constituent Ga, N and Mg atoms are clearly visible. **e**, A schematic of the superlattice structure from **d**, detailing the positions of the constituent atoms. **f**, Atomically resolved EDS spectra across a localized portion of superlattices. Intensity peaks indicate the relative positions of the atomic planes for each element. This region is shown in a HAADF-STEM image (top) in which the arrow marks the line-scanning direction. **g**, Atomically resolved EDS elemental maps for Ga, N and Mg in the same region as **f**. Scale bars: 10 nm (**a**); 2 nm (**b**); 1 nm (**c**); 500 pm (**f**, top); 500 pm (**g**).

the segregation of interstitial Mg (Mg$_i$) into the atomic sheet does not disrupt the original lattice symmetry of the hexagonal GaN.

Furthermore, the Mg layer tends to repel Ga, leading to the nearest GaN monolayers exhibiting opposite polarity when positioned below and above the Mg layer (Fig. 1f). Furthermore, in the immediate neighbouring GaN monolayers adjacent to the Mg layer, the intensity of Ga decreases to approximately three-quarters of the original peak values, whereas the intensity of Mg greatly increases relative to the otherwise noise-like signals (Fig. 1f). This observation is consistent with the theoretical model, which predicts the energetically stable configuration of a monolayer of interstitial Mg segregation (2D-Mg$_i$) onto (0001) GaN[28], favouring a quarter substitutional occupation of Mg in the nearest Ga layers.

## Polarity transition induced by pairs of 2D-Mg$_i$ sheets

A single 2D-Mg$_i$ induces a polarity-inversion region in the shape of a pyramid, as evidenced by pyramidal inversion domains, which are well-observed defects in heavily Mg-doped GaN[29]. We have observed that when a second 2D-Mg$_i$ is brought into proximity with the first, the volume of the polarity-inversion region reduces and an inverted truncated pyramid shape is formed. This effect results from the partial recovery of polarity to reduce the energy. As a result, a pair of 2D-Mg$_i$ sheets experience an attractive force, similar to the attraction between two edge dislocations of opposite sign caused by the cancellation of the strain field. This stabilizes the structure, allowing them to align vertically and expand into a larger-diameter domain, underscoring the driving force to form an ordered superlattice structure.

Figure 2a–d shows the edge zone of vertically aligned 2D-Mg$_i$ sheets featuring gradual transitions between Ga-polar and N-polar GaN, and Fig. 2e resolves a single atomic plane of GaN exhibiting a gradual transition of polarity across several atoms. Notably, we observed that N atoms remain in their original positions, whereas Ga atoms transition from lower sites (N-polar) to upper sites (Ga-polar) relative to N atoms. Figure 2f schematically presents the extended edge zone of a pair of 2D-Mg$_i$ sheets, emphasizing the polarity transitions from N-polar to Ga-polar GaN that occur both perpendicular and parallel to the Mg sheets. The schematic on the left details the sign of net charges and the direction and intensity of the polarization field that results from the polarity transition, including changes in the spontaneous polarization and piezoelectric polarization along the [0001] direction (*c* axis), where the net polarization field intensity diminishes to zero before incrementally increasing in the opposite direction. This results in unbalanced polarization-induced charges owing to the gradient of the polarization field[14]. Because hole doping is possible from the

polarization field-induced unbalanced charges[30], the potential for hole generation arises from the periodic polarity transition in GaN induced by the 2D-Mg sheets. This makes the concept of 2D-Mg doping particularly important and offers a unique insight into new dimensions in the study of polarization-induced hole doping.

## Interstitial intercalation and compressive strain

The rare occurrence of a metal–semiconductor superlattice structure represents a form of interstitial intercalation, in contrast to the commonly observed substitutional intercalation that can be further divided according to stoichiometry. This is a key factor determining whether such intercalation results in a compound material or a structure (Fig. 3a). For instance, the layered hexagonal MAX phase adheres to a strict stoichiometric formula, $M_{n+1}AX_n$, where $n = 1$ to 4, M is an early transition metal, A is a group 13 or 14 element and X is carbon and/or nitrogen[31,32]. The intercalation sheets extend across entire layers without terminating in the matrix, classifying it as a material. By contrast, the case represented by the Guinier–Preston zone features the substitutional intercalation of atomic sheets that are embedded, and invariably terminated, in the matrix, indicating it is a structure because of the absence of stoichiometry[33]. Similarly, in the MiGs configuration, the intercalant sheets of 2D-Mg$_i$ always terminate in the GaN matrix, classifying it as a structure. Moreover, unlike substitutional intercalation structures in which the continuity of the matrix atomic planes is disrupted by the substitutional sheets, the continuity of GaN atomic planes is preserved in interstitial intercalation, highlighting an important distinction.

Assuming non-lattice swelling, a reduction in the number of GaN atomic planes results in increased strain. When the number of GaN atomic planes between a pair of Mg sheets reduces to six, the uniaxial strain in hexagonal GaN can exceed −12% (Supplementary Fig. 1). This is confirmed by the strain maps based on the atom-column by atom-column distance measurements (Fig. 3b–d), in which the average lattice constant *c* in unintercalated and intercalated GaN is 5.18 Å and 4.55 Å, respectively. This results in a compressive strain, $\varepsilon_c$, parallel to the *c* axis, of −12.2%. Furthermore, the average lattice constant *a* changes from 3.18 Å in unintercalated GaN to 3.25 Å in intercalated GaN, indicating a tensile strain, $\varepsilon_a$, perpendicular to the *c* axis, of +2.2% (Fig. 3d). The resulting Poisson ratio, $v = -\varepsilon_a/\varepsilon_c$, calculated to be 0.18, aligns well with that measured for wurtzite GaN[34], validating the uniaxial strain and preservation of lattice symmetry in GaN under high-strain conditions. Further data displaying similar findings are provided in Extended Data Fig. 2. Given the high elasticity modulus of GaN (around 295 GPa[35]), the elastic strain induces an exceptionally high stress, exceeding 20 GPa.

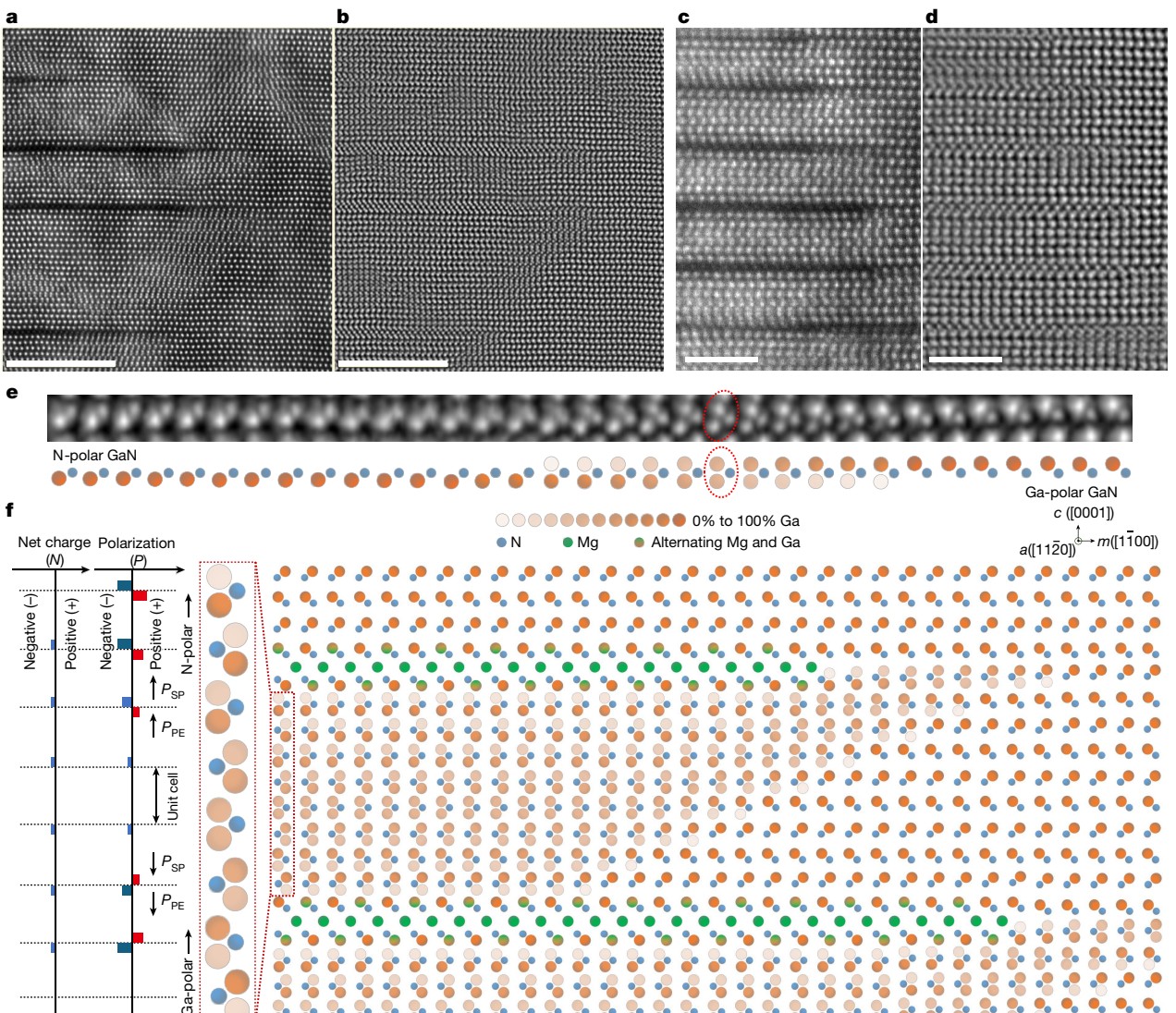

**Fig. 2 | Polarity transition induced by 2D-Mg_i intercalant sheets. a,b**, HAADF-STEM (**a**) and iDPC-STEM (**b**) images of the same region showing the edge of vertically aligned 2D-Mg_i sheets along the [11$\bar{2}$0] zone axis (*a* axis). **c,d**, Magnified HAADF-STEM (**c**) and iDPC-STEM (**d**) images of the same region along the [11$\bar{2}$0] zone axis (*a* axis). **e**, Top, an iDPC-STEM image that atomically resolves a monolayer of GaN, demonstrating a gradual transition in polarity over a few atoms. N atoms remain in their original positions but Ga atoms transition from lower sites (N-polar) to upper sites (Ga-polar) relative to N atoms. The red ellipse highlights a line of GaN (along the *a* axis and perpendicular to the page) in the region of the polarity transition. Bottom, a schematic illustrating the transition shown in the iDPC-STEM image. **f**, A schematic of the area at the edge of a pair of 2D-Mg_i sheets, emphasizing the polarity transitions from N-polar to Ga-polar GaN occurring both perpendicular and parallel to the Mg sheets. The gradient of Ga occupancy at specific sites is represented by varying shades of orange (from 0% to 100%), indicating the average occupancy of Ga. The multicoloured symbols represent alternating occupancy by Mg and Ga along the *a* axis. Left, the net charges resulting from the polarity transition are shown, including changes in the polarization field (spontaneous polarization (**P**_SP) and piezoelectric polarization (**P**_PE)) along the *c* axis. This leads to unbalanced polarization-induced charges, *N*, resulting from the gradient of the polarization field, **P**, where $N = 1/q|\nabla \cdot \mathbf{P}|$ and *q* is the elementary charge (ref. 14). Scale bars, 10 nm (**a**–**d**).

In general, the elastic strain of a solid exceeding a few tenths of a percent is easily subject to relaxation by plastic deformation[36]. Similar to the record-high elastic strains achieved in dislocation-free nanowires[12], the unprecedented high elastic strain of greater than 10% in thin-film materials may provide fresh clues to the elastic strain engineering of nanomaterials[37].

## Perfect lattice match between GaN and Mg

The main prerequisite for forming a superlattice between two types of material is a minimal lattice mismatch[38]. We compare the lattice parameters and other relevant fundamental properties[39,40] of GaN and Mg in Table 1. Despite GaN being a wide-bandgap semiconductor with mixed ionic and covalent bonding, and Mg being a metal featuring metallic bonding, these two dissimilar materials share the same crystal structure (a hexagonal *P*6₃*mc* space group). It is intriguing that the lattice mismatch between hexagonal GaN and hexagonal Mg (in-plane, $(a_{GaN} - a_{Mg})/a_{GaN} = -0.4\%$; out-of-plane, $(c_{GaN} - c_{Mg})/c_{GaN} = -0.3\%$) is negligibly small, within the error range of experimentally measured values. This mismatch is one to two orders of magnitude lower than that of common substrates for GaN epitaxy, such as Si (in-plane, +17%), sapphire (in-plane, −16%) and SiC (in-plane, +3.5%)[41]. We think that the perfect lattice match between GaN and Mg greatly reduces the mismatch strain, having a critical role in the spontaneous formation of 2D-Mg_i on (0001) GaN. This may also explain why Mg segregates easily in GaN.

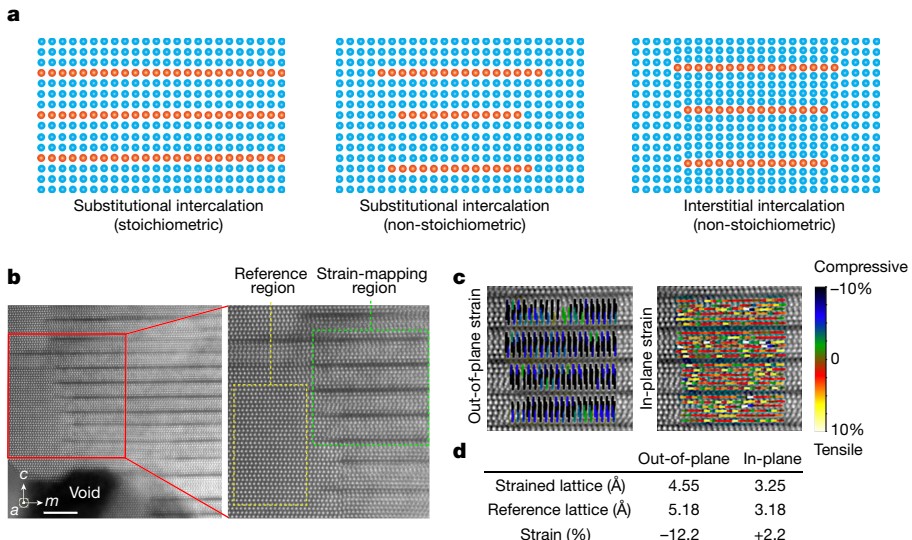

**Fig. 3 | High uniaxial compressive strain in the MiGs nanostructures from interstitial intercalation. a**, A schematic of the three cases of intercalation of atomic sheets into non-van der Waals solids, represented by the MAX phase (left), the Guinier–Preston zone (middle) and the MiGs structure (right). **b**, Drift-corrected HAADF-STEM images with atomic resolution showing a typical region of the MiGs structure alongside a void (left). In the magnified view (right), the strain-mapping region and reference region are delineated by green and yellow dashed boxes, respectively. **c**, The out-of-plane ($\varepsilon_c$, left) and in-plane ($\varepsilon_a$, right) strain maps show atom-by-atom distance measurements in the strain-mapping region. **d**, Summary of the mean lattice constants and strain values in **c**. Scale bar, 5 nm (**b**).

## Modified physical properties by incorporating MiGs

The highly strained hexagonal GaN in the MiGs structure exhibits intriguing physical properties that result from the modification of band structures, which also influence phonon–electron interactions and carrier-transport properties[42,43]. The uniaxial compressive strain decreases the $c/a$ ratio, manifesting in an exceptionally low $c/a$ of 1.415 for hexagonal GaN. This also increases an internal parameter (the relative displacement between Ga and N sublattices along the $c$ axis) and the bandgap (Extended Data Fig. 3a,c,d). The high strain also shifts the energy ordering of the three valence bands, indicating a reversal of the crystal-field splitting energy sign compared with unstrained wurtzite GaN[18,19] (Extended Data Figs. 3b and 4 and Supplementary Note 1). This valence-band alteration greatly reduces the effective mass of holes along the $c$ axis. As a result, uniaxially compressed GaN should exhibit greatly enhanced hole mobility along the $c$ axis owing to the dominance of split-off hole bands and the small effective mass of split-off holes along the Γ-to-A path. (Γ and A indicate (0, 0, 0) and (0, 0, ½) in reciprocal lattice units. In the wurtzite structure such as GaN, the Γ-to-A path lies along the $c$ axis).

To evaluate the enhanced hole transport along the $c$ axis (the out-of-plane direction), we used terahertz time-domain ellipsometry (THz-TDE). Compared with the van der Pauw method, which measures only in-plane conductivity, $\sigma_\perp$, THz-TDE measures both $\sigma_\perp$ and out-of-plane conductivity, $\sigma_\parallel$. Our TDE results indicate a six-fold increase in the optical conductivity of GaN after incorporating MiGs (Extended Data Fig. 6 and Methods), implying greatly enhanced out-of-plane mobility, $\mu_\parallel$, potentially the result of a reduced effective mass (averaging reduced to 30% of the original value) from the predominance of split-off holes. Typically, carrier mobility and concentration are inversely related, but Mg intercalation into GaN increases both, which may provide insight into conductivity enhancement in semiconductors.

In our electrical characterizations, we demonstrate that incorporating the MiGs structure modifies the surface potential of GaN. This provides considerable technological advantages, enhancing the barrier height of n-type Schottky barrier diodes on undoped GaN and enabling the otherwise difficult realization of ohmic contact on p-type GaN.

After the MiGs structure is formed by varying the annealing temperature on an unintentionally doped GaN (intrinsically behaving as lightly n-doped), we observed a linear correlation between the annealing temperature and the voltage intercepts of the reverse bias line (indicative of built-in potential) on the $1/C^2$–$V$ curve (Fig. 4a), indicating an increase in the built-in potential. This trend provides compelling evidence of more ionized acceptors at higher annealing temperatures, corresponding to higher coverage of the MiGs structure. We also show that the acceptors introduced by the MiGs structure contributed to additional depletion-region depth (Fig. 4b). The origin of ionized acceptors in the MiGs structure is probably induced by the polarization field owing to the periodic polarity transition in GaN (Fig. 2d), rather than interstitial Mg, which is electrically inactive. Further state-of-the-art characterization tools are needed to verify this assumption. This phenomenon can also be used to increase the barrier height of Schottky barrier diodes on n-type GaN[44].

In addition to n-type GaN, the incorporation of MiGs on an existing p-type GaN can greatly reduce the specific contact resistivity. This is beneficial because of the low processing cost and the compatibility of MiGs fabrication with the 'front-end-of-line' process in the semiconductor industry. We have previously reported enhanced ohmic contact by annealing metallic Mg thin films onto GaN[45,46]. However, the exact mechanism, particularly the nature of the so-called 'MgGaN' layer[46], was not elucidated.

**Table 1 | Comparison of the physical properties of GaN and Mg**

| Material | Lattice | Bonding type | Lattice constants | | Modulus of elasticity (GPa) | Electrical conductivity |
|---|---|---|---|---|---|---|
| | | | $c$ (Å) | $a$ (Å) | | |
| GaN | Hexagonal close packed | Mixed ionic and covalent | 5.185 | 3.189 | 295 | Semiconductor |
| Mg | Hexagonal close packed | Metallic | 5.200 | 3.202 | 44 | Metal |

Data from refs. 39,40.

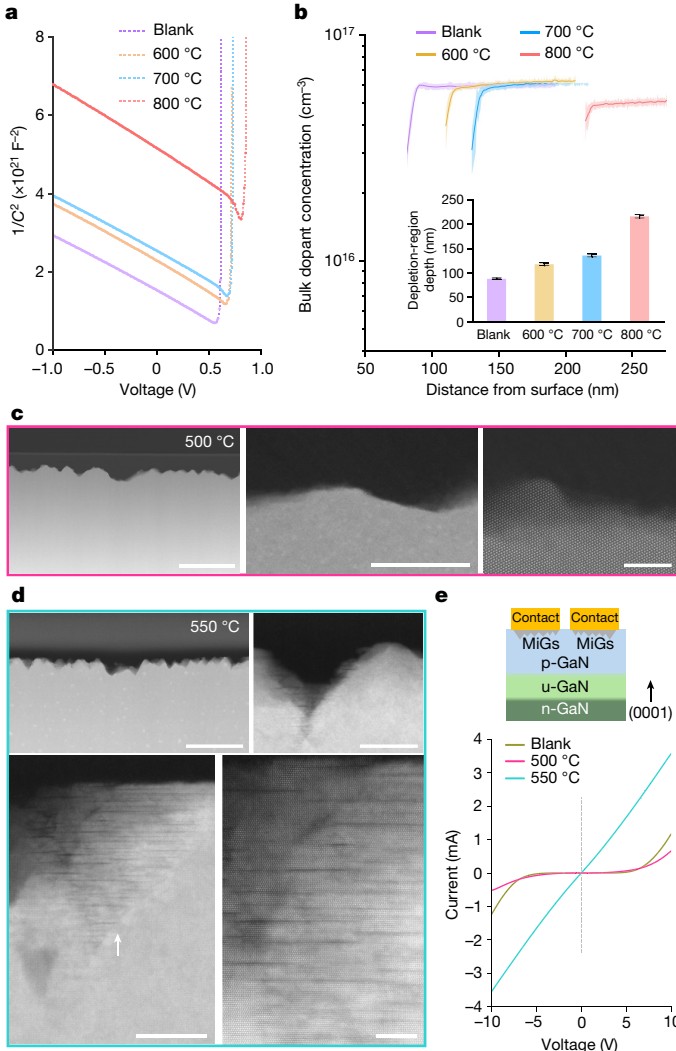

**Fig. 4 | Electrical properties of MiGs on n-type and p-type GaN. a**, The capacitance–voltage characteristics ($1/C^2$–$V$ plot) of a lightly n-doped (0001) GaN epilayer on an n-GaN substrate, annealed with Mg at varying temperatures, are compared with those of a blank sample. Frequency, 100 kHz; a.c. level, 20 mV. **b**, Bulk dopant concentration as a function of depth from the surface, extracted from the $1/C^2$–$V$ plot (error bands, s.d.; $n = 5$). Inset, the varying depletion-region depths in lightly n-doped (0001) GaN, annealed with Mg at varying temperatures, compared with the blank sample (error bars, s.d., $n = 5$). **c**, STEM images with progressively magnified views demonstrate that an initial p-type GaN epilayer, annealed with Mg at 500 °C for 10 min, exhibits a rough surface but without signs of Mg intercalation. Scale bars: 200 nm (left), 20 nm (middle), 5 nm (right). **d**, STEM images with progressively magnified views indicate that the p-type GaN epilayer on u-GaN (unintentionally doped, semi-insulating GaN)/n-GaN substrate, annealed with Mg at 550 °C for 10 min, exhibits a rough surface with Mg intercalation characteristic of the early-stage formation of a MiGs structure (arrow). Scale bars: 200 nm (upper left), 20 nm (upper right), 20 nm (lower left), 5 nm (lower right). **e**, Current–voltage ($I$–$V$) characteristics of the transfer-length-method test structure fabricated on the p-type GaN samples, annealed with a selective-area-patterned metallic Mg film at 500 °C (10 min) and 550 °C (10 min) compared with a blank sample.

By controlled experiments combining STEM observation and $I$–$V$ characteristic analyses, we have conclusively identified that the MiGs structure is responsible for the ohmic contact. We show that the p-type GaN annealed with metallic Mg at 500 °C for 10 min, followed by an acid clean, results in a roughened surface but without signs of Mg intercalation (Fig. 4c). This sample did not exhibit ohmic behaviour in the $I$–$V$ characteristics, and neither did the blank sample (without

Mg deposition), indicating that surface roughening alone does not lead to ohmic contact. Conversely, the sample annealed at a slightly higher temperature shows intercalation of short Mg sheets, indicative of the early-stage formation of MiGs structure (Fig. 4d). By contrast, the $I$–$V$ characteristics showed immediate, effective ohmic contact with good linearity, demonstrating a clear association with the formation with MiGs (Fig. 4e). We also note that the threshold temperature and time for the onset of MiGs formation may vary slightly depending on the initial Mg concentration in GaN. Typically, it is higher (for example, 600 °C for 10 min) for unintentionally doped GaN and lower (550 °C for 10 min) for GaN that is initially heavily doped with Mg.

The MiGs structure leads to greatly reduced contact resistivity ($\rho_c$) owing to the increased tunnelling probability. This effect is probably a result of the combination of factors already identified, including an increased acceptor concentration and a decreased effective mass, as described by the formula[47,48]:

$$\rho_c \propto \exp\left(\frac{2\varphi_B}{q\hbar}\sqrt{\frac{\varepsilon_s m^*}{N_a}}\right) \quad (1)$$

where $\varphi_B$ is the barrier height, $q$ is the elementary charge, $\hbar$ is the reduced Planck constant, $\varepsilon_s$ is the permittivity, $m^*$ is the effective mass and $N_a$ is the acceptor concentration.

The enhanced hole transport in MiGs structure also contributes greatly to the exceptional performance of GaN p–n junction diodes[45] and p-type GaN Schottky barrier diodes[49], highlighting the considerable technological potential of these nanostructures for widespread electronic-device applications.

As well as the modified electronic band structure and the altered optical and electrical properties, the enhancement of thermal transport arising from strained GaN is on the horizon, along with other emerging physical properties. It is also meaningful to draw wisdom from such spontaneously formed structures to fabricate artificial metamaterials with precise composition and uniform thickness control[13,50]. Consequently, the layered structure of MiGs may offer a new tool to study the band structure and transport properties of metal–semiconductor superlattices, opening avenues for the development of advanced materials and devices.

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

# Methods

## Thermal annealing of metallic Mg on GaN

This intercalation occurs at low cost and with accessible conditions, specifically the annealing of a bare metallic Mg film on GaN in either a nitrogen or argon atmosphere at atmospheric pressure. This process is characterized by a rapid diffusion that requires only a few minutes at temperatures in the range 550–800 °C, which is fully compatible with existing semiconductor fabrication processes.

We first deposited metallic Mg thin film, without a capping layer, onto single-crystalline wurtzite GaN (hexagonal $P6_3mc$ space group) using conventional physical vapour-deposition methods, such as electron-beam evaporation or sputtering. Subsequently, these samples were then annealed at elevated temperatures in atmospheric pressure, typically exceeding 550 °C for 10 min. Notably, higher annealing temperatures corresponded to reduced times required for the intercalation process. When heated above the melting point of Mg (650 °C), molten Mg also proved effective in producing this effect. The rapid interstitial diffusion of Mg within GaN resulted in its segregation into single monolayer atomic sheets. These sheets then expanded in size, resembling climbing motion of edge dislocations, and aligned vertically (along the $c$ axis) with each other in an even and orderly manner. However, given the spontaneous and diffusion-driven nature of this reaction, the nucleation of Mg intercalants within the GaN matrix tended to be inhomogeneous, leading to a non-uniform distribution of the resulting MiGs domains (Extended Data Fig. 1).

Atmospheric pressure is strongly recommended to prevent the early sublimation of Mg before it reacts with GaN. Because Mg evaporates easily at low pressure, for example in the high-vacuum condition in a typical molecular-beam epitaxy (MBE) growth environment ($10^{-8}$ to $10^{-12}$ torr, which is roughly $10^{-6}$ to $10^{-10}$ Pa), the boiling point of Mg is below 200 °C[51], whereas under 1 atmosphere, the boiling point of Mg increases to more than 1,000 °C, which is much higher than the melting point of 650 °C[52]. Therefore, if a 50-nm-thick Mg film is annealed in a rapid thermal annealing furnace at 700 °C under a vacuum of 10 Pa, given that the boiling point of Mg is approximately 500 °C, the Mg should have sublimated during the temperature ramp-up phase.

Before Mg deposition, if the GaN samples contain a top p-type GaN epitaxial layer that was initially doped with Mg during epitaxy, the samples were first subjected to a conventional annealing process (750 °C for 10 min) for Mg activation and hydrogen depassivation[53].

We then deposited an amorphous, pure, bare metallic Mg thin film, approximately 50 nm thick, onto the GaN samples using an ACS-4000 (ULVAC) sputtering system with a 3 N (99.9% purity) Mg target.

After deposition, the handling of metallic Mg film requires special attention to prevent oxidation and dissolution during the lift-off process in liquids. Generally, the oxidation resistance of Mg is considered good in dry air up to approximately 400 °C, and up to about 350 °C in moist air, with humidity having an important role in Mg corrosion as it forms $Mg(OH)_2$ (ref. 54). Based on our experience, at room temperature the oxidation resistance of Mg is robust; the as-deposited Mg thin film (50 nm) retains its metallic lustre when kept in air for days. We did not take special measures to prevent its oxidation, and the thermal annealing of the Mg film was typically conducted within one day after deposition. However, Mg is prone to oxidation in dry air above 500 °C, so pure nitrogen or argon is needed during annealing. Nonetheless, we acknowledge that the Mg film surface was probably oxidized and covered by a thin oxide layer as a result of its exposure to air. This might also explain the extra oxygen that was incorporated into the MiGs structure in the secondary ion mass spectrometry (SIMS) profiles (Supplementary Fig. 3).

For photolithography and patterning of Mg, the lift-off process of photoresist and Mg film was done in liquid solution. We found that Mg nanofilm tended to dissolve in deionized water, losing its metallic colour after just a few minutes, so we avoided using deionized water. Instead, we opted for pure isopropyl alcohol (IPA), finding that Mg remains stable in IPA, which is sufficient for the lift-off process. IPA is routinely used instead of deionized water in semiconductor processing practices. We also used N-methyl-2-pyrrolidone (or acetone) followed by methanol for the lift-off process before using IPA.

The GaN samples, after undergoing blank deposition or patterning of Mg film, were annealed at various temperatures: 500 °C, 550 °C, 600 °C, 700 °C and 800 °C, all for 10 min. Unless specified otherwise in comparative experiments for annealing temperature, we set the default annealing temperature at 800 °C. We found that the formation of superlattice structure was characterized by a rapid diffusion process that required only 10 min or less at temperatures ranging from 550 °C to 800 °C. Temperatures between 800 °C and 1,000 °C are feasible but made no substantial difference compared with temperatures of around 800 °C. However, annealing at temperatures higher than 1,000 °C requires a masking layer to prevent incongruent decomposition of GaN. Furthermore, annealing at temperatures of more than 1,000 °C favours substitutional diffusion of Mg[55], which could lead to thermal instability of the superlattice structure and decomposition of the interstitial Mg sheets. Temperatures lower than 550 °C might be possible but could require much longer annealing times[45].

After cooling, the samples were cleaned with either hydrochloric acid or aqua regia. This cleaning step was essential for removing any Mg residues and compounds that were soluble in acid, which might have formed during exposure to moisture, hot air or hot nitrogen.

## Preparation of GaN samples

We used a diverse range of GaN samples to demonstrate the accessibility and versatility of the Mg intercalation process. Our findings indicate that, regarding the formation and morphology of the superlattice structure, there is no great difference between n-type GaN and p-type GaN, except for the initial Mg content in these samples. Specifically, n-type GaN does not contain Mg initially, whereas p-type GaN is already enriched with Mg. These pre-existing Mg atoms could contribute to the formation of Mg sheets. Consequently, the presence of Mg in p-type GaN may slightly lower the diffusion temperature or the time required to form the MiGs structure. Thus, including studies on both n-type and p-type GaN samples in our research may highlight the general applicability of this method, regardless of their conductivity type and film quality. As a result, our GaN samples varied in initial doping concentration (heavily n-doped, which is a commercial n-type GaN substrate without re-epitaxy, and lightly n-doped, which is unintentionally doped GaN, lightly p-doped and heavily p-doped) and substrate type (commercial GaN templates, consisting of GaN buffer layers on sapphire substrates, and commercial GaN substrates).

1. For samples used for the HAADF-STEM and iDPC-STEM observations, except for the STEM observation with $I$–$V$ characteristics in Fig. 4, the unintentionally doped GaN 2 μm thick was grown by metal–organic vapour phase epitaxy on a commercial (0001) heavily doped n-type GaN substrate 400 μm thick and 2 inches (51 mm) in diameter with a free electron concentration of around $10^{18}$ cm$^{-3}$.

2. For samples used for the STEM-cathodoluminescence (CL) characterizations, these GaN substrates with epitaxy-ready smoothness were used.

3. For GaN samples used for plan-view CL and cross-sectional polarized CL measurements, we used epitaxy layers consisting of 200-nm-thick p-type GaN with [Mg] = $6 \times 10^{18}$ cm$^{-3}$ followed by 1-μm-thick unintentionally doped GaN on the above-mentioned GaN substrate.

4. For GaN samples used for terahertz time-domain ellipsometry, the epitaxy layers consisted of p-type GaN around 450 nm thick, one with [Mg] = $6 \times 10^{18}$ cm$^{-3}$ and another with [Mg] = $2 \times 10^{19}$ cm$^{-3}$, followed by 3.9-μm-thick unintentionally doped GaN ([C] = $2 \times 10^{16}$ cm$^{-3}$,

[O] = $1 \times 10^{16}$ cm$^{-3}$ and [Si] below the detection limit) on commercial sapphire substrates 2 inches in diameter. We also prepared another GaN sample with the same 3.9-µm-thick unintentionally doped GaN (without p-type GaN) on the above-mentioned sapphire substrate and a bare sapphire substrate.

5. For GaN samples used for $C–V$ profiling, the epitaxy layers consisted of unintentionally doped GaN roughly 2 µm thick with a net doping concentration of $6 \times 10^{16}$ cm$^{-3}$ on the above-mentioned GaN substrate.

6. For GaN samples used for backside secondary-ion mass spectrometry (SIMS), the epi-layers consisted of 400-nm-thick p-type GaN ([Mg] = $7 \times 10^{18}$ cm$^{-3}$) and 2.5-µm-thick unintentionally doped GaN with [Si] = $7 \times 10^{16}$ cm$^{-3}$ on the above-mentioned GaN substrate.

7. For GaN samples used for $I–V$ characteristics and the combined STEM observations, the samples were the same with those in item (6). The conventional top degenerately doped p-type contact layer (the p$^+$ contact layer) intended for ohmic contact purposes was not grown intentionally.

## STEM-based characterizations
STEM samples were prepared using a Thermo Fisher Helios 660 focused ion beam (FIB) STEM. Before FIB sample preparation, a protective carbon layer was deposited on the sample surface to reduce Ga FIB-induced damage. Samples were thinned with starting and final energies of 30 kV and 1 kV, respectively. The resulting lamellae were less than 50 nm thick. HAADF-STEM imaging was performed on a probe corrected Hitachi HD 2700 STEM operated at 200 kV with a convergence semi-angle of 23.5 mrad.

The iDPC-STEM imaging was performed on a probe corrected Thermo Fisher Themis 80-300 kV STEM operated at 200 kV with a convergence semi-angle of 25.3 mrad using a four-quadrant DF4 detector[56].

For the elemental EDS maps of Ga, N and Mg, the measurements were taken using a Thermo Fisher Super-X system in the same STEM set-up as that used for the iDPC data. The Super-X system consists of four silicon drift detectors for improved X-ray collection efficiency and offers good sensitivity and high spatial resolution capability. A radial Wiener filter was used for the elemental EDS maps of Ga, N and Mg.

STEM-CL characterization involved sample preparation by FIB using a Thermo Fisher Scientific Helios 660 with lamella specimens approximately 500 nm thick. The STEM-CL measurements were done on a JEOL JEM-2100 F analytical electron microscope at an acceleration voltage of 120 kV and a beam diameter of 1.0 nm. High-resolution STEM observations associated with the STEM-CL measurements were performed on a JEOL JEM-ARM200F analytical electron microscope at an acceleration voltage of 200 kV and a beam diameter of 0.15 nm.

## Strain determination
For the in-plane and out-of-plane strains, they are defined by the following formulas, respectively:

$$\varepsilon_a(\%) = \frac{a_s - a_r}{a_r} \times 100 \tag{2}$$

$$\varepsilon_c(\%) = \frac{c_s - c_r}{c_r} \times 100 \tag{3}$$

where the subscript s refers to GaN in the strain-mapping region (GaN between periodic 2D-Mg$_i$ sheets excluding the tip region) and the subscript r denotes GaN in the reference region (outside the intercalation region). The terms $a$ and $c$ represent the in-plane and out-of-plane lattice constants of hexagonal GaN, respectively. These constants are obtained on the basis of the measured average distance from atom column to atom column. The average in-plane distance, $d_{\text{in-plane}}$, and out-of-plane distance, $d_{\text{out-of-plane}}$, have the following relationship with the lattice constants in a hexagonal lattice:

$$d_{\text{out-of-plane}} = \frac{1}{2}c \tag{4}$$

$$d_{\text{in-plane}} = \frac{\sqrt{3}}{2}a \tag{5}$$

The direction of the in-plane strain is parallel to the $m$ axis (the <1$\bar{1}$00> direction) and also equivalently to the $a$ axis (the <11$\bar{2}$0> direction) in this work, owing to the symmetry of strain produced by the Mg sheets parallel to both the $m$ axis and the $a$ axis. The direction of out-of-plane strain is parallel to the $c$ axis (the <0001> direction). The directions of the $m$, $a$ and $c$ axes are indicated in the lower left corner of Fig. 3b. When GaN is uniaxially strained along the $c$ axis, the Poisson ratio, $v$, is defined as:

$$v = -\frac{\varepsilon_\perp}{\varepsilon_\parallel} = -\frac{\varepsilon_a}{\varepsilon_c} \tag{6}$$

For atom-column by atom-column distance measurements, images were acquired using a frame-averaging approach to limit the effects of sample drift. Each frame-averaged dataset consisted of a minimum of 12 individual images that were then position-corrected using cross-correlation and averaged for the final image. To further limit the effects of drift, frame-averaged image series were acquired with the fast scan direction parallel to the measurement direction. The atom-column by atom-column measurements were performed by fitting each atom column position to a two-dimensional Gaussian distribution and measuring the distance between columns directly using the Atomap software package[57].

## Electronic band-structure calculation
The electronic band structures of GaN were calculated according to density functional theory (DFT) using the Vienna ab initio Simulation Package. The nuclei and core electrons were simulated by pseudopotentials generated by the projector-augmented-wave method. The exchange-correlation energy was treated in the generalized gradient approximation using the Perdew–Burke–Ernzerhof functional. In the electronic-structure calculations with the Heyd–Scuseria–Ernzerhof hybrid functional, the exact exchange amount was set to 26%, yielding a bandgap of 3.4 eV for unstrained wurtzite GaN, which aligns well with the experimental value for bulk GaN.

DFT simulations were not performed on the broader GaN and Mg superlattice for two main reasons. First, the MiGs structure, like the Guinier–Preston zone, is categorized as a structure rather than a compound material, the latter being exemplified by MAX phases. The non-stoichiometry of a structure greatly increases the computational expense of the relevant calculations. Merely constructing an Mg sheet, as depicted in Fig. 1e, would suggest that Mg sheets extend throughout the entire layer, which would result in substitutional rather than interstitial intercalation. Consequently, to honour the characteristic of interstitial intercalation, the Mg sheets must terminate within the GaN matrix. This necessitates a supercell comprising hundreds of atoms at least. Second, the gradual transition of polarity in the GaN lattice, as illustrated in Fig. 2f, substantially complicates the construction of a supercell model that properly reflects the unique features of such a structure. As such, DFT simulations on the GaN and Mg superlattice may require much more effort in future studies.

## Scanning electron microscopy (SEM)-CL mapping and polarization-resolved CL spectroscopy
SEM-CL was performed on an Allalin spectroscopic (analytical) platform from Attolight. A focused electron beam scanned the sample with normal incidence and the optical emission was collected. Angles rotationally symmetric between 18° and 46° from the incident

beam were captured. The optical beam was then analysed by using a spectrometer to obtain a luminescence spectrum for each pixel of the image. The optical analysis apparatus consisted of a monochromator fitted with a 1,024×256 charge-coupled device high-speed camera adapted for ultraviolet-visible detection (wavelengths of 200–1,100 nm). The camera enabled almost-instant acquisition of the entire emission spectrum. The samples were measured in both plan and cross-sectional views (for polarization-resolved CL only) at room temperature. The voltage of the incident electron beam was set to 10 kV. The scans were square and their size varied from 5 µm to 50 µm, depending on the width of the measured line on the sample, and the number of data points was kept at 128×128. The integration time at each pixel was set to 10 ms, resulting in a total acquisition time for one image of 165 s. Acquired luminescence spectra resulted from electronic transitions around the bandgap of GaN. After acquisition, the spectrum at each pixel was approximatively fitted with a Gaussian curve. Peak energy was determined from the Gaussian centre.

## THz-TDE

Free carrier absorption exists in the terahertz region, which allows the characterization of electrical conductivity[58]. This method is preferred to the conventional van der Pauw method because the latter only measures sheet resistivity (the inverse of in-plane conductivity, $\sigma_\perp$), whereas THz-TDE encompasses both in-plane ($\sigma_\perp$) and out-of-plane ($\sigma_\parallel$) components[59].

Our THz-TDE was performed using the Tera Evaluator, a commercial terahertz time-domain ellipsometer from Nippo Precision, with a reliable spectral range of 1.0–2.5 THz. The angle of incidence was fixed at 70° (ref. 58). To characterize the change in polarization of the terahertz waves after reflection on the sample, the ellipsometric parameters, namely amplitude ratio (tan $\Psi$) and phase difference ($\Delta$) between the $p$ and $s$ polarization components, were measured[59].

These ellipsometric parameters were initially measured from a bare sapphire substrate. This was followed by measurements on a 3.99-µm-thick unintentionally doped (lightly n-doped) GaN on the aforementioned sapphire substrate, and measurements of three 430-nm-thick p-type/3.99-µm-thick unintentionally doped (lightly n-doped) GaN/sapphire samples (one of them has incorporated MiGs nanostructures, formed by annealing of 50 nm metallic Mg film onto the entire GaN at 800 °C and 10 min). Three p-type GaN samples formed two control groups: the first consisted of p-type GaN samples with varying initial Mg doping concentrations ($7 \times 10^{18}$ cm$^{-3}$ and $2 \times 10^{19}$ cm$^{-3}$); the second comprised the lower-Mg-doped p-type GaN samples, with and without MiGs phases, named samples 1 and 2, respectively. The higher-Mg-doped GaN sample without MiGs phases was known as sample 3 (Extended Data Fig. 6a).

During measurements, the equipment chamber was purged with dry air to minimize terahertz absorption by water vapour, and the ambient temperature was maintained at 22 °C. The terahertz spot size on the sample surface is 10 mm for the major diameter and 3 mm for the minor diameter. For the analysis, a three-layer optical model, approximating the sample structure, was constructed. This model was then used to calculate the predicted response on the basis of Fresnel equations, which describe the light reflection at the interface between each layer, taking into account thickness and optical constants. The calculated values were subsequently compared with the experimental data, and the solution was optimized using regression analysis.

In the MiGs-incorporated p-type GaN epilayer, the measured optical and transport properties represent a composite of both p-type GaN and the MiGs phase. Fitting the conductivity spectra to the Drude model reveals the inverse of scattering time at the crossing point of the $\sigma'$ and $\sigma''$ curves (Extended Data Fig. 6e). Sample 3 had a shorter scattering time, $\tau$, and higher conductivity, $\sigma$, consistent with its higher ionized impurity concentration. However, the scattering time for sample 2 was similar to that of sample 1. The sheet resistivity and hole concentration for these samples, also measured by the van der Pauw method, are summarized in Extended Data Fig. 6i for comparison. We observed that the resistivity measured by THz-TDE, denoted as $\rho_{DC}$, of sample 1 and sample 3 closely aligns with their sheet resistivity measured using the van der Pauw method, validating the optical model used in our THz-TDE analysis. By contrast, sample 2 exhibits a substantial decrease in $\rho_{DC}$ compared with its sheet resistivity. The six-fold conductivity increase suggests there are important factors beyond the increased hole concentration, minor changes in the dielectric constant and surface roughness. Given the consistent scattering time in samples 1 and 2, it implies that sample 2 has greatly enhanced out-of-plane mobility, potentially resulting from a reduced effective mass from the predominance of split-off holes along the $c$ axis.

Based on the Drude model, in which conductivity ($\sigma$) is proportional to carrier concentration and scattering time ($\tau$) and inversely proportional to effective mass, our preliminary estimates (using data from Extended Data Fig. 6i) indicate that the average effective mass of holes in sample 2 is reduced to approximately 30% of that in sample 1.

For more specific details, in THz-TDE, $p$- and $s$-polarized terahertz waves were irradiated onto a sample at an oblique angle, and the physical properties of the sample are derived from the change in the polarization state of terahertz waves on reflection. The measured values (tan $\Psi$ and $\Delta$) are defined from the ratio of the reflection coefficients for $p$ and $s$ polarizations, $\tilde{r}_p$ and $\tilde{r}_s$:

$$\tilde{\rho} = \tan\Psi \exp i\Delta = \frac{\tilde{r}_p}{\tilde{r}_s} = \left(\frac{\tilde{E}_{r,p}}{\tilde{E}_{i,p}}\right) \bigg/ \left(\frac{\tilde{E}_{r,s}}{\tilde{E}_{i,s}}\right) \tag{7}$$

where $\tilde{E}_r$ and $\tilde{E}_i$ represent the reflected and incident electric fields, respectively. In the measurement, the incident field is oriented such that $\tilde{E}_{i,p} = \tilde{E}_{i,s}$. Hence, equation (7) can be simplified to $\tilde{\rho} = \tan\Psi \exp i\Delta = \tilde{E}_{r,p}/\tilde{E}_{r,s}$, and it follows that

$$\tan\Psi = \frac{|\tilde{E}_{r,p}|}{|\tilde{E}_{r,s}|} \text{ and } \Delta = \delta_{r,p} - \delta_{r,s} \tag{8}$$

where $\delta$ represents phase. Therefore, tan $\Psi$ and $\Delta$ represent the amplitude ratio and phase difference, respectively, between the $p$ and $s$ polarizations. These parameters are referred to as ellipsometric parameters.

The reflection coefficients for $p$ and $s$ polarizations are described by Fresnel equations:

$$\tilde{r}_{jk,p} = \frac{\tilde{n}_k \cos\theta_j - \tilde{n}_j \cos\theta_k}{\tilde{n}_k \cos\theta_j + \tilde{n}_j \cos\theta_k}$$
$$\tilde{r}_{jk,s} = \frac{\tilde{n}_j \cos\theta_j - \tilde{n}_k \cos\theta_k}{\tilde{n}_j \cos\theta_j + \tilde{n}_k \cos\theta_k} \tag{9}$$

where the subscripts $j$ and $k$ represent the media of light propagation, $\tilde{n}$ is the complex refractive index expressed by $\tilde{n} = n - i\kappa$, $\theta_j$ is the angle of incidence and $\theta_k$ is the transmission angle, which can be calculated using Snell's law.

For a bulk sample that consists of only a single layer, the complex refractive index can be obtained directly from the measured ellipsometric parameters:

$$\tilde{n}_1 = \tilde{n}_0 \sin\theta_0 \sqrt{\left(\frac{1 - \tilde{\rho}}{1 + \tilde{\rho}}\right)^2 \tan^2\theta_0 + 1} \tag{10}$$

where $\tilde{n}_0$ is the complex refractive index of the ambient (air) and $\theta_0$ is the angle of incidence.

For a thin film/substrate structure, the complex refractive index of the film can be obtained numerically using the following expression:

$$\tilde{\rho} = \frac{\tilde{r}_p}{\tilde{r}_s} = \left[ \frac{\tilde{r}_{01,p} + \tilde{r}_{12,p}\exp(-i2\beta)}{1 + \tilde{r}_{01,p}\tilde{r}_{12,p}\exp(-i2\beta)} \right] \bigg/ \left[ \frac{\tilde{r}_{01,s} + \tilde{r}_{12,s}\exp(-i2\beta)}{1 + \tilde{r}_{01,s}\tilde{r}_{12,s}\exp(-i2\beta)} \right] \quad (11)$$

where $\beta$ represents the film phase thickness given by $\beta = 2\pi d\tilde{n}_1\cos\theta_1/\lambda$, $\lambda$ is the wavelength and $d$ is the film thickness.

For a three-layer sample structure, the ellipsometric parameter can be expressed as:

$$\tilde{\rho} = \frac{\tilde{r}_p}{\tilde{r}_s} = \left[ \frac{\tilde{r}_{01,p} + \tilde{r}_{123,p}\exp(-i2\beta_1)}{1 + \tilde{r}_{01,p}\tilde{r}_{123,p}\exp(-i2\beta_1)} \right] \bigg/ \left[ \frac{\tilde{r}_{01,s} + \tilde{r}_{123,s}\exp(-i2\beta_1)}{1 + \tilde{r}_{01,s}\tilde{r}_{123,s}\exp(-i2\beta_1)} \right] \quad (12)$$
$$= \tan\Psi\exp i\Delta$$

where $\tilde{r}_{01}$ represents the reflection coefficient for the ambient–first layer interface and $\tilde{r}_{123}$ is the reflection coefficient for the second layer and substrate, and is in turn given by

$$\tilde{r}_{123} = \frac{\tilde{r}_{12} + \tilde{r}_{23}\exp(-i2\beta_2)}{1 + \tilde{r}_{12}\tilde{r}_{23}\exp(-i2\beta_2)} \quad (13)$$

The measured ellipsometric parameters for the sapphire substrate and n-type GaN/sapphire sample are shown in Extended Data Fig. 6f. Using equations (10) and (11), the complex refractive index of sapphire and of the n-type GaN film were obtained, respectively, as shown in Extended Data Fig. 6g. The extinction coefficient of the n-type GaN film shows an increasing trend towards low frequencies, which can be attributed to free carrier absorption. Extended Data Fig. 6h shows the complex refractive-index spectra obtained for the p-type GaN layers in the p-GaN/n-GaN/sapphire samples using equation (12). The spectra are fitted to the Drude model. In these measurements, the optic axis of the samples is normal to the surface.

It is important to note that the optical constants were calculated under the assumption of an isotropic model, despite GaN being a uniaxial crystal. Thus, the solutions are only an approximation of the dielectric function and do not represent exact optical constants, although they can be more appropriately regarded as pseudo-dielectric functions. Regardless, they provide insights into the differences between the optical responses of samples with and without MiGs phases and offer semi-quantitative evidence of an enhancement in conductivity in Mg-intercalated GaN.

## Van der Pauw method

After the THz-TDE measurements, Hall-effect measurements (the van der Pauw method) at room temperature were performed on the above-mentioned p-type GaN samples (ResiTest8400 system) using a magnetic field with an amplitude of 0.5 T. The samples were diced into small pieces and the van der Pauw pattern for each measurement consisted of 1.5 mm × 1.5 mm. Ni/Au electrodes were formed by electron-beam evaporation. The contact metal was annealed at 525 °C for 5 min in $O_2$ ambient to reduce the contact resistance.

## SIMS

SIMS was performed at Eurofins EAG Laboratories. For backside SIMS measurement, samples were first polished and thinned to approximately 1 μm before being sputtered from the backside. $Cs^+$ and $O^-$ primary ion beams were used to detect negative secondary ions (from C, H, O and Si) and positive secondary ions (Mg), respectively.

## Electrical characterizations

For $C$–$V$ profiling, Ni (70 nm)/Au (70 nm) stacks were electron-beam evaporated to form circular Schottky-contact electrodes (around 220 μm in diameter) without sintering. A shadow mask was used for patterning, chosen over photolithography to prevent potential damage from developing agents to the sample surface. Ti (20 nm)/Au (100 nm) stacks were electron-beam evaporated on the backside of the n-type

GaN substrates for ohmic contact without sintering. All $C$–$V$ measurements were performed at a frequency of 100 kHz using an E4980A LCR meter by Agilent.

For measurement of $I$–$V$ characteristics combined with STEM observations, the samples were prepared with half of the region blank deposited with Mg and the other half was patterned with Mg using maskless photolithography (Nanosystem Solutions DL-1000). After this, the samples underwent the previously mentioned annealing process and were subsequently cleaned with acid. In the next step, using maskless photolithography, Ni (20 nm)/Au (150 nm) was electron-beam evaporated onto regions approximately identical to those where Mg had previously been deposited. This was done with the aid of alignment photolithography, achieving a precision of around 2 μm, to ensure the absence of Mg in the regions between the Ni/Au electrodes. Finally, sintering was carried out at 525 °C for 5 min in an $O_2$ atmosphere to form the ohmic contact. After this, the samples were diced such that the half entirely covered with Mg-annealed surface was designated for STEM observations, whereas the other half, with selectively patterned Mg, was used for evaluating ohmic contact resistance through $I$–$V$ characterization. An Agilent B1505A analyser was used to measure $I$–$V$ characteristics in this work.

## Data availability

All data generated or analysed during this study are included in the published article and its Supplementary Information. Source data are provided with this paper.

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

**Acknowledgements** This work was partly supported by Japan Science and Technology Agency as part of the Adopting Sustainable Partnerships for Innovative Research Ecosystem (ASPIRE) programme, grant JPMJAP2311, and the Advanced International Collaborative Research Program (AdCORP), grant JPMJKB2303. It was also partly supported by the Ministry of Education, Culture, Sports, Science, and Technology (MEXT) of Japan under the Program for Creation of Innovative Core Technology for Power Electronics, grant JPJ009777, the METI Monozukuri R&D Support Grant Program for SMEs, grant JPJ005698, and JSPS KAKENHI, grant JP24H00317. B.S. acknowledges the JSPS Invitational Fellowship (award L22558). We thank H. Dycus for assistance with STEM observations and analysis, which led to the discovery of the MiGs structure; Y. Zhang and C. Addiego for the iDPC-STEM characterizations and analyses; N. Kawasaki, S. Hyashi and H. Sako for STEM-CL and STEM-EELS characterizations and analyses; M. Fouchier and C. Monachon for high-resolution CL mapping and polarization-dependent CL spectroscopy characterizations and analyses; X. Zhou for assistance with the annealing of Mg in an Ar atmosphere; Y. Lin, T. Kumabe and X. Li for assistance with various experiments; and D. Jena, H. G. Xing, Y.-H. Xie, J. Marian, K. L. Wang, T.-Y. Seong, K. Shiraishi and T. Kumabe for discussions and guidance.

**Author contributions** J.W. conceived the idea of annealing metallic Mg on GaN. J.W. and H.A. coordinated the design of the characterizations. S.L. developed the methodology for the processing route related to Mg annealing. W.C. and H.W. designed and conducted the GaN epitaxial growths. J.W. and B.S. conducted the Mg-annealing experiments. J.W., W.C. and W.L. analysed the STEM and CL measurement data. E.K. and N.I. performed some of the HAADF-STEM observations. J.W. and S.L. carried out DFT simulations. V.C.A., T.I. and M.N.

performed THz-TDE characterizations and analysed the data. B.S. carried out the $C–V$ measurements and analysed the data. J.W. performed $I–V$ measurements and Hall-effect measurements. J.W. led the analysis of experimental data. Y.H. provided resources and assistance for the experiments. J.W., W.C., W.L., V.C.A. and B.S. edited figures and wrote the manuscript. H.A. supervised the project. All authors contributed to the discussion of the results and revision of the paper.

**Competing interests** J.W. and H.A. have filed a patent application directly related to the content of this paper (application number: 2023-011149; application date: 27 January 2023; title of invention: 'Semiconductor device and method for manufacturing semiconductor device'). The authors acknowledge that this constitutes a potential competing interest and have fully disclosed the nature of this interest in accordance with the relevant policies. The other authors declare no competing interests.

**Additional information**
**Correspondence and requests for materials** should be addressed to Jia Wang or Hiroshi Amano.

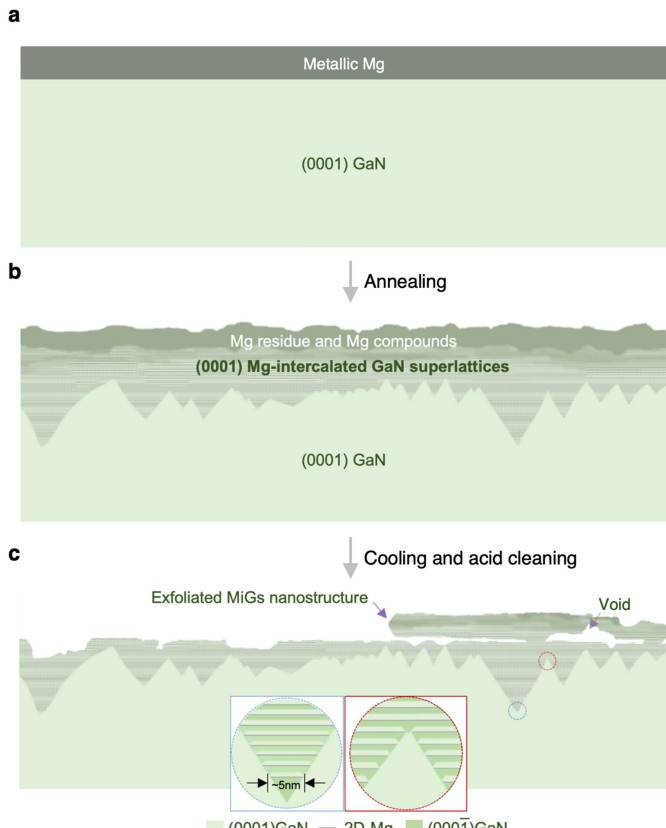

**Extended Data Fig. 1 | Spontaneous formation of MiGs nanostructure by thermal annealing of metallic Mg on GaN. a**, Step-1: Deposition of metallic Mg film on (0001) GaN. **b**, Step-2: The intercalation behaviour of Mg sheets into GaN at elevated temperature. **c**, Step-3: Post-cooling, and removal of acid-soluble residues, associated with the random exfoliation of MiGs nanofilms. The STEM observations of the MiGs phases and discussions are in Extended Data Fig. 5 and Supplementary Note 3.

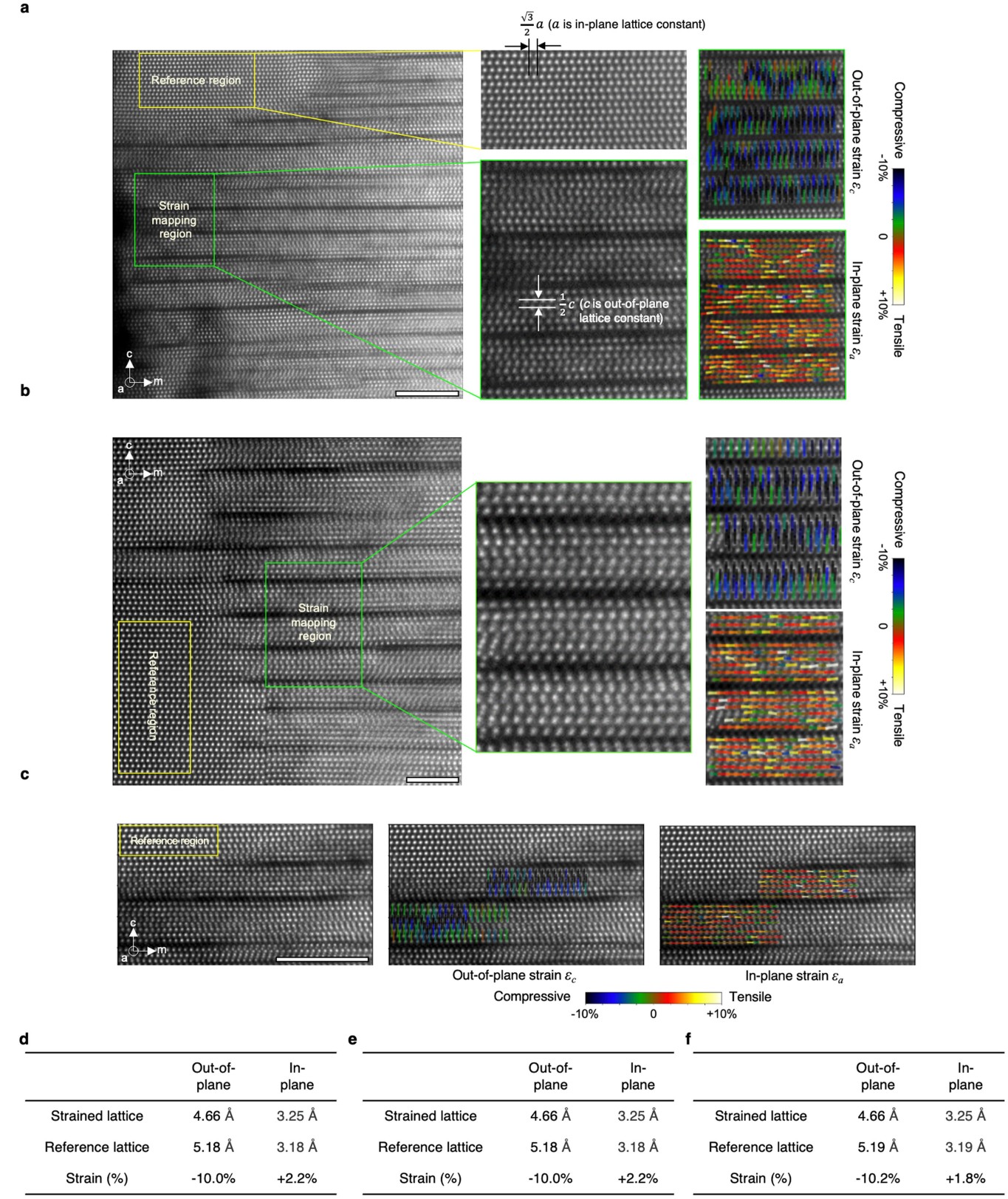

**Extended Data Fig. 2 | Strain mapping of GaN in the MiGs nanostructures.**
**a–c**, Multiple drift-corrected HAADF-STEM images containing a portion of MiGs structures in which GaN lattice is strained and a portion of unstrained GaN as reference regions outlined by yellow boxes. Green boxes highlight the areas where strain mapping was performed within the MiGs nanostructures. Magnified views of both the strain-mapped and reference regions are presented on the right side of each image. The maps for out-of-plane strain (perpendicular to Mg sheets) and in-plane strain (parallel to Mg sheets) are plotted based on atom column-by-atom column measurements (see Methods section for details). Scale bars, 5 nm. **d–f**, Summaries of the mean lattice parameters and strain values in **a–c**, respectively.

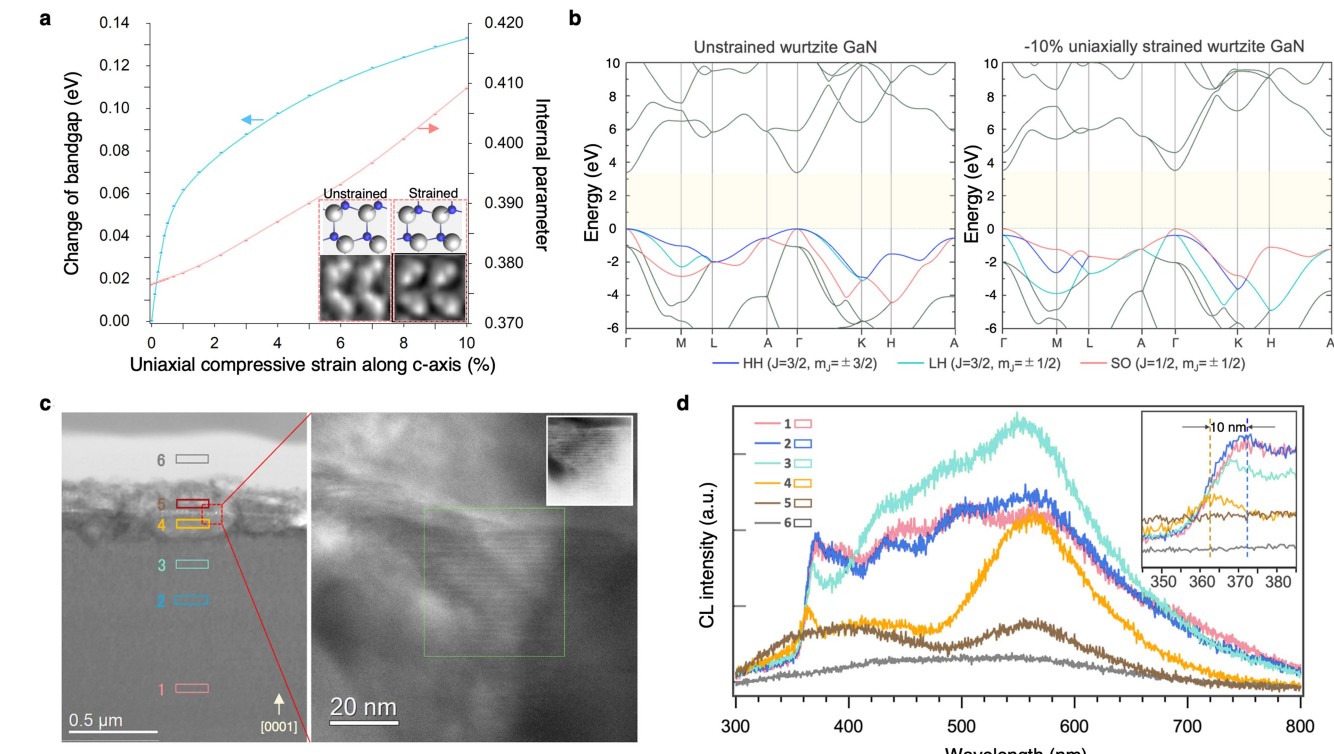

**Extended Data Fig. 3 | Changes in the valence band structure of strained GaN. a**, Calculated bandgap changes of wurtzite GaN and internal parameter under high uniaxial compressive strain. Insets depict schematic representations and iDPC-STEM images of unstrained GaN lattices outside the MiGs structure and strained GaN lattices within the MiGs structure. **b**, Calculated electronic band structures of unstrained wurtzite GaN (left) and −10% strained wurtzite GaN (right). **c**, STEM image of an FIB-prepared thick GaN lamella with MiGs structure distributed on the surface, where a magnified bright field (BF)-STEM image is placed on the right, with an inset showing a HAADF-STEM image of the area in the green dashed box of the BF-STEM image, confirming the presence of MiGs structures. Thicker lamella is required for STEM- cathodoluminescence (CL) analysis which compromises the STEM imaging resolution. **d**, STEM-CL spectra comparing photoemissions from nano-scale sampling locations far from, close to, and inside the MiGs structures formed by annealing metallic Mg deposited on an n-type GaN substrate. Positions of sampling locations are indicated in **c** accordingly.

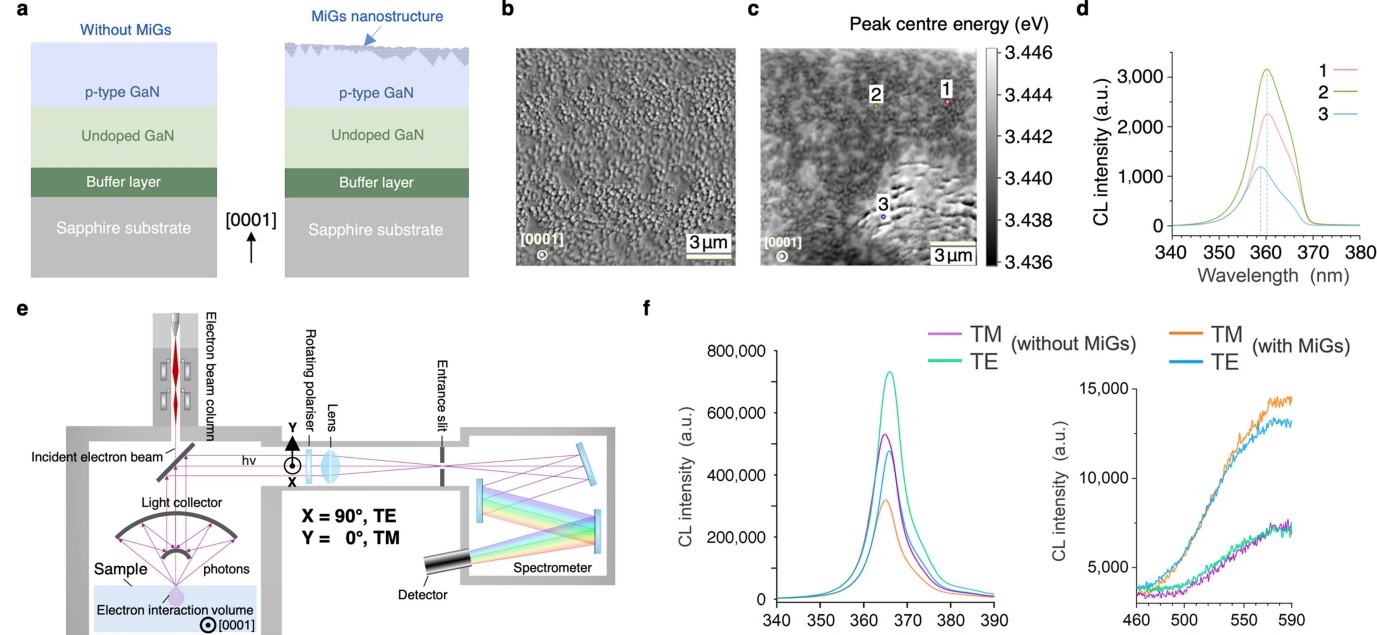

**Extended Data Fig. 4 | Plan-view scanning electron microscopy (SEM)-CL and cross-sectional polarization-resolved SEM-CL spectra of Mg-intercalated p-type GaN. a**, Cross-sectional schematic illustrations of the p-type GaN with and without MiGs nanostructures. The p-type GaN was initially grown by MOVPE, featuring in-situ Mg doping at a density of $6 \times 10^{18}$ cm$^{-3}$. **b**, SEM image (in secondary electron mode) displaying the plan-view morphology of a typical Mg-annealed GaN surface (surface of the p-GaN with MiGs nanostructure). **c**, Integrated CL spectra image mapping the position of peak centre energy around the near-band-edge (NBE) emission. The imaged region corresponds to that shown in **b**.

**d**, Localized spectra of NBE at the three sampling points indicated in **c**. **e**, Schematic illustration of the polarization-resolved CL spectroscopy setup (0° and 90°). **f**, Polarization-resolved CL spectra of the p-type GaN samples with and without MiGs nanostructures. This shows the CL intensities of transverse magnetic (TM) (0° polarization) and transverse electric (TE) (90° polarization) emissions in both the NBE band and yellow luminescence band. The p-type GaN with MiGs nanostructures demonstrates an increased TM emission in the yellow luminescence band.

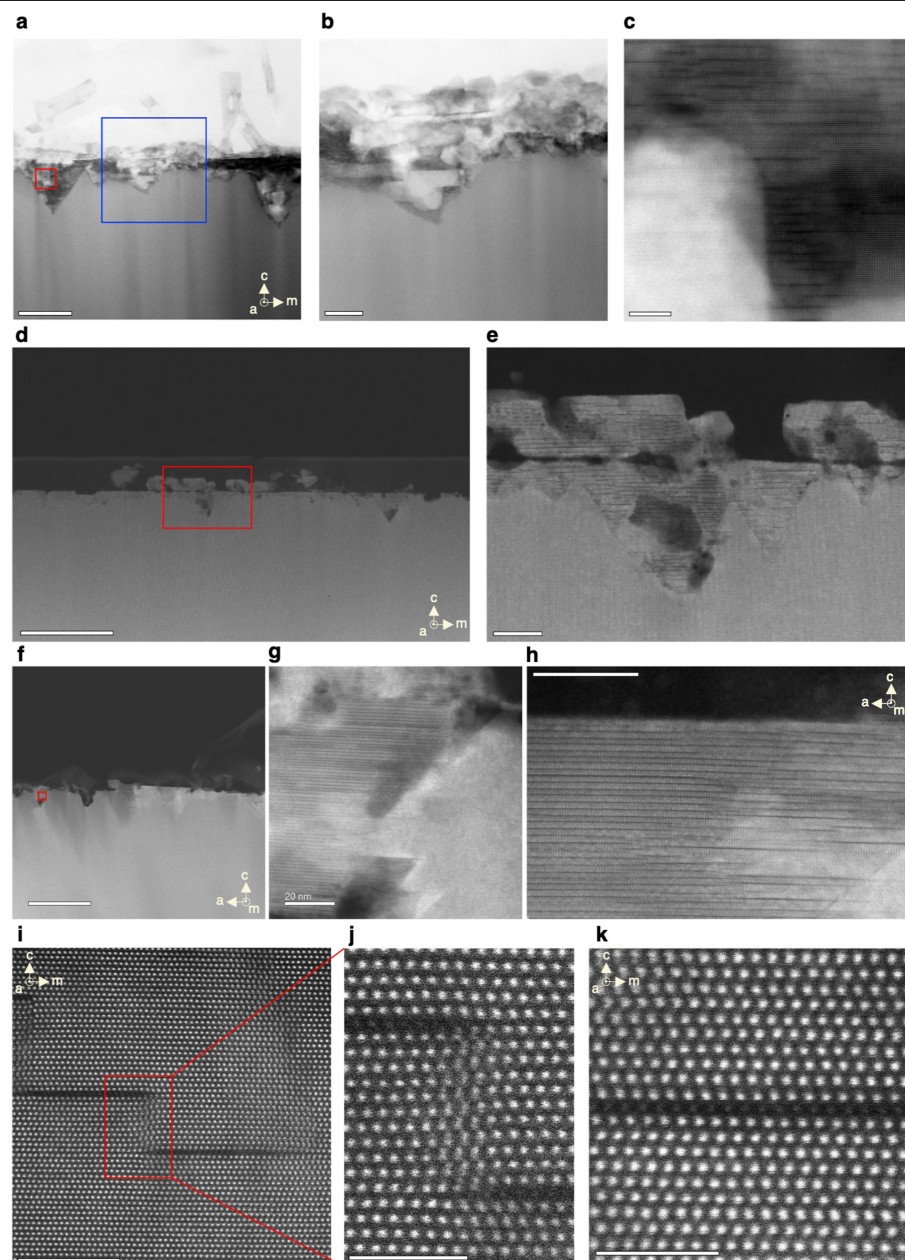

**Extended Data Fig. 5 | STEM observation of Mg intercalation into GaN.**
**a**, Annular dark field (ADF)-STEM image of Mg-annealed GaN (in default annealing condition of 800 °C for 10 min). Scale bar, 200 nm. **b**, Magnified view of the area within the blue box in **a**, revealing the amorphous phase comprising a mixture of Mg and Mg compounds. Scale bar, 50 nm. **c**, Magnified view of the area within the red box in **a**. Scale bar: 5 nm. **d**–**f**, HAADF-STEM images of the acid-cleaned, Mg-annealed GaN under the same annealing conditions, showing that the amorphous phase can be removed by acid clean. The MiGs phases forming giant inverted pyramids are distinctly visible, along with the observable exfoliation of the MiGs phase at the top, where **e** is the magnified view of the region within

the red box in **d**. Scale bars: 200 nm, 20 nm, 500 nm, respectively. **g**, Magnified view of the region in the red box in **f**. Scale bar: 20 nm. **h**, HAADF-STEM image showing a detailed MiGs structure. Scale bar: 20 nm. **i**, Atomically resolved image illustrating two Mg intercalant sheets before vertical alignment, with each sheet approximately 5 nm in size, constrained by the inversion domain region volume. Scale bar, 5 nm. **j**, A magnified view of the region in the red box in **i**. Scale bar, 2 nm. **k**, Atomically resolved HAADF-STEM image along the [11$\bar{2}$0] zone axis (*a*-axis), revealing the Ga atom layer (ABAB ordering) with an inserted single Mg monolayer (C sites), resulting in the ABCAB ordering. Scale bar, 2 nm.

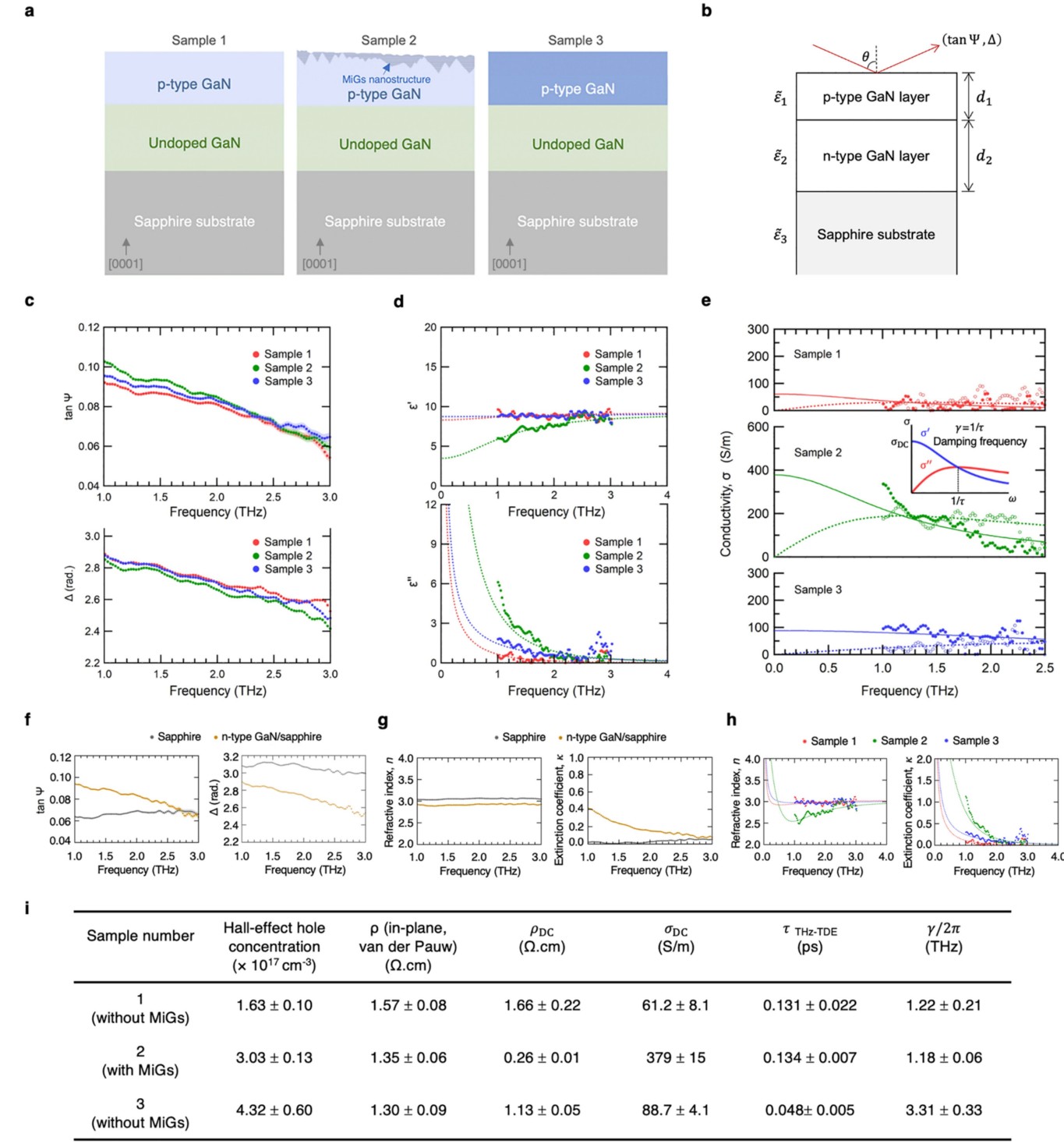

| Sample number | Hall-effect hole concentration ($\times 10^{17}$ cm$^{-3}$) | $\rho$ (in-plane, van der Pauw) ($\Omega$.cm) | $\rho_{DC}$ ($\Omega$.cm) | $\sigma_{DC}$ (S/m) | $\tau$ THz-TDE (ps) | $\gamma/2\pi$ (THz) |
|---|---|---|---|---|---|---|
| 1 (without MiGs) | $1.63 \pm 0.10$ | $1.57 \pm 0.08$ | $1.66 \pm 0.22$ | $61.2 \pm 8.1$ | $0.131 \pm 0.022$ | $1.22 \pm 0.21$ |
| 2 (with MiGs) | $3.03 \pm 0.13$ | $1.35 \pm 0.06$ | $0.26 \pm 0.01$ | $379 \pm 15$ | $0.134 \pm 0.007$ | $1.18 \pm 0.06$ |
| 3 (without MiGs) | $4.32 \pm 0.60$ | $1.30 \pm 0.09$ | $1.13 \pm 0.05$ | $88.7 \pm 4.1$ | $0.048 \pm 0.005$ | $3.31 \pm 0.33$ |

**Extended Data Fig. 6** | See next page for caption.

**Extended Data Fig. 6 | Terahertz time-domain ellipsometry (THz-TDE) on the MiGs-incorporated p-type GaN. a**, Schematic illustrations of three samples, each with a 400 nm-thick p-type GaN epilayer, are shown. These form two control groups: one varying in-situ Mg doping concentration (Sample 1 vs. 3) and the other comparing samples with and without MiGs nanostructures (Sample 1 vs. 2). **b**, Schematic illustration of the samples featuring a three-layer structure: p-type GaN/n-type GaN/sapphire, irradiated by incoming terahertz electromagnetic waves. Each layer is denoted as 1, 2, and 3, respectively. The thickness parameters, $d_1$ and $d_2$, are 400 nm and 3.9 µm, respectively. The optical constants, $\tilde{\varepsilon}_2$ and $\tilde{\varepsilon}_3$, were obtained from separate measurements, elaborated in the Methods section. The optical model approximates the p-type GaN layer with MiGs nanostructure on top as a single layer, with the acquired optical constants for $\tilde{\varepsilon}_1$ of Sample 2 assumed to be an average response of layer 1. **c**, Ellipsometric parameters (tan Ψ, Δ) measured by THz-TDE. The shading corresponds to the uncertainty in the measured values. The uncertainty region is not visible as it is significantly smaller than the measured values, with a standard deviation of less than 0.003 for the ellipsometric parameters. **d**, Real ($\varepsilon'$) and imaginary ($\varepsilon''$) parts of the relative permittivity obtained for the p-GaN layers, denoted as $\tilde{\varepsilon}_1$, are complex relative permittivity of the top, represented as $\varepsilon' - i\varepsilon''$. The increase in $\varepsilon''$ at lower frequencies is attributed to THz absorption by free carriers. A higher $\varepsilon''$ indicates greater absorption and, consequently, higher conductivity. **e**, Real (solid circles) $\sigma'$ and imaginary (hollow circles) $\sigma''$ parts of the conductivity $\tilde{\sigma}$, derived from the relative permittivity spectra of the p-GaN layers, are represented as $\tilde{\sigma} = \sigma' - i\sigma'' = -i\omega\varepsilon_0 (\varepsilon_{s,\mathrm{GaN}} - \tilde{\varepsilon})$. The fitting range is set within 1.0–2.5 THz to exclude noise at higher frequencies, which is not shown. The best fit results from a global analysis of the real and imaginary components. By fitting the conductivity spectra to the Drude model, the DC conductivity $\sigma_{\mathrm{DC}}$ (translated to static resistivity $\rho_{\mathrm{DC}}$), the scattering time $\tau$, and the damping frequency $\gamma$ can be estimated by $\tilde{\sigma} = \sigma_{\mathrm{DC}}/(1 + i\omega\tau)$, as summarized in **i**. **f**, Ellipsometric parameters (tan Ψ, Δ) measured from the sapphire substrate and the n-type GaN/sapphire sample. **g**, Complex refractive index $\tilde{n} = n - i\kappa$ obtained for sapphire and the n-type GaN layer. **h**, Complex refractive index spectra obtained for the p-type GaN layers and fitted to the Drude model. **i**, Summary and comparison of the measured transport properties by THz-TDE and Hall-effect van der Pauw methods.