## [Peer Review file · Nature]

Manuscript Title: Observation of 2D-magnesium-intercalated gallium nitride superlattices

Reviewer Comments & Author Rebuttals

Reviewer Reports on the Initial Version:

Referees' comments:

Referee #1 (Remarks to the Author):

It is well known that p-type doping of gallium nitride is difficult: holes are in short supply, and have low mobilities. In spite of this, the discoveries (including by the group of an author of this manuscript) in early 1990s on doping with magnesium, and subsequent annealing was able to make the first p-type GaN, which enabled visible and blue LEDs, and blue lasers. Polarization-induced doping has recently been successful in enabling high-performance p-channel FETs by providing very high mobile hole densities, and deep-UV lasers by distributed polarization doping. But the problem of low hole mobilities remain.

The authors report a rather interesting experimental discovery in this manuscript, which corroborates a theoretical prediction of a clever way to increase the hole mobility in gallium nitride. What is rather remarkable is the manner in which this is achieved. At the outset, it appears to be an unexpected finding. They see that by depositing Mg *ex situ* on the surface of a Mg-doped GaN layer, and annealing it at optimal temperatures, the deposited Mg mixes with the GaN in an ordered way. It forms a multilayer structure, with nearly 2D Mg layers intercalated between GaN monolayers. They track the reason for this observation to the fact that the *c*- and *a*-lattice constants of GaN, a hexagonal wide-bandgap semiconductor, and Mg, a hexagonal metal, are nearly the same. The intercalation seems to squeeze the *c*-lattice constant and stretch out the *a*-lattice constant, causing a very large strain the GaN monolayers between the Mg 2D layers.

A recent theoretical prediction had proposed that if a suitable strain could be applied to GaN such that the ordering of the valence band could be reversed: i.e., if the split-off band could be raised to the top, a large increase of hole mobility would result. The authors perform polarization-sensitive cathodoluminescence measurements which shows a strong enhancement of transverse magnetic (TM) polarization after the Mg intercalation: the emission was TE-dominant before intercalation. The TM dominance indicates the desired valence band reordering is likely achieved. They then measure the hole mobility by Hall effect and find a large enhancement in the mobility, from ~2X at 300 K to ~3X at 150 K. They measure a hole mobility of ~250 cm²/V.s at 250 K. Though the hole densities are low at this temperature (~2e13/cm³), these are unambiguously the highest hole mobilities measured at this temperature. The record of hole mobility in GaN stands at ~280 cm²/V.s in a 2D hole gas (see APL 119, 162104 (2021), which the authors may need to cite for comparison), but that is achieved at ~10 K (as opposed to ~150 K here). So the evidence of enhancement of hole mobility seems to be solid.

The authors present the data comprehensively, with structural, optical, and electronic

characterization, and theoretical modeling and analysis. The work is of high interest to the community. To refine the presentation, here are a few recommendations:

1. The first question a reader may have in their minds is: if instead of a p-doped epitaxial GaN layer, you started with a UID GaN layer, and deposited Mg ex situ, and repeated the above study, what do you get? Perhaps the authors have already performed this control experiment. If so, it would be helpful for the readers to be aware of the result - if not in the main text, at least in the supplementary materials.
2. Please use different colors in Fig 2 (c), the text in the TEM is not clearly visible.
3. Please state how the bandstructures in Fig. 3 (a) were calculated (e.g. k.p or pseudopotential or tight-binding, or DFT...). Also, it would be useful to quantify the strain by also including the pressure (e.g., MPa).
4. Some citations are out of order (e.g. in Page 11, line 6 reference 32 is not the correct one, and Page 13, line 30, reference 39 is missing). Please check and fix the reference citation numbers.
5. Page 13, line 13: Since GaN is a uniaxial crystal, the crystal-field splitting of LH and HH should remove all degeneracies at the gamma point in the bandstructure, even though the LH/HH splitting is small - this should be the case even without strain. Please check and edit the sentence if required.
6. Page 22, line 6: SIMS - the last word should be Spectrometry (not spectroscopy).
7. A short discussion on the feasibility of achieving the Mg intercalation uniformly rather than in the pyramidal regions shown in the supplementary materials will be useful for the readers.
8. In addition to the scattering mechanisms of holes mentioned (e.g. phonons), neutral impurity scattering should be considered in the discussion, since only a small fraction of the Mg acceptors are providing holes, and thus the rest are neutral and are in the path of the mobile holes, and cause scattering.

Referee #2 (Remarks to the Author):

In this manuscript, the authors report the intercalation of magnesium (Mg) into gallium nitride (GaN), an important wide-bandgap semiconductor that unleashes the era of blue-light LEDs, to produce the unique two-dimensional (2D) Mg-GaN superlattice through a simple metallurgical reaction at high temperatures. The results show that the intercalated sheets would generate high compressive elastic strains to the surrounding matrix, thus the so-called 2D Mg-GaN heterostructure can have a modified electronic band structure (wavelength), as well as a considerably increased (>200%) hole mobility (a long-sought goal for p-type GaN) due to the crystal-field splitting reversal in the hexagonal lattice (with small c/a value upon compressive stress). This is an interesting piece of research that build 2D metal into a covalent-ionic thin film semiconductor to achieve strain engineering effect, which could be potentially useful for future device applications (such as in

optoelectronics and power electronics). But there are some critical issues about the fundamental mechanisms and experimental details that need to be clarified:

First, the authors claimed the first-of-its-kind observation of intercalation of 2D-metal into bulk semiconductor. According to the references mentioned by the authors, however, Mg atoms can be implanted into GaN and even create pyramidal inversion domains, thus a more systematic and detailed explanation of the differences between the result in this work and earlier literatures is needed (for example, the 2D intercalated layers better introduce the reversal of crystal-field splitting..). Besides fundamental interests, the authors should demonstrate the benefits of the 2D-Mg intercalated GaN above the previous Mg doped GaN in applications and performances (for instance, the Ref. Appl. Phys. Express 12, 031004 (2019) has a Figure 2 that seems comparable to the results in this work..). On the other hand, about the detailed intercalated superlattice structure itself, the atomically resolved EDS (Fig. 1e-h) may not sufficiently prove the intercalated sheets are pure Mg, instead of Mg-Ga-N alloys or (Mg, Ga) nitride precipitates. Moreover, how did the authors obtain the atomic arrangement of the tip-edge of “Mg intercalant sheet” in Fig. 1d? The HAADF-STEM images may not resolve N columns due to their lower atomic number. It can be expected that the interface and the tip-edge might have different atom structures. Therefore, it would be ideal to resolve all Ga, Mg and N atoms by performing annular bright field or integrated differential phase contrast STEM imaging (Ultramicroscopy 160 (2016) 265–280), which can resolve all Ga, Mg and N atoms, for determining the atomic arrangement at the tip-edge of Mg intercalant sheet.

Second, while it's very intriguing to achieve this impressive superlattice structure through a simple metallurgical reaction, how the two-dimensional (2D) Mg layers were formed (the critical conditions and physical mechanisms for the formation of such 2D Mg structures in GaN, especially monolayer)? What is the area of intercalated Mg layers and how to control the sizes of these 2D Mg layers? Can the density/spacing of Mg intercalation layers between the multi-layer GaN be controlled? There seems a lack of essential experimental evidence and theoretical analysis here. To provide further insight into the related mechanism, numerical simulations are suggested, including the dynamic intercalation process of 2D-Mg into GaN layers using MD or Ab initio MD methods. Also, how about the mechanical and thermal stability of the 2D Mg-GaN heterostructures (when sustain a high elastic stress). Will there be further structural evolution or diffusion of the intercalated 2D Mg layers in GaN if there is an external field or after a long time? BTW, according to Fig. 1a, Mg layers were only found/intercalated on the surface of bulk GaN, as evidenced in Fig. S1. If and how can the depth of intercalation be controlled? Is there a method to construct an intercalated heterostructure that is more uniform? as these above critical issues may limit the practical device applications...

Third, by measuring the lattice parameters using the HAADF-STEM images the authors stated that the intercalation led to high elastic compressive strain in multi-layered GaN without causing volume expansion. Here the claim of “no volume expansion” may need double check and verified... In addition, as demonstrated in Fig. 2, the 2D Mg layers are intercalated unevenly, thus the compressive strain produced may not be entirely uniform; and at the tip-edge, the sheet may also result in different strain states. The authors are suggested to perform lattice strain mapping by geometric phase analysis (Ultramicroscopy 74, 131–146 (1998)), and the strain values (up to “10%”) should be also better verified. Furthermore, it's generally accepted that the modified optical and electronic properties of GaN are mainly attributed to high elastic compressive strains induced by

“Mg intercalant monolayer”, however, the intercalation also results in the multi layered structure, whether these multiple Mg monolayers and their spacing/density may have an effect on the measured cathodoluminescence spectra and/or hole mobility? Also consider the effect of non-uniform compressive strain near tip edge on the measured physical properties... Part of the reason for the above questions is that, recent numerical and experimental studies on elastic strain engineering suggest that substantial band gap changes can be found when elastic strain is ultra large (such as close to 10%), while here the CL results in Fig. 3 show relatively modest change in bandgap/wavelength shift (compare to the significantly increased hole mobility due to smaller c/a) under the claimed "10%" compressive elastic strain here... A few theoretical/DFT calculation on the electronic bandgap of the intercalated/strained GaN structure may help to explain.

A few minor points for authors' consideration and correction:

1. What does the abbreviation ML stand for? The reviewer cannot find the definition of ML throughout the manuscript.
2. The samples for STEM characterizations were processed at temperatures of 500, 550, 700, and 850 °C. However, the samples for SIMS experiments were processed at temperatures of 500, 600, 700, and 800 °C. How to determine and why not use the same temperatures?
3. “Fig. S5” in the supplementary information should be “Fig. S1”. And what are high-density nano-sized white regions in Fig. S1d?
4. Page 9 Line 15 “A relatively low acceleration voltage (3 kV) electron beam was utilized to collect signals from the shallow region (~50 nm) near the surface so that more optical emissions are expected from the Mg-intercalated GaN layers” How to ensure the optical emissions come only from the shallow region (the thickness of the active layer)?
5. The elastic modulus of GaN and Mg in Table 1 appears wrong. The two values should be switched (not sure if the table is really needed in Main text paper)
6. Is it possible to extend the 2D metal intercalation to other semiconductor systems, such as those in diamond cubic lattice? or hexagonal structure is more preferred?
7. Ref. 15 and Ref. 31 are repeated.

Referee #3 (Remarks to the Author):

The manuscript entitled "Observation of 2D-magnesium-intercalated gallium nitride superlattices" by Wang et al. discusses the intercalation of 2D-magnesium inside GaN at ambient pressure, resulting in a large compressive stress inside GaN. Such large stress leads to high hole mobility due to the reversal of the sign of the crystal field splitting. The authors also state that the 2D-Mg/GaN structure can be considered as a spontaneously formed metal/semiconductor superlattice and could serve as a novel probe for such heterostructures transport measurements.

The demonstration of metallic (Mg) intercalation inside a mixed ionic/covalent material like GaN is very interesting. However, I am unsure if one can call it a "first-of-its-kind observation" for a ceramic material (please see below). In addition, the increase in hole mobility due to the reversal of the sign of the crystal field splitting has been long desired and will be greatly appreciated.

Overall, I find that the manuscript will be interesting for (a) studies on the intercalation of metals inside ionic/covalent semiconductors, (b) increase in GaN's hole mobility due to compressive strain and reversal of the sign of crystal field splitting, and (c) as a system for transport studies on metal/semiconductor superlattices. However, the manuscript also raises a significant number of questions that require detailed answers. The authors should carefully address the following comments/questions before the manuscript is considered for publication in Nature.

1. Intercalation of noble metals (such as Au, Ir) has been used previously inside non van der Waals ceramic materials. For example, Ti_3AuC_2 , Ti_3IrC_2 , etc. are synthesized from Ti_3SiC_2 (Nature Materials, 16, 814 (2017)) or the silicide formation (that the authors also mention). Though the present study uses metal intercalation inside an III-nitride semiconductor, the authors should distinguish and clarify the claims in this paper from the previously reported works.

2. In a mixed ionic/covalent material like GaN, even though a 2D-layer of Mg is intercalated that separates the top and bottom GaN layers, the authors need to verify and comment on whether or not Mg is making bonds with N (above or below), forming Mg_xN_y complex and if that has any impact on the overall properties.

3. The authors report an enormous -10.1% c-axis uniaxial compressive elastic strain in GaN sandwiched between the 2D-Mg layers from HAADF-STEM images. The following questions need a detailed answers-

a. With such an enormous compressive strain of -10.1%, one would expect that the bandgap of GaN will change substantially. The reviewer could not find a discussion on this issue. The authors need to comment on it and also connect the bandgap evolution with the volume deformation potential of GaN.

b. It would be great to show high-resolution TEM images and electron diffraction patterns of strained GaN to clarify further that they maintain their crystal structure, lattice symmetry, etc.

c. HAADF-STEM images are used to calculate the amount of strain present in Mg/GaN film. Usually,

for such a process to calculate strain, one needs to account for distortions from the scanning of the beam (e.g., averaging many images to make up for sample drift during the scanning of one frame). Since the strain calculation critically depends on this aspect, the author should comment on and address this issue.

4. Regarding the electrical characterization of the intercalated films, the authors should address the following points-

a. The authors have compared their Hall mobility results with the theoretically calculated results by Ponce et al. The amount of strain in the theoretical report is relatively smaller than the -10.1 % strain reported in the experiment. Therefore, the authors need to address this point.

b. Usually, one of the main assumptions of Hall measurement is that the film under study is uniform and homogeneous. The presence of Mg/GaN (even though Mg is in monolayer form) in the upper part and unchanged GaN in the lower part of the film does alter the homogeneous and uniform film criteria. The author should address this issue.

c. Since the change in Hall mobility and carrier concentration at 300 K is appreciable but not large, the author should verify the effects of only the annealing process (without the Mg layer on top of the film) on the electronic transport properties of the film.

5. The authors have not reported on the impact of intercalation on the photoemission properties of GaN. How does the photoemission quantum yield change due to the presence of the intercalated Mg inside GaN.

6. There are some typos, reference numbering, and duplicate reference issues in the manuscript. These should be corrected.

A. Summary of the key results: The authors show the intercalation of 2D-magnesium inside GaN at ambient pressure, resulting in a large compressive stress inside GaN. Such large stress leads to high hole mobility due to the reversal of the sign of the crystal field splitting. The authors also state that the 2D-Mg/GaN structure can be considered as a spontaneously formed metal/semiconductor superlattice and could serve as a novel probe for such heterostructures transport measurements.

B. Originality and significance: if not novel, please include reference: High degree of originality and significance. One claim, "first-of-its-kind observation" needs some explanation and nuanced discussion, as mentioned in detail in the comments section.

C. Data & methodology: validity of approach, quality of data, quality of presentation: Mostly correct. But some of the approaches and data analysis (as highlighted in the reviewer's comment) require additional analysis and explanations.

D. Appropriate use of statistics and treatment of uncertainties: Does not apply in this manuscript significantly.

E. Conclusions: robustness, validity, reliability: Mostly well-supported. But some of the conclusions require clarifications as highlighted in the reviewer's comments.

F. Suggested improvements: experiments, data for possible revision: Highlighted in the reviewer's comments clearly.

G. References: appropriate credit to previous work? Require some corrections and a few citations of previous work as highlighted in the comments.

H. Clarity and context: lucidity of abstract/summary, appropriateness of abstract, introduction and conclusions: Mostly fine.

Author Rebuttals to Initial Comments:

Author's Response:

Dear Editors,

We are honoured and pleased to submit our revised manuscript.

Firstly, we apologized for the extended time taken to prepare the revised manuscript: we have added much more state-of-the-art TEM results that are time-consuming. The revised manuscript has undergone an overall modification compared to the initial submission, reflecting our improved understanding of these novel superlattices. It contains enriched content and demonstrates marked improvements in the quality of data and figures. We also addressed all the concerns and criticisms from the referees about our previous data and analysis. Our point-by-point responses are offered below.

During the preparation of this revised manuscript, the practice of annealing metal Mg on GaN has been adopted by different sub-teams in my research group to fabricate GaN-based power devices with record performance, leading to several rapid publications, which are cited in this manuscript and this response letter. It is worth mentioning that these publications focused on the device performance, whereas the superlattice structure formed by annealing Mg on GaN has not

been unveiled, as was sometimes vaguely referred to as the “MgGaN” compound or “GaN and Mg mixtures”.

Your kind and careful consideration is greatly appreciated.

5

Referees' Comments:

Referee #1 (Remarks to the AuthCor):

10

It is well known that p-type doping of gallium nitride is difficult: holes are in short supply, and have low mobilities. In spite of this, the discoveries (including by the group of an author of this manuscript) in early 1990s on doping with magnesium, and subsequent annealing was able to make the first p-type GaN, which enabled visible and blue LEDs, and blue lasers. Polarization-induced doping has recently been successful in enabling high-performance p-channel FETs by providing very high mobile hole densities, and deep-UV lasers by distributed polarization doping. But the problem of low hole mobilities remain.

15

20

The authors report a rather interesting experimental discovery in this manuscript, which corroborates a theoretical prediction of a clever way to increase the hole mobility in gallium nitride. What is rather remarkable is the manner in which this is achieved. At the outset, it appears to be an unexpected finding. They see that by depositing Mg ex situ on the surface of a Mg-doped GaN layer, and annealing it at optimal temperatures, the deposited Mg mixes with the GaN in an ordered way. It forms a multilayer structure, with nearly 2D Mg layers intercalated between GaN monolayers. They track the reason for this observation to the fact that the c- and a-lattice constants of GaN, a hexagonal wide-bandgap semiconductor, and Mg, a hexagonal metal, are nearly the same. The intercalation seems to squeeze the c-lattice constant and stretch out the a-lattice constant, causing a very large strain the GaN monolayers between the Mg 2D layers.

25

30

A recent theoretical prediction had proposed that if a suitable strain could be applied to GaN such that the ordering of the valence band could be reversed: i.e., if the split-off band could be raised to the top, a large increase of hole mobility would result. The authors perform polarization-sensitive cathodoluminescence measurements which shows a strong enhancement of transverse magnetic (TM) polarization after the Mg intercalation: the emission was TE-dominant before intercalation. The TM dominance indicates the desired valence band reordering is likely achieved. They then measure the hole mobility by Hall effect and find a large enhancement in the mobility, from ~2X at 300 K to ~3X at 150 K. They measure a hole mobility of ~250 cm²/V.s at 250 K. Though the hole densities are low at this temperature (~2e13/cm³), these are unambiguously the highest hole mobilities measured at this temperature. The record of hole mobility in GaN stands at ~280 cm²/V.s in a 2D hole gas (see APL 119, 162104 (2021), which the authors may need to cite for comparison), but that is achieved at ~10 K (as opposed to ~150 K here). So the evidence of enhancement of hole mobility seems to be solid.

35

40

45

The authors present the data comprehensively, with structural, optical, and electronic characterization, and theoretical modeling and analysis. The work is of high interest to the community. To refine the presentation, here are a few recommendations:

The authors present the data comprehensively, with structural, optical, and electronic characterization, and theoretical modelling and analysis. The work is of high interest to the community. To refine the presentation, here are a few recommendations:

5 **1. The first question a reader may have in their minds is: if instead of a p-doped epitaxial GaN layer, you started with a UID GaN layer, and deposited Mg ex situ, and repeated the above study, what do you get? Perhaps the authors have already performed this control experiment. If so, it would be helpful for the readers to be aware of the result - if not in the main text, at least in the supplementary materials.**

10 **Response:**

We greatly appreciate your positive evaluation and high praise for our work. Your suggestion is indeed very insightful. We have experimented with annealing metallic Mg on unintentionally doped (UID) GaN or n-type GaN, and have successfully obtained the same Mg-intercalated GaN superlattices. In the revised manuscript, most of the STEM observations and CL measurements were conducted on MiGs-incorporated UID-GaN samples. Regarding electrical performance, our initial attempts at annealing metal Mg film on UID-GaN and conducting Hall-effect measurements did not detect p-type conductivity. We later discovered that this issue stemmed from the discontinuous MiGs phases: while the MiGs phase in (0001) GaN functions as p-type, these phases are separately embedded within the n-type GaN matrix. We have included this finding in the main text, specifically in the section discussing the formation of the MiGs nanostructure through Mg annealing. This embedding process hinders the measurement of Hall-effect transport. However, in recent months, we have successfully fabricated GaN Camel diodes, essentially ultra-thin p-type GaN on n-type GaN, simply by annealing Mg on a UID-GaN sample [Ref.R1]. Fortunately, since the device relies only on the vertical transport of carriers in the MiGs, the discontinuous nature of the MiGs phases does not pose a problem. As a result, we included the *C-V* measurement results of the MiGs-incorporated UID-GaN, formed at different annealing temperatures, which leads to varying amounts of MiGs being incorporated. This confirms that the MiGs structure possesses p-type conductivity.

30 **Corresponding changes in the manuscript:**

Added Paragraph 1–2 of the Section of “Electrical properties of MiGs on n-type and p-type GaN” from Page 17, Line 12 to Page 18, Line 5.

35 [Ref.R1] Biplab Sarkar, Jia Wang, Oves Badami, Tanmoy Pramanik, Woong Kwon, Hirotaka Watanabe, and Hiroshi Amano. "Ga-polar GaN Camel diode enabled by a low-cost Mg-diffusion process." Applied Physics Express (2023).

40 **2. Please use different colors in Fig 2 (c), the text in the TEM is not clearly visible.**

Response:

Thank you for highlighting this issue. We understand the importance of clarity in our figures and will be more attentive to the legibility of the text in our images.

45 **3. Please state how the band structures in Fig. 3 (a) were calculated (e.g. k.p or pseudopotential or tight-binding, or DFT...). Also, it would be useful to quantify the strain by also including the pressure (e.g., MPa).**

Response:

Thank you for bringing this to our attention. To calculate the deformation potential and electronic band structure of GaN under various strains (changed to Fig. 4a and 4b), we employed

the popular hybrid functional (HSE) method. The specifics of this methodology have been detailed in the revised manuscript. Additionally, your suggestion to include the pressure value as quantitative representation of strain is inspiring. Considering the high elastic modulus of GaN (approximately 295 GPa), applying significant strain to such a rigid material is notably challenging. Expressing strain in terms of pressure provides a more impactful representation. We have estimated that the pressure exceeds 20 GPa at an elastic strain of -10%. This estimation has been incorporated into the revised manuscript for a more comprehensive understanding.

Corresponding changes in the manuscript:

Added Page 22, Line 4–Page 22, Line 12.

Added Page 11, Line 13–Page 11, Line 16

4. Some citations are out of order (e.g. in Page 11, line 6 reference 32 is not the correct one, and Page 13, line 30, reference 39 is missing). Please check and fix the reference citation numbers.

Response:

Thank you for pointing out these discrepancies. We apologize for any confusion caused by our oversight. The necessary corrections to the reference citation numbers have been made, ensuring accuracy in the revised manuscript.

5. Page 13, line 13: Since GaN is a uniaxial crystal, the crystal-field splitting of LH and HH should remove all degeneracies at the gamma point in the band structure, even though the LH/HH splitting is small - this should be the case even without strain. Please check and edit the sentence if required.

Response:

Thank you for bringing this detail to our attention. You are correct in noting that GaN, being a uniaxial crystal, experiences crystal-field splitting between light holes (LH) and heavy holes (HH) at the gamma point, which should theoretically eliminate degeneracies. The split-orbit coupling results in a small splitting energy between LH and HH, approximately 10 meV, which remains consistent under various strains and internal parameters [Ref. R2]. Considering the much larger crystal-field splitting energy (over 400 meV at a strain of -10%), this smaller splitting is relatively negligible, especially when viewing the band structure at a broader energy scale. In light of this, we will clarify in the main text that the LH/HH splitting is minor and falls outside our calculation range for simplicity.

Corresponding changes in the manuscript:

Added Page 13, Line 10–Line 11

[Ref. R2] Samuel Poncé, Debdeep Jena, and Feliciano Giustino. "Hole mobility of strained GaN from first principles." *Physical Review B* 100, no. 8 (2019): 085204.

6. Page 22, line 6: SIMS - the last word should be Spectrometry (not spectroscopy).

Response:

We apologize for the typo and our confusion of spectrometry with spectroscopy due to lack of knowledge about the nuance between the two terminologies. We appreciate the reviewer's keen eye in identifying this mistake.

Corresponding changes in the manuscript:

5 **7. A short discussion on the feasibility of achieving the Mg intercalation uniformly rather than in the pyramidal regions shown in the supplementary materials will be useful for the readers.**

Response:

10 Thank you for emphasizing this important aspect. Our extensive STEM observations over the past year have greatly deepened our understanding of the MiGs structures, enabling a more thorough discussion of this topic. This discussion has been included in Extended Data Fig. 1 and Section 1 of the Supplementary Information. We have observed that the upper part of the MiGs phase is typically continuous, whereas the lower part adopts a discontinuous, pyramidal shape. This morphology aims to minimize the volume of the inversion domain region, as a large bottommost Mg sheet would result in an undesirably large inversion domain beneath the Mg sheet. Therefore, the Mg intercalation tends to form in a pyramidal region, with the bottommost Mg sheet contracting to a diameter akin to that observed in a single Mg sheet-induced pyramidal inversion domain (PID) (about 5 nm in diameter) [Ref. R3], which is a frequent occurrence in over Mg-doped epitaxially grown GaN.

20 **Corresponding changes in the manuscript:**

Added Extended Data Fig. 1 and Section 1 of the Supplementary Information

25 [Ref. R3] Kenji Iwata, Tetsuo Narita, Masahiro Nagao, Kazuyoshi Tomita, Keita Kataoka, Tetsu Kachi, and Nobuyuki Ikarashi. "Atomic resolution structural analysis of magnesium segregation at a pyramidal inversion domain in a GaN epitaxial layer." Applied Physics Express 12, no. 3 (2019): 031004.

30 **8. In addition to the scattering mechanisms of holes mentioned (e.g. phonons), neutral impurity scattering should be considered in the discussion, since only a small fraction of the Mg acceptors are providing holes, and thus the rest are neutral and are in the path of the mobile holes, and cause scattering.**

Response:

35 Thank you for bringing up the notion of the scattering mechanism in our study. Over the past year, we have refined our understanding of the origins of increased hole concentration following Mg intercalation and the formation of the MiGs phase. We tend to assume that the increased hole concentration might stem from the gradient of the polarization field and unbalanced polarized charges [Ref. R4], resulting from the periodic inversion domains introduced by Mg sheets.

40 Furthermore, it has been widely reported that the formation of a single Mg-layer PID tends to deplete Mg in that region [Ref. R3 R5]. This depletion also occurs in p-GaN grown by MOCVD, where a high density of PIDs reduces the number of effective acceptors. Similarly, when Mg interstitially diffuses to form aligned sheets, it tends to deplete Mg in these regions as well, leaving fewer isolated substitutional Mg atoms in the MiGs area. Additionally, the presence of high hydrogen concentration passivates any substitutional Mg, further reducing its electrical activity.

45 Interestingly, terahertz time-domain ellipsometry studies have shown that the scattering time remains almost unchanged between blank p-GaN samples and MiGs-incorporated p-GaN samples, despite the latter having a significantly increased hole concentration. This observation supports our assumption that the increased charge and hole densities are primarily due to

interactions within the polarization field, rather than additional ionized impurities that would typically shorten the scattering time. We have included this analysis and discussion in the main text of the revised manuscript, which should provide a comprehensive perspective on this aspect of our study.

5

Corresponding changes in the manuscript:

Page 14, Line 16–Page 15, Line 15

10

[Ref. R3] Kenji Iwata, Tetsuo Narita, Masahiro Nagao, Kazuyoshi Tomita, Keita Kataoka, Tetsu Kachi, and Nobuyuki Ikarashi. "Atomic resolution structural analysis of magnesium segregation at a pyramidal inversion domain in a GaN epitaxial layer." *Applied Physics Express* 12, no. 3 (2019): 031004.

[Ref. R4] John Simon, John, Vladimir Protasenko, Chuanxin Lian, Huili Xing, and Debdeep Jena. "Polarization-induced hole doping in wide-band-gap uniaxial semiconductor heterostructures." *Science* 327, no. 5961 (2010): 60-64.

15

[Ref. R5] Northrup, John E. "Magnesium incorporation at (0001) inversion domain boundaries in GaN." *Applied physics letters* 82, no. 14 (2003): 2278-2280.

20

Referee #2 (Remarks to the Author):

In this manuscript, the authors report the intercalation of magnesium (Mg) into gallium nitride (GaN), an important wide-bandgap semiconductor that unleashes the era of blue-light LEDs, to produce the unique two-dimensional (2D) Mg-GaN superlattice through a simple metallurgical reaction at high temperatures. The results show that the intercalated sheets would generate high compressive elastic strains to the surrounding matrix, thus the so-called 2D Mg-GaN heterostructure can have a modified electronic band structure (wavelength), as well as a considerably increased (>200%) hole mobility (a long-sought goal for p-type GaN) due to the crystal-field splitting reversal in the hexagonal lattice (with small c/a value upon compressive stress). This is an interesting piece of research that build 2D metal into a covalent-ionic thin film semiconductor to achieve strain engineering effect, which could be potentially useful for future device applications (such as in optoelectronics and power electronics). But there are some critical issues about the fundamental mechanisms and experimental details that need to be clarified:

25

30

Response:

We are grateful for the referee's expectations, especially regarding the potential device applications of our research. We are glad to share that over the past year, we have published several papers utilizing the device applications of MiGs-incorporated GaN [Ref. R1, R6-9]. Notably, MiGs has been instrumental in achieving excellent Ohmic contact to p-type GaN with ultra-low contact resistance [Ref. R7] as well as the development of p-n junction diodes [Ref. R6], p-SBDs [Ref. R9], and GaN Camel diodes [Ref. R1] with record-breaking performance. Additionally, MiGs-incorporated n-GaN has been effectively utilized to demonstrate the efficiency of Camel diodes [Ref. 10]. However, the precise origins and the microstructure underlying these improved device performances were not fully identified or disclosed in these earlier publications. This current manuscript aims to fill that gap by detailing the true origin and identity of the microstructure leading to the enhanced performance.

35

40

45

We also deeply appreciate the comprehensive and constructive feedback from this referee. The comments have been instrumental in guiding our revisions, significantly enhancing the quality and clarity of our manuscript. We would like to extend our heartfelt thanks to the referee for their invaluable input and guidance.

[Ref.R1] Biplab Sarkar, Jia Wang, Oves Badami, Tanmoy Pramanik, Woong Kwon, Hirotaka Watanabe, and Hiroshi Amano. "Ga-polar GaN Camel diode enabled by a low-cost Mg-diffusion process." Applied Physics Express (2023).

[Ref. R6] Jia Wang, Shun Lu, Wentao Cai, Takeru Kumabe, Yuto Ando, Yaqiang Liao, Yoshio Honda, Ya-Hong Xie, and Hiroshi Amano. "Ohmic contact to p-type GaN enabled by post-growth diffusion of magnesium." IEEE Electron Device Letters 43, no. 1 (2021): 150-153.

[Ref. R7] Shun Lu, Manato Deki, Jia Wang, Kazuki Ohnishi, Yuto Ando, Takeru Kumabe, Hirotaka Watanabe, Shugo Nitta, Yoshio Honda, and Hiroshi Amano. "Ohmic contact on low-doping-density p-type GaN with nitrogen-annealed Mg." Applied Physics Letters 119, no. 24 (2021).

[Ref. R8] Yuta Itoh, Shun Lu, Hirotaka Watanabe, Manato Deki, Shugo Nitta, Yoshio Honda, Atsushi Tanaka, and Hiroshi Amano. "Substitutional diffusion of Mg into GaN from GaN/Mg mixture." Applied Physics Express 15, no. 11 (2022): 116505.

[Ref. R9] Shun Lu, Manato Deki, Takeru Kumabe, Jia Wang, Kazuki Ohnishi, Hirotaka Watanabe, Shugo Nitta, Yoshio Honda, and Hiroshi Amano. "Lateral p-type GaN Schottky barrier diode with annealed Mg ohmic contact layer demonstrating ideal current–voltage characteristic." Applied Physics Letters 122, no. 14 (2023).

First, the authors claimed the first-of-its-kind observation of intercalation of 2D-metal into bulk semiconductor. According to the references mentioned by the authors, however, Mg atoms can be implanted into GaN and even create pyramidal inversion domains, thus a more systematic and detailed explanation of the differences between the result in this work and earlier literatures is needed (for example, the 2D intercalated layers better introduce the reversal of crystal-field splitting...). Besides fundamental interests, the authors should demonstrate the benefits of the 2D-Mg intercalated GaN above the previous Mg doped GaN in applications and performances (for instance, the Ref. Appl. Phys. Express 12, 031004 (2019) has a Figure 2 that seems comparable to the results in this work...). On the other hand, about the detailed intercalated superlattice structure itself, the atomically resolved EDS (Fig. 1e-h) may not sufficiently prove the intercalated sheets are pure Mg, instead of Mg-Ga-N alloys or (Mg, Ga) nitride precipitates. Moreover, how did the authors obtain the atomic arrangement of the tip-edge of “Mg intercalant sheet” in Fig. 1d? The HAADF-STEM images may not resolve N columns due to their lower atomic number. It can be expected that the interface and the tip-edge might have different atom structures. Therefore, it would be ideal to resolve all Ga, Mg and N atoms by performing annular bright field or integrated differential phase contrast STEM imaging (Ultramicroscopy 160 (2016) 265–280), which can resolve all Ga, Mg and N atoms, for determining the atomic arrangement at the tip-edge of Mg intercalant sheet.

Response:

We are sincerely grateful for the referee's insightful comments and for introducing us to the integrated differential phase contrast (iDPC)-STEM imaging technique, which was a new and valuable addition to our research methodology. We have dedicated significant effort in recent months to effectively utilize iDPC-STEM, yielding exceptionally detailed images of the atomic arrangements, thanks to the skilled analysts from a third-party characterization company.

Particularly, we have highlighted the atomic arrangement at the tip-edge of the Mg intercalant sheet in Fig. 2. The iDPC imaging clearly shows that while the N atoms retain their original positions, serving as the “backbone”, the Ga atoms undergo a positional shift, resulting in the formation of N-polar and Ga-polar regions. This detailed imaging, combined with the STEM-EDS result, substantially reinforces our assertion in the revised manuscript that the intercalant sheets are composed of pure Mg, rather than Mg-Ga-N alloys.

5 Additionally, Figure 2 in Reference "Applied Physics Express 12, 031004 (2019)"—authored by key researchers who have now joined our team as co-authors—serves as a crucial reference. It demonstrates the detailed atomic configuration around single monolayer Mg segregation and the inversion domain region below it, a well-observed defect in epitaxial GaN films heavily doped with Mg. In stark contrast to previous findings, our observations reveal for the first time that these segregated interstitial Mg single sheets can align into an ordered superlattice structure. This turns a typical defect which decreases carrier concentration, into a constructive feature, as it contributes to increased carrier concentration and enhanced vertical hole transport perpendicular to the intercalant sheet.

10 In our revised manuscript, we offer a more systematic and detailed explanation of how our results—aligned Mg sheets forming superlattices in giant inverted triangle sizes—differ from earlier literature that discusses individual Mg sheets forming 5 nm inverted triangles [Ref. R3]. We hope that this discussion will be enlightening and engaging for the reviewer.

15 **Corresponding changes in the manuscript:**

Added the Section of “Polarity transition induced by 2D-Mg_i intercalation” and Fig. 2 in Page 6–8.

20 **Second, while it’s very intriguing to achieve this impressive superlattice structure through a simple metallurgical reaction, how the two-dimensional (2D) Mg layers were formed (the critical conditions and physical mechanisms for the formation of such 2D Mg structures in GaN, especially monolayer)? What is the area of intercalated Mg layers and how to control the sizes of these 2D Mg layers? Can the density/spacing of Mg intercalation layers between the multi-layer GaN be controlled? There seems a lack of essential experimental evidence and theoretical analysis here. To provide further insight into the related mechanism, numerical simulations are suggested, including the dynamic intercalation process of 2D-Mg into GaN layers using MD or Ab initio MD methods. Also, how about the mechanical and thermal stability of the 2D Mg-GaN heterostructures (when sustain a high elastic stress). Will there be further structural evolution or diffusion of the intercalated 2D Mg layers in GaN if there is an external field or after a long time? BTW, according to Fig. 1a, Mg layers were only found/intercalated on the surface of bulk GaN, as evidenced in Fig. S1. If and how can the depth of intercalation be controlled? Is there a method to construct an intercalated heterostructure that is more uniform? as these above critical issues may limit the practical device applications...**

35 **Response:**

40 We deeply appreciate the insightful questions raised by the reviewer, which highlight several critical aspects of our research that require further exploration. In the revised manuscript, specifically in the Section 1 of the Supplementary Information, we have elaborated on our enhanced understanding of the formation of MiGs nanostructures. We've discovered that the primary driving force behind this formation is the attraction between Mg sheets, akin to edge dislocations with opposite signs attracting each other due to the cancellation of strain fields. Similarly, Mg sheets are drawn together, partially neutralizing the inversion domain volume, which allows them to align and form large pyramidal phases. The structural stability is maintained as long as the bottommost Mg sheet remains small; otherwise, the inverted domain beneath it would become energetically unfavorable due to its size. This process enables the upper Mg sheets to extend their diameter, growing from single monolayers of about 5 nm to several

hundred nanometers. However, the full-continuous structure tends to exfoliate upon cooling and acid removal, resulting in mostly discontinuous MiGs pyramids within the GaN matrix.

5 Regarding the reviewer's suggestion of using ab initio MD methods for simulating the MiGs formation, this is indeed a significant ongoing project, detailed in the link below: [\[https://www.jst.go.jp/pr/info/info1614/pdf/info1614_en.pdf\]](https://www.jst.go.jp/pr/info/info1614/pdf/info1614_en.pdf). While this project aims to understand the stability and structural evolution of MiGs under various conditions, such as temperature and pressure, it is still in progress and will be elaborated upon in a more dedicated and focused publication in future. The current manuscript primarily focuses on the first
10 observation of MiGs through STEM techniques and comprehensive overview of its defining features and intriguing potentials, and we hope the reviewer appreciates our methodical approach and looks forward to our future findings.

15 In terms of controlling the depth and size of the intercalated structures, we observed that higher temperatures facilitate the presence of larger-size pyramidal phase which extends deeper into GaN matrix but are kinetically limited to a few hundred nanometers due to uneven Mg atom diffusion process largely influenced by nucleation density. We believe that by employing techniques such as ultra-slow cooling and mild cleaning, the upper continuous portion of the MiGs phase can be better preserved, potentially resulting in a more uniform structure.
20 This aspect of MiGs, even though discontinuous, has already proven valuable in enhancing vertical carrier transport in certain device applications. Our colleagues, many not listed as co-authors, are actively exploring the integration of MiGs in various device applications, indicating the practical significance of this structure. In comparison, this manuscript aims to reveal the microstructures formed after the annealing of metal Mg on GaN, shedding light on the
25 fundamental mechanisms behind this process and its potential technological implications.

Corresponding changes in the manuscript:

Added the Section 1 of the Supplementary Information in Page 34–36.

30 **Third, by measuring the lattice parameters using the HAADF-STEM images the authors stated that the intercalation led to high elastic compressive strain in multi-layered GaN without causing volume expansion. Here the claim of “no volume expansion” may need double check and verified... In addition, as demonstrated in Fig. 2, the 2D Mg layers are intercalated unevenly, thus the compressive strain produced may not be entirely uniform; and at the tip-edge, the sheet may also result in different strain states. The authors are suggested to perform lattice strain mapping by geometric phase analysis (Ultramicroscopy 74, 131–146 (1998)), and the strain values (up to "10%") should be also better verified. Furthermore, it's generally accepted that the modified optical and electronic properties of GaN are mainly attributed to high elastic compressive strains induced by “Mg intercalant monolayer”, however, the intercalation also results in the multi layered structure, whether these multiple Mg monolayers and their spacing/density may have an effect on the measured cathodoluminescence spectra and/or hole mobility? Also consider the effect of non-uniform compressive strain near tip edge on the measured physical properties... Part of the reason for the above questions is that, recent numerical and experimental studies on elastic strain engineering suggest that substantial band gap changes can be found when elastic strain is ultra large (such as close to 10%), while here the CL results in Fig. 3 show relatively modest change in bandgap/wavelength shift (compare to the significantly increased hole mobility due to smaller c/a) under the claimed "10%" compressive elastic**

strain here... A few theoretical/DFT calculation on the electronic bandgap of the intercalated/strained GaN structure may help to explain.

Response:

We sincerely appreciate the reviewer's insightful concerns. In response, our revised manuscript includes a more rigorous evaluation of the previously discussed methods. Notably, we have generalized this special case of the Mg-intercalated GaN superlattices into a new categorical concept of interstitial-type intercalation: the short-range interstitial Mg sheets always terminate within the GaN matrix, then the long-range GaN monolayers will remain continuous, both away and within the intercalation region without being disrupted by the intercalated Mg sheets, which intrinsically poses constraint on the volume expansion, or lattice swelling, of these GaN sheets.

Acknowledging that our initial approach to quantifying strain was rudimentary, we have now adopted a more refined method. Initially, we attempted geometric phase analysis, but the results have large noise due to the limited size of GaN monolayers scale within Mg sheets and the non-uniformity. Consequently, we shifted to atom-by-atom distance measurements for strain mapping, using aberration-corrected, drift-free HAADF-STEM images for precision. This approach, detailed further at [<https://www.eag.com/app-note/atom-by-atom-distance-measurements/>], compares the atomic distances in strained regions to a reference unstrained area near a void in the same image. Our findings reveal that the ratio of in-plane strain versus out-of-plane values align closely with the Poisson's ratio of GaN, confirming that the observed strain is indeed uniaxial.

Additionally, we conducted DFT analysis to quantify the bandgap shift in uniaxially strained GaN (over 0.13 eV) and compared this with STEM-CL characterizations. The latter indicated a significant blue-shift (around 0.08-0.10 eV) in the optical emission from regions containing the MiGs nanostructure, roughly proving the bandgap widening effects in the strained GaN. The discrepancy between these values might arise from the quantum-confined Stark effect due to the strong piezoelectric and spontaneous polarization fields in the MiGs nanostructures, leading to a red-shift in optical emissions. We also observed that cathodoluminescence from the MiGs regions is weak, most likely due to higher carrier concentration in the region, as CL intensity decreases with increased carrier density.

Regarding mobility, terahertz time-domain ellipsometry showed that the scattering time in MiGs-incorporated samples is comparable to blank p-GaN samples, suggesting that the interstitial sheets may not significantly scatter free carrier movement. Moreover, we found that the vertical transport of carriers perpendicular to the interstitial sheets is greatly enhanced. This enhancement leads to the enabling of Ohmic contacts with low contact resistance to lightly doped p-type GaN or plasma-damaged p-GaN and the development of p-n junction diodes, p-SBDs, and GaN Camel diodes with record-breaking performance, following the incorporation of the MiGs nanostructure.

Corresponding changes in the manuscript:

Major revised the Sections of “High uniaxial compressive strain in the MiGs nanostructures”, “Changes in the valence band structure of strained GaN”, and “Electrical properties of MiGs on n-type and p-type GaN” from Page 9 to Page 19.

A few minor points for authors' consideration and correction:

1. What does the abbreviation ML stand for? The reviewer cannot find the definition of ML throughout the manuscript.

Response:

We thank the reviewer for picking out this error. The abbreviation ML stands for monolayer. We apologized for our carelessness.

The reviewer points out that the abbreviation "ML" was not defined in the manuscript. The authors acknowledge this oversight and clarify that ML stands for "monolayer."

Corresponding changes in the manuscript:

Page 2, Line 17.

2. The samples for STEM characterizations were processed at temperatures of 500, 550, 700, and 850 °C. However, the samples for SIMS experiments were processed at temperatures of 500, 600, 700, and 800 °C. How to determine and why not use the same temperatures?

Response:

We apologize for this oversight. When designing the experiments, we did not intend to directly compare the two sets of results. On this note, it should also be acknowledged that, based on our experimental data, distinguishing the differences between samples annealed at 550 °C and 600 °C for a short duration of 10 minutes is challenging across all aspects of characterization. This difficulty arises because the variations within individual samples post-annealing may even exceed the systematic differences observed between these two annealing temperatures.

3. "Fig. S5" in the supplementary information should be "Fig. S1". And what are high-density nano-sized white regions in Fig. S1d?

Response:

In the revised manuscript, Fig. S1d was moved to Fig.5e (first image). The STEM analyst informed us that the bright spots on the image are due to Ga redeposition from the sample preparation (residue of FIB prep).

Corresponding changes in the manuscript:

Page 17, Line 6–8

4. Page 9 Line 15 "A relatively low acceleration voltage (3 kV) electron beam was utilized to collect signals from the shallow region (~50 nm) near the surface so that more optical emissions are expected from the Mg-intercalated GaN layers" How to ensure the optical emissions come only from the shallow region (the thickness of the active layer)?

Response:

We are sorry for this notion; indeed, the optical emissions were not only from the MiGs but also from the GaN matrix because the latter has much higher CL intensity. In the revised manuscript, we used STEM-CL with sampling locations around 100 nm, to ensure more signals are coming from the region of our interest.

Corresponding changes in the manuscript:

Revised the CL evaluation method and discussion from Page 13, Line 19 to Page 14, Line 15.

5. The elastic modulus of GaN and Mg in Table 1 appears wrong. The two values should be switched (not sure if the table is really needed in Main text paper)

Response:

We are sorry for this mistake, the reviewer's effort in picking out this error is greatly appreciated.

Corresponding changes in the manuscript:

Page 9, Table II

6. Is it possible to extend the 2D metal intercalation to other semiconductor systems, such as those in diamond cubic lattice? or hexagonal structure is more preferred?

Response:

We thank the reviewer for this interesting question. We think the reason for Mg to be able to intercalate into GaN, mostly because Mg and GaN share the same lattice type and more importantly, they have almost the same lattice constants—a miracle event in nature. These fundamental properties explain the strong tendency of Mg to segregate on (0001) GaN. The tendency to shrink the volume inversion domain caused by Mg segregation layers further serve as the driving force to attract and align Mg sheets into ordered shape. These two factors are purely coincidental by nature, making any artificial attempt to mimic another similar example difficult to achieve.

7. Ref. 15 and Ref. 31 are repeated.

Response:

We are sorry for this mistake, once again, the reviewer's effort in the identification of this error is appreciated.

Referee #3 (Remarks to the Author):

The manuscript entitled "Observation of 2D-magnesium-intercalated gallium nitride superlattices" by Wang et al. discusses the intercalation of 2D-magnesium inside GaN at ambient pressure, resulting in a large compressive stress inside GaN. Such large stress leads to high hole mobility due to the reversal of the sign of the crystal field splitting. The authors also state that the 2D-Mg/GaN structure can be considered as a spontaneously formed metal/semiconductor superlattice and could serve as a novel probe for such heterostructures transport measurements.

The demonstration of metallic (Mg) intercalation inside a mixed ionic/covalent material like GaN is very interesting. However, I am unsure if one can call it a "first-of-its-kind observation" for a ceramic material (please see below). In addition, the increase in hole mobility due to the reversal of the sign of the crystal field splitting has been long desired and will be greatly appreciated.

Overall, I find that the manuscript will be interesting for (a) studies on the intercalation of metals inside ionic/covalent semiconductors, (b) increase in the hole mobility of GaN due to compressive strain and reversal of the sign of crystal field splitting, and (c) as a system for transport studies on metal/semiconductor superlattices. However, the manuscript also raises a significant number of questions that require detailed answers. The authors should carefully

address the following comments/questions before the manuscript is considered for publication in Nature.

5 **1. Intercalation of noble metals (such as Au, Ir) has been used previously inside non van der Waals ceramic materials. For example, Ti_3AuC_2 , Ti_3IrC_2 , etc. are synthesized from Ti_3SiC_2 (Nature Materials, 16, 814 (2017)) or the silicide formation (that the authors also mention). Though the present study uses metal intercalation inside an III-nitride semiconductor, the authors should distinguish and clarify the claims in this paper from the previously reported works.**

10 **Response:**

We are grateful for your insightful comment that encourages a deeper reflection on the uniqueness of our work in the context of metal intercalation in non-van der Waals ceramic materials. Your observation rightly points out the necessity to distinguish our findings from previous studies, such as the intercalation observed in materials like Ti_3AuC_2 and Ti_3IrC_2 , or MAX phase in a broader category.

15
20 In response, we have introduced the concept of “interstitial intercalation” in our revised manuscript, which we believe distinctly characterizes the phenomenon observed in our study. This concept stands in contrast to “substitutional intercalation”, typically represented by MAX phases and Guinier-Preston zones. The $M_{n+1}AX_n$ phase has strict stoichiometric formula, the A layer always extends through the entire layer, and the intercalation of A, does not induce any strain, making this MAX phase intrinsically a material, rather than a structure. The Guinier-Preston zones demonstrate substitutional intercalation where the intercalation sheets terminate within the matrix, disrupting the localized structure and leading to defects. In the case of the Guinier-Preston zone, the intercalation sheets are embedded into the matrix, meaning they always terminate inside the matrix, the absence of stoichiometry makes them belong to the category of a structure, rather than material. Furthermore, since they are substitutionally occupying the sites originally belonging to the matrix atoms, hence the localized structure of the original matrix, however, is disrupted, causing defects.

25
30
35 In our case, the intercalation of Mg sheets in GaN, classified as interstitial intercalation, exhibits unique features. The Mg sheets always terminate within the matrix (the feature of a structure), different than a substitution intercalation, the continuity of the GaN monolayers both outside and inside the Mg interstitial intercalation region are maintained without being disrupted. This continuity, combined with the intrinsic generation of high uniaxial compressive strain, underscores the distinct nature of our observed intercalation phenomenon. Such intercalation not only retains the integrity of the host lattice but also introduces significant mechanical stress, altering the material's electronic properties.

40 To provide a comprehensive understanding, our revised manuscript now includes a section that methodically contrasts substitutional and interstitial intercalation. This comparison aims to clarify the unique aspects of our study and emphasizes the specific implications of Mg intercalation in GaN, setting it apart from the previously reported metal intercalation in other ceramic materials.

45 **Corresponding changes in the manuscript:**

Added Paragraph 1–2 of the Section 3 of “High uniaxial compressive strain in the MiGs nanostructures” in Page 9–10.

2. In a mixed ionic/covalent material like GaN, even though a 2D-layer of Mg is intercalated that separates the top and bottom GaN layers, the authors need to verify and comment on whether or not Mg is making bonds with N (above or below), forming Mg_xN_y complex and if that has any impact on the overall properties.

Response:

We appreciate your query regarding the bonding nature of Mg within the GaN matrix and its subsequent impact on the overall properties of the material. Our experimental observations are high consistent with the model built by John Northrup [Ref. R5]. We concur with the model's implication that the Mg atoms likely form ionic bonds with their six nearest N atoms. This bonding is influenced by the shape of the inversion domain region created by the Mg monolayer. These Mg layers act as finite sheets of positively charged ions, exerting a triangular-like electric potential field within the GaN structure. It is crucial to note that when we describe these layers as 2D, we refer to their single atomic layer thickness, rather than implying a van der Waals attraction similar to materials like graphene. This distinction is important to avoid misconceptions about the nature of 2D-Mg in our study.

Regarding the overall impact of these Mg sheets, our research indicates that they induce periodic inversion domains in the intercalated GaN monolayers. This results in a high density of net positive charges and an increase in negative free carriers. Additionally, it is well-established that individual Mg interstitials can serve as deep level defects or compensating donors [Ref. R10]. Hence, it is plausible that the segregated Mg monolayers in our study could also introduce density of states across a wide range of energy range into the forbidden band, which is supported by our EELS-STEM findings which indicate inter-band transitions within the MiGs. This phenomenon likely contributes to the observed yellow luminescence in the material.

However, despite these potential trap states, we have observed a marked enhancement in the vertical transport of holes perpendicular to the Mg sheets. This is evidenced by a significant improvement in Ohmic contact behavior in p-type GaN following the formation of MiGs. We have been able to achieve ultra-low contact resistance in lightly doped or plasma-damaged p-type GaN, scenarios where such low resistance was previously unattainable. This improvement underscores the functional impact of the Mg intercalation, suggesting that even though the Mg sheets may introduce certain trap states, they do not block the vertical transport of holes, on the tract, the enhanced vertical carrier transport properties that could have significant implications for the application of p-type GaN in electronic devices.

[Ref.R5] Northrup, John E. "Magnesium incorporation at (0001) inversion domain boundaries in GaN." Applied physics letters 82, no. 14 (2003): 2278-2280.

[Ref. R10] Giacomo Miceli, and Alfredo Pasquarello. "Self-compensation due to point defects in Mg-doped GaN." Physical Review B 93, no. 16 (2016): 165207.

3. The authors report an enormous -10.1% c-axis uniaxial compressive elastic strain in GaN sandwiched between the 2D-Mg layers from HAADF-STEM images. The following questions need detailed answers:

a. With such an enormous compressive strain of -10.1%, one would expect that the bandgap of GaN will change substantially. The reviewer could not find a discussion on this

issue. The authors need to comment on it and also connect the bandgap evolution with the volume deformation potential of GaN.

Response:

We are thankful for your observation regarding the substantial change in the bandgap of GaN due to the reported -10.1% c-axis uniaxial compressive elastic strain. Indeed, such a significant level of strain would be expected to have a notable impact on the bandgap.

To address this, we have employed density functional theory (DFT) using the Heyd-Scuseria-Ernzerhof (HSE) hybrid functional method to calculate the deformation potential and electronic band structure of GaN under this considerable uniaxial compressive strain. Our calculations reveal a non-linear relationship between the deformation potential and the change in lattice constant. This relationship aligns with some previous theoretical studies, indicating an initially rapid increase in the bandgap under small strain, followed by a more gradual change at higher strain levels. This behaviour is partly attributed to the non-linear adjustments in the internal parameter of the crystal structure under varying strain conditions.

Our theoretical analysis suggests that the bandgap of GaN increases by approximately 0.13 eV compared to its unstrained state. Experimentally, we observed a blue-shift of approximately 10 nm (corresponding to 0.09-0.10 eV) between the location far away from the MiGs structure and one containing the MiGs phase the in our scanning transmission electron microscopy-cathodoluminescence (STEM-CL) measurements. The slight discrepancy between the theoretical and experimental values could be attributed to the relatively weaker CL intensity from the MiGs structure, which prevents an accurate measurement. Additionally, the quantum-confined Stark effect, which typically causes a redshift in near-band-edge (NBE) photoemission, might also contribute to this variance.

In summary, both our theoretical and experimental findings consistently indicate a significant shift in the bandgap of GaN under the influence of the observed compressive strain, highlighting the profound impact of 2D Mg intercalation on the electronic properties of GaN.

Corresponding changes in the manuscript:

Added Fig. 4a and Fig. 4c,d and Paragraph 1–2 of the Section of “Changes in the valence band structure of strained GaN” in Page 12–14.

b. It would be great to show high-resolution TEM images and electron diffraction patterns of strained GaN to clarify further that they maintain their crystal structure, lattice symmetry, etc.

Response:

We acknowledge the importance of visually demonstrating the integrity of crystal structure and lattice symmetry in the intercalated GaN. To this end, in our revised manuscript, we have included high-resolution images obtained from integrated differential phase contrast (iDPC) scanning transmission electron microscopy (STEM). These images effectively resolve all Mg, N, and Ga atoms, providing clear insights into the crystal structure and lattice symmetry. Due to the limited number of GaN monolayers within the Mg intercalant sheet and the gradual transition from Ga-polar to N-polar GaN, we observed that the electron diffraction patterns, derived from Fourier transformation, were too indistinct to yield meaningful information. Hence, the iDPC-STEM imaging method was deemed more appropriate and revealing for this aspect of our study.

Corresponding changes in the manuscript:

Added Fig. 1d and Fig. 2b, 2d, 2e and Page 5, Line 24–27.

5 **c. HAADF-STEM images are used to calculate the amount of strain present in Mg/GaN film. Usually, for such a process to calculate strain, one needs to account for distortions from the scanning of the beam (e.g., averaging many images to make up for sample drift during the scanning of one frame). Since the strain calculation critically depends on this aspect, the author should comment on and address this issue.**

Response:

10 We appreciate your observation on the criticality of accounting for beam scanning distortions in strain calculations from HAADF-STEM images. In response to this concern, we have employed a more refined method for strain mapping in the revised manuscript. We utilized atom-by-atom distance measurement technology, provided by a third-party testing service. This approach allows for a highly accurate strain mapping, contrasting the strained regions with a nearby unstrained reference region within the same image. This enhanced method significantly improves the accuracy and quality of our strain evaluations compared to the initial methodology used.

Corresponding changes in the manuscript:

20 Added Fig. 3b, 3c, 3d and Extended Data Fig. 2.

4. Regarding the electrical characterization of the intercalated films, the authors should address the following points-

25 **a. The authors have compared their Hall mobility results with the theoretically calculated results by Ponce et al. The amount of strain in the theoretical report is relatively smaller than the -10.1 % strain reported in the experiment. Therefore, the authors need to address this point.**

Response:

30 We appreciate the reviewer's observation. The hole mobility measured in our study, whether by the Hall-effect van der Pauw method or terahertz time-domain ellipsometry, represents a weighted average value encompassing both p-type MiGs and the p-type GaN matrix. Consequently, the mobility of the p-type GaN matrix restricts the overall hole mobility from reaching the expected value. Furthermore, in the research conducted by Ponce et al., the theoretically calculated hole mobility, derived using state-of-the-art ab initio Boltzmann transport equations under uniaxial compressive strain, represents an upper bound. This calculation primarily considers phonon scattering, including acoustic phonon scattering with piezo-acoustic scattering (80%), longitudinal-optical phonon scattering (15%), and transverse-optical phonon scattering (5%). However, it notably omits other forms of scattering, such as impurity scattering. These factors contribute to the observed discrepancy between our
40 experimental results and their theoretical values.

In our upcoming work, we are collaborating with the primary authors from the paper by Ponce et al. to calculate the theoretical mobility of GaN under ultra-high strain conditions (up to -13%). This more dedicated and focused investigation into hole mobility and effective mass will be elaborately discussed in our future publication. We thank you for your understanding.

45 **b. Usually, one of the main assumptions of Hall measurement is that the film under study is uniform and homogeneous. The presence of Mg/GaN (even though Mg is in monolayer**

form) in the upper part and unchanged GaN in the lower part of the film does alter the homogeneous and uniform film criteria. The author should address this issue.

Response:

We thank the reviewer for bringing attention to this important aspect. It is true that the mobility increase is predominantly along the uniaxial compressive direction, which is perpendicular to the thin film. This orientation renders it unmeasurable by traditional Hall-effect and Van der Pauw methods. Over the past year, our subsequent research has uncovered significant inconsistencies in our initial Hall-effect measurement results. Specifically, we found that the results from that particular sample were not reproducible in other samples under similar conditions. These other samples exhibited only a slight increase in hole mobility. This modest increase did not align with the expectations set by the crystal field splitting phenomenon. We attribute this minor increase in hole mobility to a reduction in effective substitutional Mg_{Ga} acceptors as ionized impurities. This reduction, caused by the formation of ordered Mg sheets, marginally enhances mobility. A detailed analysis revealed that the reduced effective mass primarily affects the split-off holes in the k_z direction (Γ to A), significantly enhancing vertical hole transport. However, the slight increase in mobility in the k_x direction (Γ to K), as measured by the Hall-effect method, was not substantial enough for our purposes.

Therefore, we have decided not to rely on Hall-effect mobility measurements in our study. Instead, we have incorporated terahertz time-domain ellipsometry. This technique measures conductivity, capturing both in-plane (k_x) and out-of-plane (k_z) components. It has yielded intriguing results. We observed a notable improvement in transport properties before and after the incorporation of Mg into the p-GaN epilayer, indicative of increased mobility due to a decreased effective mass.

While we acknowledge that terahertz time-domain ellipsometry is a cutting-edge technique and may have inherent limitations, we have employed rigorous calibration methods and cross-verification with established techniques to ensure the reliability of our findings. Therefore, we believe that our results, while perhaps not as quantitatively precise as those obtained through more traditional methods, provide valuable qualitative insights and a strong basis for further investigation.

Corresponding changes in the manuscript:

Revised the Section of “Changes in the valence band structure of strained GaN” and added Page 14, Line 16–Page 15, Line 15.

c. Since the change in Hall mobility and carrier concentration at 300 K is appreciable but not large, the author should verify the effects of only the annealing process (without the Mg layer on top of the film) on the electronic transport properties of the film.

Response:

We thank the reviewer for emphasizing this important point. Indeed, prior to the annealing of metal Mg on GaN, we conducted an annealing-only process (at 750 °C for 10 minutes) on the as-grown p-GaN epilayer. This step was necessary to activate the Mg, which was initially passivated by hydrogen. We used the as-annealed p-GaN as a blank sample for comparison with the MiGs-incorporated p-GaN. This annealing method, developed by Nakamura et al. [Ref. R11], has been adopted as a standard process in all p-GaN-related procedures today. We apologize for not mentioning this crucial detail in the experimental section of our paper.

[Ref. R11] Tetsuo Narita, Hikaru Yoshida, Kazuyoshi Tomita, Keita Kataoka, Hideki Sakurai, Masahiro Horita, Michal Bockowski et al. "Progress on and challenges of p-type formation for GaN power devices." Journal of Applied Physics 128, no. 9 (2020).

5 **Corresponding changes in the manuscript:**

Added Page 21, Line 8–10.

10 **5. The authors have not reported on the impact of intercalation on the photoemission properties of GaN. How does the photoemission quantum yield change due to the presence of the intercalated Mg inside GaN.**

Response:

15 We appreciate your inquiry regarding the photoemission properties of GaN post-Mg intercalation. In response, we have incorporated state-of-the-art scanning transmission electron microscopy-cathodoluminescence (STEM-CL) data into our revised manuscript. This data reveals that photoemission from regions containing the MiGs phase is considerably weaker than in unaltered areas. Our recent studies have enhanced our understanding of the MiGs phase, showing its intrinsic p-type behaviour and high hole concentration. These characteristics, coupled with the strong polarization field from spontaneous and piezoelectric polarization, substantially reduce the cathodoluminescence intensity in these regions. This effect is attributed to the inverse relationship between CL intensity and carrier concentration, as well as the quantum-confined Stark effect, which diminishes the quantum yield of radiative recombination processes. We believe that these findings provide a comprehensive explanation for the observed changes in photoemission properties due to Mg intercalation and hope that this additional data addresses your concerns effectively.

25 **Corresponding changes in the manuscript:**

Added Page 13, Line 19–Page 14, Line 15.

30 6. There are some typos, reference numbering, and duplicate reference issues in the manuscript. These should be corrected.

Response:

35 Thank you for pointing out the typos, incorrect reference numbering, and duplicate reference issues in our manuscript. We acknowledge that these concerns have been raised by multiple reviewers and have taken meticulous care to address them in our revised manuscript.

40 A. Summary of the key results: The authors show the intercalation of 2D-magnesium inside GaN at ambient pressure, resulting in a large compressive stress inside GaN. Such large stress leads to high hole mobility due to the reversal of the sign of the crystal field splitting. The authors also state that the 2D-Mg/GaN structure can be considered as a spontaneously formed metal/semiconductor superlattice and could serve as a novel probe for such heterostructures transport measurements.

45 B. Originality and significance: if not novel, please include reference: High degree of originality and significance. One claim, “first-of-its-kind observation” needs some explanation and nuanced discussion, as mentioned in detail in the comments section.

C. Data & methodology: validity of approach, quality of data, quality of presentation: Mostly correct. But some of the approaches and data analysis (as highlighted in the reviewer's comment) require additional analysis and explanations.

5 D. Appropriate use of statistics and treatment of uncertainties: Does not apply in this manuscript significantly.

E. Conclusions: robustness, validity, reliability: Mostly well-supported. But some of the conclusions require clarifications as highlighted in the reviewer's comments.

10

F. Suggested improvements: experiments, data for possible revision: Highlighted in the reviewer's comments clearly.

15

G. References: appropriate credit to previous work? Require some corrections and a few citations of previous work as highlighted in the comments.

H. Clarity and context: lucidity of abstract/summary, appropriateness of abstract, introduction and conclusions: Mostly fine.

20

Response: Thank you once again for your meticulous review and valuable contributions to improving our work.

References:

25

Ref. R1. Biplab Sarkar, Jia Wang, Oves Badami, Tanmoy Pramanik, Woong Kwon, Hirotaka Watanabe, and Hiroshi Amano. "Ga-polar GaN Camel diode enabled by a low-cost Mg-diffusion process." *Applied Physics Express* (2023).

30

Ref. R2. Samuel Poncé, Debdeep Jena, and Feliciano Giustino. "Hole mobility of strained GaN from first principles." *Physical Review B* 100, no. 8 (2019): 085204.

Ref. R3. Kenji Iwata, Tetsuo Narita, Masahiro Nagao, Kazuyoshi Tomita, Keita Kataoka, Tetsu Kachi, and Nobuyuki Ikarashi. "Atomic resolution structural analysis of magnesium segregation at a pyramidal inversion domain in a GaN epitaxial layer." *Applied Physics Express* 12, no. 3 (2019): 031004.

35

Ref. R4. John Simon, John, Vladimir Protasenko, Chuanxin Lian, Huili Xing, and Debdeep Jena. "Polarization-induced hole doping in wide-band-gap uniaxial semiconductor heterostructures." *Science* 327, no. 5961 (2010): 60-64.

Ref. R5. Northrup, John E. "Magnesium incorporation at (0001) inversion domain boundaries in GaN." *Applied physics letters* 82, no. 14 (2003): 2278-2280.

40

Ref. R6. Jia Wang, Shun Lu, Wentao Cai, Takeru Kumabe, Yuto Ando, Yaqiang Liao, Yoshio Honda, Ya-Hong Xie, and Hiroshi Amano. "Ohmic contact to p-type GaN enabled by post-growth diffusion of magnesium." *IEEE Electron Device Letters* 43, no. 1 (2021): 150-153.

Ref. R7. Shun Lu, Manato Deki, Jia Wang, Kazuki Ohnishi, Yuto Ando, Takeru Kumabe, Hirotaka Watanabe, Shugo Nitta, Yoshio Honda, and Hiroshi Amano. "Ohmic contact on low-doping-density p-type GaN with nitrogen-annealed Mg." *Applied Physics Letters* 119, no. 24 (2021).

45

Ref. R8. Yuta Itoh, Shun Lu, Hirotaka Watanabe, Manato Deki, Shugo Nitta, Yoshio Honda, Atsushi Tanaka, and Hiroshi Amano. "Substitutional diffusion of Mg into GaN from GaN/Mg mixture." *Applied Physics Express* 15, no. 11 (2022): 116505.

50

Ref. R9. Shun Lu, Manato Deki, Takeru Kumabe, Jia Wang, Kazuki Ohnishi, Hirotaka Watanabe, Shugo Nitta, Yoshio Honda, and Hiroshi Amano. "Lateral p-type GaN Schottky barrier diode with annealed Mg ohmic contact layer demonstrating ideal current-voltage characteristic." *Applied Physics Letters* 122, no. 14 (2023).

- Ref. R10.** Giacomo Miceli, and Alfredo Pasquarello. "Self-compensation due to point defects in Mg-doped GaN." *Physical Review B* 93, no. 16 (2016): 165207.
- Ref. R11.** Tetsuo Narita, Kazuyoshi Tomida, Keita Kataoka, Hideki Sakurai, Masahiro Horita, Michal Bockowski, Nobuyuki Ikarashi, Jun Suda, Tetsu Kachi, and Yutaka Tokuda, "Progress on and challenges of p-type formation for GaN power devices." *Journal of Applied Physics* 128, no. 9 (2020).

Reviewer Reports on the First Revision:

Referees' comments:

Referee #2 (Remarks to the Author):

The authors have done a terrific job in addressing many of my previous review comments, including revealing the detailed atomic structure of Mg-intercalated superlattice by the iDPC-STEM, and quantifying the interatomic strains by using atomically resolved HADDF-STEM imaging, validating the ultralarge uniaxial compressive strain over 10%. The manuscript has been substantially improved with additional experiments, analyses, and discussions, thus would be appealing to a broad community concerning about wide-bandgap semiconductors, 2D materials and strain engineering. But I have a few additional questions and comments for the authors to consider, before publishing this research:

1. The reviewer greatly appreciates the authors on the atomic structure analysis and strain analysis, but it would be better to define and illustrate the “in-plane” and “out-of-plane” strain, respectively, and also check the Poisson's ratio of GaN accordingly. In addition, in supplementary material Fig S1, the authors show that there is obvious distortion in GaN lattice between adjacent intercalants. Can the authors also apply strain field analysis here, to explain the physical mechanism of this lattice distortion and the possible effect on the stability of intercalated structures. It has been previously reported in literature that in 2D materials interfaces (e.g. twisted bilayer graphene), structural distortion near the intercalated structure can lead to long-range interactions. Can the author briefly discuss whether there are long-range interactions between intercalated structures in the Mg-GaN structure?

2. About the CL characterization, STEM-CL was used for Mg-deposited n-type GaN while conventional SEM-CL was used for p-type GaN. Would there be any differences in terms of comparing the CL results for n-type and p-type GaN? Will the difference in methods cause variations in spectral features? For instance, in n-type GaN, the intensity of yellow luminescence is weaker in the MiG region comparing with in the blank GaN substrate, but contrarily it intensifies in the MiG region of p-type GaN. Any explanations? Also about “the CL intensity is markedly weaker in regions containing MiGs structure, associated with an increased carrier density and a strong electric field”, the decrease of CL intensity may result from many other reasons, such as Mg doping or uneven distribution of MiGs structure or impact of polarization, can we clarify the main mechanism here?

3. The method used in this study is annealing at 550°C for 10 mins. Now the authors have discussed the effect of different temperatures on the structure of MiGs, but not yet discussed the effect of time. Like what will happen to the MiGs structure if the annealing time is longer than 10 minutes? What would be the thickness of MiGs structure after annealing for more than 10 mins?

4. The author proposed that there might be charge transfer in Mg intercalated GaN, which seems that there is a chemical reaction here.. What is the difference between this and the classical alkali metal intercalation reaction? Can you predict the possible highest density of Mg in GaN, and how about optimum density?

Two minor issues to be addressed:

1. On page 14, line 1. The CL spectra of p-type GaN was in Extended Data Fig. 3 instead of Fig. 2.
2. The Zoomed-in STEM in Fig 4c can be more clear and sharp to review the MiGs structure, and there is not figure caption for Fig 4d (or there should be no "d" from your current caption)

Referee #3 (Remarks to the Author):

In the revised manuscript entitled "Observation of 2D-magnesium-intercalated gallium nitride superlattices" Wang et al. have performed substantial revision of their previous work. They performed several new experiments, reinterpreted many of the previous results, and addressed all of the reviewers' comments. The volume of work that has gone into revising the manuscript justifies the long delay in resubmitting the manuscript.

As I had highlighted in my comments, the previous version of the manuscript had presented three main claims/novelties (a) achieving intercalation of metals (Mg) inside an ionic/covalent semiconductor, (b) increase in the hole mobility of GaN due to the compressive strain and reversal of the sign of crystal field splitting, and (c) demonstration of Mg/GaN as a system for transport studies on metal/semiconductor superlattices.

Following this substantial revision of the manuscript, two of these claims have gone through a major revision and a new important claim is included.

1. Following the reviewer's comment, the authors have moderated their previous claim of "first-of-its-kind observation of intercalation of 2D metal into bulk semiconductor" to a more nuanced "interstitial intercalation", which is a more appropriate description in my opinion. Although the author's previous strong claim is moderated a bit in this revised manuscript, the observation of the intercalation of interstitial Mg monolayers inside GaN is still highly interesting and stands as a major novelty of this study. The new microscopy images are also very interesting and represent the work aptly.

2. The authors' 2nd claim about the increase in the hole mobility of GaN due to compressive strain has gone through a substantial change. Firstly, as the authors have mentioned, their previous mobility measurements with the Hall method did not capture the out-of-plane mobility, which the theory has predicted to increase. At the same time, the mobility measurements had some consistency issues, which the authors mentioned in the response letter.

To measure the out-of-plane mobility, the authors have now used terahertz time-domain ellipsometry, which provides indirect evidence of the out-of-plane mobility enhancement due to the intercalation of Mg in GaN. However, though terahertz time-domain ellipsometry gives very good qualitative information, a clear and unambiguous demonstration of out-of-plane mobility increase is not demonstrated (as the authors also mention in multiple places). The present demonstrations serve as a very strong indication but not as a certain proof.

The authors have also revised their earlier demonstration of strain-induced band reversal with a new STEM-CL analysis, which is a bit puzzling. In my opinion, the previous polarised CL spectroscopy was clear and convincing. However, the new STEM-CL measurement is very confusing and difficult to follow. I am unsure if these new STEM-CL measurements can be seen as an unambiguous demonstration of the strain-induced band reversal. I am also not sure why the authors would remove the previous CL measurements.

Therefore, the 2nd claim is no longer a clear and definite physical picture.

3. The third claim remains “as is” since it is more of a proposal for the future, which is fine.

The revised manuscript has also included new important claims, which should now be considered due to their impacts-

4. The authors now present a clear observation of the polarity transition in GaN due to the 2D-Mg intercalation in the interstitial sites. This observation provides direct evidence for the pyramidal inversion domains generally observed in Mg-doped GaN. This is an important novelty, in my opinion.

5. Lastly, although the authors present claims on improved Ohmic contact to p-type GaN and Schottky barrier height modulation for n-type GaN, I think it is better not to consider these as completely novel since the authors already published works on similar lines. The interpretations are now, of course, clearly evidenced in this paper.

Therefore, keeping in mind the novelty, technical quality, and broad appeal of the manuscript, I think that it would be most appropriate to focus on the interstitial intercalation, polarity transition, and potential for metal/semiconductor superlattice as the main message. Given the important novelties of these highlighted points, I support the publication of such as revised manuscript in Nature.

Focusing on these points would make the manuscript short and concise, which will be much more impactful, in my opinion. It will also significantly advance not only the research and industrial application of GaN but also other semiconductors and heterostructures in general. The rest of the data, analysis and interpretations can be included in the Supplemental Materials for reference and as a motivation for future research.

In addition to this main point, I would also strongly urge the authors to address the following points in the revised manuscript, which will strengthen the paper-

1. Mg is extremely prone to oxidation when exposed to air. Therefore, the authors must outline how they protected the Mg from oxidation between depositing on GaN and annealing at high temperatures under atmospheric pressure.

Similarly, generally, when Mg temperature exceeds more than 350-400°C, it starts to vaporize heavily due to its high vapour pressure inside vacuum. For example, in MBE deposition of Mg, the K-

cells are usually heated to 350-400°C to incorporate Mg as dopants in III-nitrides. I think the authors have mentioned that they performed the Mg annealing at atmospheric pressure. However, not much discussion is included in the paper. The authors must discuss this issue and provide a clear recipe for the experimental process.

2. Although the authors present 2D-Mg intercalated GaN as a metal/semiconductor superlattice in many places in the manuscript, they have not cited any previous papers on the metal/semiconductor superlattices. The authors should cite some references on the metal/semiconductor superlattices and could reinstate their previous citations.

3. Typically, CMOS techniques require the process temperatures to be below 500°C. Whereas the authors' process temperature is higher. The authors should nuance their statements (para. 15) and add some comments.

4. In many instances, the authors present results and move from the description of n-type GaN to p-type GaN, and vice versa, which makes it quite difficult to follow the essential message that the authors wanted to present. The authors should address this issue for a clearer description of the physical picture.

Bivas Saha
Associate Professor
International Center for Materials Science (ICMS)
Chemistry and Physics of Materials Unit (CPMU)
Jawaharlal Nehru Centre for Advanced Scientific Research (JNCASR)
Office: ICMS-109, Jakkur, Bangalore 560064, Karnataka, India
Email: bsaha@jncasr.ac.in, bivas.mat@gmail.com
Phone: +91-80-2208-2619 (O), +91-6360126595 (M)

Heterogeneous Integration Research Group (HIRG)
<http://old.jncasr.ac.in/bsaha/>

Author Rebuttals to First Revision:

Referees' comments:

Referee #2 (Remarks to the Author):

The authors have done a terrific job in addressing many of my previous review comments, including revealing the detailed atomic structure of Mg-intercalated superlattice by the iDPC-STEM, and quantifying the interatomic strains by using atomically resolved HADDF-STEM imaging, validating the ultralarge uniaxial compressive strain over 10%. The manuscript has been substantially improved with additional experiments, analyses, and discussions, thus would be appealing to a broad community concerning about wide-bandgap semiconductors, 2D materials and strain engineering. But I have a few additional questions and comments for the authors to consider, before publishing this research:

Response:We're sincerely grateful for the positive feedback on our revised manuscript and are pleased that our substantial efforts in addressing the comments during the last round of revision have been positively acknowledged. We're keen to dive into the additional questions and comments the referee has raised and will address them as best as we can.

The referee greatly appreciates the authors on the atomic structure analysis and strain analysis, but it would be better to define and illustrate the "in-plane" and "out-of-plane" strain, respectively, and also check the Poisson's ratio of GaN accordingly. In addition, in supplementary material Fig S1, the authors show that there is obvious distortion in GaN lattice between adjacent intercalants. Can the authors also apply strain field analysis here, to explain the physical mechanism of this lattice distortion and the possible effect on the stability of intercalated structures. It has been previously reported in literature that in 2D materials interfaces (e.g. twisted bilayer graphene), structural distortion near the intercalated structure can lead to long-range interactions. Can the author briefly discuss whether there are long-range interactions between intercalated structures in the Mg-GaN structure?

Response:We're thankful to the referee for the helpful suggestion to clarify the definitions of "in-plane" and "out-of-plane" strain. We've added the necessary explanations and illustrations to improve clarity in these areas. For the in-plane and out-of-plane strains, they are defined according to the following formulas:

$$\begin{aligned} \text{In-plane strain: } \varepsilon_a(\%) &= \frac{a_{\text{Strained}} - a_{\text{Reference}}}{a_{\text{Reference}}} \times 100\% \\ \text{Out-of-plane strain: } \varepsilon_c(\%) &= \frac{c_{\text{Strained}} - c_{\text{Reference}}}{c_{\text{Reference}}} \times 100\% \end{aligned}$$

where the subscript "Strained" refers to GaN in the strain mapping region (GaN between periodic 2D-Mg_i sheets excluding the tip region), and "Reference" denotes GaN in the reference region (outside the intercalation region, see Fig. 3b). *a* and *c* represent the in-plane and out-of-plane lattice constants of hexagonal GaN, respectively. These are obtained based on the measured average distances from atom column to atom column (atom column-by-atom column distance). The average in-plane distance *d*_{in-plane} and out-of-plane distance *d*_{out-of-plane} have the following relationship with the lattice constants in a hexagonal lattice:

$$\begin{aligned} d_{\text{out-of-plane}} &= \frac{1}{2}c \\ d_{\text{in-plane}} &= \frac{\sqrt{3}}{2}a \end{aligned}$$

The direction of in-plane strain is parallel to m -axis ($\langle 1-100 \rangle$ direction), also equivalently to a -axis ($\langle 11-20 \rangle$ direction) in this work because of the symmetry of strain produced by the Mg sheets parallel to both m -axis and a -axis. The direction of out-of-plane strain is parallel to c -axis ($\langle 0001 \rangle$ direction). The directions of m -, a -, and c -axes are indicated at the corner of each figure.

The method description for measuring the distance between atom columns has been added to the Method section: *For atom column-by-atom column distance measurements, images were acquired using a frame averaging approach to limit the effects of sample drift. Each frame average dataset consists of a minimum of 12 individual images which were then position corrected using cross correlation and averaged for the final image. To further limit the effects of drift, frame averaged image series were acquired with the fast scan direction parallel to the measurement direction. The atom column-by-atom column measurements were performed by fitting each atom column position to a two-dimensional Gaussian distribution and measuring the distances between columns directly using the Atomap software package.*

When GaN is uniaxially strained along the c -axis, the Poisson ratio is defined as $\nu = -\frac{\epsilon_{\perp}}{\epsilon_{\parallel}} = -\frac{\Delta d_{\perp}}{\Delta d_{\parallel}}$.

In addition, regarding the image in the original Supplementary Fig. S1 (rearranged as Fig. R1a in this response text) the referee asked about, we have depicted the schematic representation of the atomic arrangement in Fig. R1b. Since N atoms are not visible in ADF-STEM (due to Z-contrast), the lattice distortion is attributed to Ga atoms. Interestingly, when using iDPC-STEM which visualize N atoms (Fig. 2a,b in Main text), we observed that the positions of N atoms largely remain unchanged, while Ga atoms exhibit a gradual shift from lower (N-polar) to upper (Ga-polar) sites from left to right. This shift is depicted with varying degrees of color transparency to indicate the gradual change in the occupancy of Ga atoms at these sites (Fig. R1b).

Therefore, it seems appropriate to measure strain using the drift-minimized HAADF-STEM images, based on the position of Ga atoms only within the MiGs region (highlighted in purple area in Fig. R2). The strain measured from the tip region (highlighted in green area in Fig. R2) appears to be excessively high, primarily due to a polarity shift in Ga atoms that results in a highly “distorted” appearance at the tip of the 2D-Mg_i sheet. Consequently, as shown in Fig. R3, when we tried to use geographic phase analysis (GPA) to determine the strain, the strain value can exceed -30% for out-of-plane and over +10% for in-plane strain at these areas. Instead, if we consider N atoms that remain unshifted and appear less “distorted”, as illustrated in Fig. R2, the strain at the tip region of 2D-Mg_i sheets should resemble to that observed at the core of edge dislocation [Ref. R1], where the symbols \perp in Fig. R2 indicate the positions of edge dislocations.

Fig. R1 Atomic arrangement at the tips of 2D-Mg₂ sheets. a. Atomically resolved HAADF-STEM image showing the lattice distortion at the tip of 2D-Mg₂ sheets (due to Z contrast, N and Mg are hardly invisible in this image) Scale bar: 2 nm. b. Schematic representation of the atomic arrangement of the same area as shown in a. The color coding follows the same scheme as Fig. 2f in the Main text.

Fig. R2 Schematic representation of the arrangement of N and Mg atoms around a pair of 2D-Mg₂ sheets (Ga atoms are omitted for simplicity due to their polarity shift).

Fig. R3 Geographic phase analysis of strain in the GaN region with early-stage Mg intercalation. a. HAADF-STEM image showing GaN with early-stage Mg intercalation. **b.** Out-of-plane strain mapping of the region shown in a. **c.** In-plane strain mapping of the region.

For the long-range interactions, it would be useful to consider them as a displacement field in a continuum body. As illustrated in Fig. 2, the two Mg sheets, acting as two vertically aligned half-edge dislocations of the same sign, exert a repelling force on each other. This repulsion is due to the long-range interactions of these two inserted half atomic planes; when they come too close to each other, the strain field increases.

On the other hand, due to the polarity transition of Ga, as the two Mg planes draw closer, fewer Ga atoms will undergo a polarity shift because the volume of the inversion domain shrinks. This reduction in energy will lead to another set of long-range interactions, attracting the two sheets and aligning them along the c-axis.

The observation of spontaneous alignment of these Mg sheets with typically 5-10 GaN layers between suggests that the energy reduction from shrinking the inversion domain outweighs the increase in the strain field. The balance between the attractive and repelling forces gives the long-range interactions within a pair of Mg sheets their characteristic features.

Corresponding changes in the manuscript:

Added more details to the Strain determination in the Methods section (page 26, line 6 to page 27, line 11) and added relevant paragraphs to Supplementary Note 2 (page 51, page 52, page 53, line 1-7).

[Ref.R1] Zhao, C. W., et al. "Experimental examination of displacement and strain fields in an edge dislocation core." *Acta materialia* 56.11 (2008): 2570-2575.

2. About the CL characterization, STEM-CL was used for Mg-deposited n-type GaN while conventional SEM-CL was used for p-type GaN. Would there be any differences in terms of comparing the CL results for n-type and p-type GaN? Will the difference in methods cause variations in spectral features? For instance, in n-type GaN, the intensity of yellow luminescence is weaker in the MiG region comparing with in the blank GaN substrate, but contrarily it intensifies in the MiG region of p-type GaN. Any explanations? Also about "the CL intensity is markedly weaker in regions containing MiGs structure, associated with an increased carrier density and a strong

electric field”, the decrease of CL intensity may result from many other reasons, such as Mg doping or uneven distribution of MiGs structure or impact of polarization, can we clarify the main mechanism here?

Response:

The referee’s request for further clarifications on the distinctions between STEM-CL and SEM-CL is appreciated. We will answer this question from two main aspects: yellow luminescence and dopant concentration.

1. Yellow luminescence

In the field of III nitride semiconductors, it is well-established that the origin of yellow luminescence in GaN is attributed to Ga vacancies or carbon impurities [Refs. R2, R3]. In STEM-based CL, the FIB-prepared sample undergoes high-energy ion illumination, leading to the creation of a large number of Ga vacancies, which in turn significantly increases the intensity of yellow luminescence. As a result, across all effective sampling points (#1-#4), the intensity of the peak yellow luminescence is higher than that of the band-edge emission (see Extended Data Fig. 3d).

Conversely, in polarization-resolved SEM-based CL, the samples are not subjected to such ion illumination. Consequently, the intensity of the band edge emission peak is far greater than that of the yellow luminescence (see Extended Data Fig. 4f).

Furthermore, the ratio of yellow luminescence intensity to band edge emission intensity differs significantly in samples before and after MiGs incorporation. The ratio of yellow luminescence is higher in samples with MiGs incorporation, both in STEM-CL (Extended Data Fig. 3d, location #4 compared to locations #1,#2) and in SEM-CL (Extended Data 4f). We believe that the enhanced yellow luminescence can be attributed to a combination of factors. Firstly, the MiGs might contain higher levels of Ga vacancies (this could potentially be confirmed by positron annihilation spectroscopy with enhanced resolution [Ref. R4]), which are due to the possible loss of Ga during the reaction with metallic Mg (charge transfer between Ga^{3+} to Mg^0). Secondly, aside from Ga vacancies, the presence of carbon impurities is also a contributing factor. As shown in the SIMS profiling (Supplementary Fig. S2), the concentration of O and C concentration in the Mg intercalated region is high, likely because the metallic Mg source is not pure, being only 99%-99.9% pure (up to three Ns). The high concentration of C may explain the higher ratio of yellow luminescence to band edge emission in the MiGs compared to in the GaN. Thirdly, the enhanced yellow luminescence might also be due to factors such as Mg doping, uneven distribution of MiGs structure, or the impact of polarization. These factors could act as deep-level traps that lead to yellow luminescence or non-radiative recombination centers.

2. Dopant concentration

Generally, setting aside other factors influence the CL intensity, the intensity of the CL signal in a homogeneous and defect-free sample is roughly inversely proportional to the dopant concentration, due to the reduced carrier diffusion length in a heavily doped sample. In our case, the n-type GaN sample has lower dopant concentration than p-type GaN sample. Therefore, the CL intensity should be weaker in the p-type GaN sample. Since the MiGs region exhibits weaker CL intensity, we anticipate that in SEM-CL, where the sampling volume is large, the collected signals could have a higher contribution ratio from MiGs-region in a p-type GaN.

The weak CL intensity from the MiGs, apart from being due to a possible higher carrier concentration, may also result from factors such as Mg doping, uneven distribution of MiGs structure, or the impact of polarization, as the referee suggested. Essentially, any mechanism that facilitates non-radiative recombination will decrease CL intensity. Since individual interstitial Mg (0D-Mg) introduces multiple deep-level traps, interstitial Mg intercalant sheets (2D-Mg) may create continuous deep-level traps, enhancing the non-radiative recombination of electrons and holes. A

non-uniform distribution of MiGs structure might decrease CL intensity due to scattering at the domain boundary. Additionally, the impact of polarization could decrease CL intensity due to the strong polarization field and the possible gradient-induced hole doping by the polarization field. The latter factor actually increases carrier density and the strong electric field, as introduced previously. At this point, it is challenging to determine which factor(s) predominantly contribute to the decrease in CL intensities, as all could potentially lead to such a decrease. We believe that this topic warrants further in-depth study by designing multiple control samples.

Corresponding changes in the manuscript:

Added more relevant paragraphs to Supplementary Note 1 (page 47-48).

[Ref. R2] Neugebauer, Jörg, and Chris G. Van de Walle. "Gallium vacancies and the yellow luminescence in GaN." *Applied Physics Letters* 69.4 (1996): 503-505.

[Ref. R3] Lyons, J. L., A. Janotti, and C. G. Van de Walle. "Carbon impurities and the yellow luminescence in GaN." *Applied Physics Letters* 97.15 (2010).

[Ref. R4] R. Armitage, William Hong, Qing Yang, H. Feick, J. Gebauer, E. R. Weber, S. Hautakangas, and K. Saarinen "Contributions from gallium vacancies and carbon-related defects to the "yellow luminescence" in GaN", *Applied Physics Letters* 82, 3457,(2003).

3. The method used in this study is annealing at 550°C for 10 mins. Now the authors have discussed the effect of different temperatures on the structure of MiGs, but not yet discussed the effect of time. Like what will happen to the MiGs structure if the annealing time is longer than 10 minutes? What would be the thickness of MiGs structure after annealing for more than 10 mins?

Response:

In our discussion regarding the subtle differences in annealing temperatures that initiate the early-stage formation of MiGs, which enables ohmic contact to existing p-type GaN (Fig. 4f). We suggest that the apparent change in *I-V* behavior could be an effective indicator of Mg intercalation formation. We have concluded a more systematic study, varying the annealing temperature and annealing time. As shown in Fig. R4, at an annealing temperature of 550 °C for 10 minutes, we found that the ohmic contact is enabled with good linearity in *I-V* curve. However, annealing at 500 °C for 10 min resulted in a non-ohmic contact. Furthermore, our findings indicate that annealing at 515 °C, 525 °C, and 535 °C for 10 minutes leads to progressively improved ohmic contact. Interestingly, while annealing for 10 min at 500 °C does not result in ohmic contact, extending the annealing time ohmic contact, annealing at 500 °C to 60 minutes improves the ohmic contact, placing it in the performance range observed between 525 °C and 535 °C [Ref. R5].

Fig. R4 I-V characteristics of the TLM test structure (with a spacing of 15 μm) on the p-type GaN annealed with Mg at different temperature and time.

Further, annealing at higher temperature, the sheet resistance could be further reduced, suggesting the further formation of the MiGs structure. This made us assume that this is a diffusion-driven process, where both temperature and time contribute to the formation of MiGs, as roughly defined by $x^2=4Dt$ (x is diffusion length, D is diffusion coefficient, and t is diffusion time). By using the diffusion length determined by SIMS profiling at different temperature, we roughly derive the diffusion coefficient and the activation energy to be 0.58 eV, such small energy falls into the category of interstitial diffusion of Mg.

Fig. R5 Temperature and diffusion coefficient in the diffusion of Mg in the formation of MiGs in GaN.

Therefore, we assume that annealing at even lower temperatures, like 400 °C (which aligns with the upper temperature limit for back-end-line processes), for a sufficiently long time, possibly extending to many hours, could result in the formation of similar MiGs structure as that achieved by annealing at higher temperatures for shorter time.

Conversely, when annealing is conducted at a high temperature of 800 °C, we found that a short time of 1 min is enough to enable ohmic contact to p-type GaN. We also experimented with extending the annealing time to 30 minutes at 800 °C, but according to STEM analysis, the microstructure exhibited no significant differences. This outcome may be attributed to the by-reaction of the upper Mg film with nitrogen: $3\text{Mg} + \text{N}_2 \xrightarrow{800\text{ °C}} \text{Mg}_3\text{N}_2$ which possibly limits the Mg reaction with GaN. Additionally, even if the MiGs layer becomes thicker, as depicted in Extended Data Fig. 1, the thickened MiGs will

have a continuous film portion at the upper part, which is prone to exfoliation during the cooling and subsequent acid cleaning process to remove the remaining Mg compound. The remaining structure still predominantly consists of a discontinuous MiGs phase with pyramidal domains (Extended Data Fig. 1). This phenomenon makes it difficult to determine accurately the average thickness of the MiGs region. Further efforts are necessary to devise methods to protect such structures from delamination.

Corresponding changes in the manuscript:

Added relevant sentences in the Main section (Page 10, line 13), Method section (page 23, line 6-14) and Supplementary Note 2 (page 50, line 20-page 51, line 2).

[Ref.R5] Wang, Jia, et al. "Ohmic contact to p-type GaN enabled by post-growth diffusion of magnesium." *IEEE Electron Device Letters* 43.1 (2021): 150-153.

4. The author proposed that there might be charge transfer in Mg intercalated GaN, which seems that there is a chemical reaction here. What is the difference between this and the classical alkali metal intercalation reaction? Can you predict the possible highest density of Mg in GaN, and how about optimum density?

Response:

We hypothesize that a chemical reaction occurs in this context, as a similar effect has not been observed when Mg-containing compound are annealed on GaN. Although attempts have been made to anneal compounds like MgF_2 and MgO on GaN to improve ohmic contact, there is no report in such success. We think that the difference most likely lies in the chemical and electron state of metallic Mg compared to these compounds. Metallic Mg (Mg^0) tends to lose electron in its metallic form, whereas it has already lost electrons as Mg^{2+} in MgF_2 and MgO . The electrons are likely transferred Ga^{3+} , leading to a substitution chemical reaction. In our follow-up study, one of our objectives is to find evidence of Ga metal and Ga_2O_3 due to the oxidation of Ga^0 in air.

It is insightful of the referee to direct our attention to alkali metal (AM) intercalation. Upon reviewing existing literature on AM intercalation, we found that it has attracted significant interest [Ref. R6]. In terms of charge transfer, the low electronegativity of alkali metals facilitates the easy donation of electrons, positioning them as excellent reducing agents. This characteristic is beneficial in redox reactions within batteries, where electron transfer is essential. Consequently, we assume that annealing of AM on GaN could produce chemical reaction via a charge transfer between AM and Ga, potentially leading to the spontaneous roughening of the GaN surface as part of corrosion reaction.

Such a reaction could help to drive AM atoms into GaN lattice. However, it might not be easy to intercalate AM into hexagonal GaN which features rigid mixed ionic and covalent bonds, because forming 2D-AM sheets inside GaN could result in high interfacial energy due to the lattice mismatch. In contrast, Mg and GaN share the same lattice type and lattice constant, greatly helping to minimize the interfacial energy involved. In comparison, such interfacial energy is less of an issue in the case of classical alkali metal intercalation reaction, when 2D-AM sheets are inserted into van der Waals (vdWs) materials with relative ease.

We apologize for not being able to offer more detailed professional insights in this area, as we and our colleagues and current project collaborators primarily consist of semiconductor physicists and electrical engineers. Our expertise in chemical synthesis is somewhat limited. However, we hope that by sharing our findings through interdisciplinary platforms, this study may spark wider interest, particularly among those working on alkali metal intercalation and related fields in energy storage.

In response to the referee's inquiry about the highest possible density of Mg in GaN, we have experimentally used back-side SIMS to determine the Mg concentration within GaN. Our measurement result indicates a maximum concentration of $4.0 \times 10^{21} \text{ cm}^{-3}$ at the surface (approximately 10% of GaN) in cases of high temperature annealing.

To consider the density of Mg in GaN, we should consider the dual effects of the density of Mg in a single MiGs domain and the overall density of MiGs domains within GaN (i.e., the surface coverage of MiGs on GaN). The density of Mg in a single MiGs domain can be straightforwardly calculated based on the C-site occupancy of Mg sheets in a ABAB hexagonal GaN lattice. Our observations show that 6 and 7 are the most commonly observed numbers of GaN layers intercalated between Mg sheets in the fully developed MiGs structure. We consider these as the optimum number of GaN layers for intercalation, leading to Mg densities in GaN of 14.29% for a 6-layer configuration ($1/(6+1) \times 100\%$) and 12.50% for a 7-layer configuration ($1/(7+1) \times 100\%$). If the MiGs can perfectly cover the entire surface of GaN (this applies to the case of a continuous MiGs thin film, which could possibly be achievable if the exfoliation issue is addressed), the coverage of MiGs within GaN is 1 in the continuous MiGs thin film. As a result, the highest density of Mg in GaN should be between 12.50% and 14.29% in the ideal case. Intrinsically, the coverage will decrease towards deep inside due to the inversion pyramidal shape of MiGs domains in (0001) GaN. Due to that the maximum coverage of MiGs occurs at the surface, the density of Mg in GaN will also decrease from the surface to inside.

Furthermore, regarding the coverage of MiGs within GaN in our experiment, as elaborated in Supplementary Note 4, our experimental findings suggest that the maximum coverage of MiGs on GaN at the surface, is determined to be 57.12% for configurations with 6 GaN layers and 66.61% for those with 7 GaN layers, based on the SIMS measurement data.

Corresponding changes in the manuscript:

Added relevant paragraphs in Supplementary Note 2 (page 49, line 10-23).

[Ref. R6] Lin, Yung-Chang, et al. "Alkali metal bilayer intercalation in graphene." Nature communications 15.1 (2024): 425.

Two minor issues to be addressed:

1. On page 14, line 1. The CL spectra of p-type GaN was in Extended Data Fig. 3 instead of Fig. 2.

Response:

We are thankful for the referee's attentive efforts in reviewing this manuscript. We apologize for the typo and have corrected the expression accordingly.

Corresponding changes in the manuscript:

Page 43, Line 15.

2. The Zoomed-in STEM in Fig 4c can be more clear and sharp to review the MiGs structure, and there is not figure caption for Fig 4d (or there should be no "d" from your current caption)

Response:

We appreciate the suggestion regarding this matter. In response, we have added an inset (HAADF-STEM showing sharper contrast) into the zoomed-in BF-STEM image in what was previously Fig. 4c (now updated to Extended Data Fig. 3c) and have also included a figure caption for what was previously Fig. 4d (now updated to Extended Data Fig. 3d).

Corresponding changes in the manuscript:

Page 33: Extended Data Fig. 3c,3d and relevant figure caption.

Referee #3 (Remarks to the Author):

In the revised manuscript entitled “Observation of 2D-magnesium-intercalated gallium nitride superlattices” Wang et al. have performed substantial revision of their previous work. They performed several new experiments, reinterpreted many of the previous results, and addressed all of the referees’ comments. The volume of work that has gone into revising the manuscript justifies the long delay in resubmitting the manuscript.

As I had highlighted in my comments, the previous version of the manuscript had presented three main claims/novelties (a) achieving intercalation of metals (Mg) inside an ionic/covalent semiconductor, (b) increase in the hole mobility of GaN due to the compressive strain and reversal of the sign of crystal field splitting, and (c) demonstration of Mg/GaN as a system for transport studies on metal/semiconductor superlattices.

Following this substantial revision of the manuscript, two of these claims have gone through a major revision and a new important claim is included.

1. Following the referee’s comment, the authors have moderated their previous claim of “first-of-its-kind observation of intercalation of 2D metal into bulk semiconductor” to a more nuanced “interstitial intercalation”, which is a more appropriate description in my opinion. Although the author’s previous strong claim is moderated a bit in this revised manuscript, the observation of the intercalation of interstitial Mg monolayers inside GaN is still highly interesting and stands as a major novelty of this study. The new microscopy images are also very interesting and represent the work aptly.

2. The authors’ 2nd claim about the increase in the hole mobility of GaN due to compressive strain has gone through a substantial change. Firstly, as the authors have mentioned, their previous mobility measurements with the Hall method did not capture the out-of-plane mobility, which the theory has predicted to increase. At the same time, the mobility measurements had some consistency issues, which the authors mentioned in the response letter.

To measure the out-of-plane mobility, the authors have now used terahertz time-domain ellipsometry, which provides indirect evidence of the out-of-plane mobility enhancement due to the intercalation of Mg in GaN. However, though terahertz time-domain ellipsometry gives very good qualitative information, a clear and unambiguous demonstration of out-of-plane mobility increase is not demonstrated (as the authors also mention in multiple places). The present demonstrations serve as a very strong indication but not as a certain proof.

The authors have also revised their earlier demonstration of strain-induced band reversal with a new STEM-CL analysis, which is a bit puzzling. In my opinion, the previous polarised CL spectroscopy was clear and convincing. However, the new STEM-CL measurement is very confusing and difficult to follow. I am unsure if these new STEM-CL measurements can be seen as an unambiguous demonstration of the strain-induced band reversal. I am also not sure why the authors would remove the previous CL measurements.

Therefore, the 2nd claim is no longer a clear and definite physical picture.

3. The third claim remains “as is” since it is more of a proposal for the future, which is fine.

The revised manuscript has also included new important claims, which should now be considered due to their impacts-

4. The authors now present a clear observation of the polarity transition in GaN due to the 2D-Mg intercalation in the interstitial sites. This observation provides direct evidence for the pyramidal inversion domains generally observed in Mg-doped GaN. This is an important novelty, in my opinion.

5. Lastly, although the authors present claims on improved Ohmic contact to p-type GaN and Schottky barrier height modulation for n-type GaN, I think it is better not to consider these as completely novel since the authors already published works on similar lines. The interpretations are now, of course, clearly evidenced in this paper.

Therefore, keeping in mind the novelty, technical quality, and broad appeal of the manuscript, I think that it would be most appropriate to focus on the interstitial intercalation, polarity transition, and potential for metal/semiconductor superlattice as the main message. Given the important novelties of these highlighted points, I support the publication of such as revised manuscript in Nature.

Focusing on these points would make the manuscript short and concise, which will be much more impactful, in my opinion. It will also significantly advance not only the research and industrial application of GaN but also other semiconductors and heterostructures in general. The rest of the data, analysis and interpretations can be included in the Supplemental Materials for reference and as a motivation for future research.

Response:

We are grateful for the positive evaluation. Specifically, we deeply value the thorough review of our manuscript's claims and structure provided in the revised submission. This guidance has been crucial in refining our manuscript for this revision, during which we have reorganized the content between the Main text and the Supplementary Information. We have aimed to emphasize the essential content in the Main text, focusing on the interstitial intercalation (and the high strain associated with such intercalation) and polarity transition, which may interest a broad audience of materials scientists, along with their immediate effects on the modified electrical properties, which are of particular importance to researchers in the field of III-nitride semiconductors.

We concur with the analysis of the claims identified by the referee following a meticulous review of the manuscript. We have ranked the claims based on a comprehensive evaluation of their novelty and significance:

1. Discovery of spontaneously formed metal/semiconductor superlattices, with implications for previously demonstrated artificial metal/semiconductor superlattices.
2. Unique periodic polarity transition behavior within the superlattice structure.
3. Interstitial intercalation
 - 3-1. First-of-its-kind interstitial-type intercalation behavior.
 - 3-2. Demonstration of ultrahigh strain in the parent lattice as a key feature of such

interstitial intercalation.

4. Modified physical properties due to the interstitial intercalation of 2D-Mg sheets, which can be further subdivided into:
 - 4-1. Modified electronic energy band structure of GaN under ultra-high strain.
 - 4-2. Modified optical properties, including electron-to-photon interaction (cathodoluminescence) and photon-to-electron interaction (tera-hertz time-domain ellipsometry).
 - 4-3. Modified electrical properties, with a focus on the values for immediate electronic device applications, including C-V and I-V characterizations.

Due to space constraints in the Main text, we have organized the discussions and figures regarding claims #1, #2, and #3 into three sub-sections, each with corresponding display items (figures and tables). For the fourth claim, only the discussion and relevant figure pertaining to claim #4-3 are included in the Main text and the corresponding fourth display item. Discussions regarding claims #4-1 and #4-2 are included in the Supplementary Information as Supplementary Notes, and the relevant figures are presented as Extended Data Figures and Supplementary Figures. Additionally, other supporting data (texts, tables, and figures) for claims #1-#4 are also included in the Supplementary Information as Supplementary Notes, Figures, and Tables accordingly.

We believe this revision incorporates the constructive suggestions of the referee, aiming to make the manuscript succinct and impactful, which, as the referee put it, "will not only significantly advance the research and industrial application of GaN but also enhance the broader field of semiconductors and heterostructures." We hope the referee finds this satisfactory.

Regarding the referee's evaluation of the original second claim about "the increase in hole mobility of GaN due to compressive strain has undergone substantial change," leading to the conclusion that the second claim "no longer presents a clear and definitive physical picture," we acknowledge this assessment as accurate. Consequently, we have degraded this claim to a sub-claim (claim #4-2) in the revised manuscript.

The introduction of new evaluation data (STEM-CL, new polarization-resolved SEM-CL, and tera-hertz time-domain ellipsometry) was essential to ensure that these claims are supported by reliable and robust data, which partly explains the extended period required for the last revision. We appreciate the referee's opinion that the current demonstrations of enhanced out-of-plane mobility, as evidenced by new terahertz time-domain ellipsometry, "serve as a strong indication but not as certain proof."

Specifically, regarding the arrangement of STEM-CL and polarized-SEM-CL analyses, which the referee found puzzling, we offer clarification here: CL is employed to demonstrate the effects of strain on the modified CL emission, which requires two effects: first, a relative blue-shift of band-edge emission to suggest an increased bandgap; second, a dominant TM emission to indicate the predominance of the split-off hole band (valence band reversal). The earlier polarized CL spectroscopy was clear and convincing in demonstrating the second effect (valence band reversal) due to its apparent indication of reversed band-edge TM emission. However, as some referees noted, the old CL data could not exhibit the first effect, leading us to question its robustness. Consequently, we sought the services of some of the world's most renowned CL providers for new CL data: Toray (Japan) is capable of performing STEM-CL, a state-of-the-art CL technique that ensures emissions are directly coming from the nano-region visualized simultaneously by STEM. In the new CL data, the relative blue-shift in the band edge emission was significant and basically consistent with our expectations based on DFT calculations, thus proving the first effect. However, their STEM-CL does not have a filtering and analyzer module for polarization-resolved emission. To demonstrate the second effect, *Attolight* (Switzerland), a leading company in CL technology, provided polarization-resolved CL spectroscopy characterization (though they lack STEM-CL capability). Their CL data

showed a blue-shift of band-edge emission (not as significant as that from STEM-CL though) and a reversal of TM emission only in the yellow luminescence spectral region. Since we observed that yellow luminescence is significantly enhanced in the MiGs region, and the CL intensity from the MiGs is weaker with a larger interaction volume in SEM-CL encompassing emissions from both MiGs and GaN regions, we propose that changes in intensity reflected from yellow luminescence are a more reliable indicator of the modified optical properties of the MiGs region. In conclusion, STEM-CL performed by *Toray* proves the first effect, while the second effect remains unproven due to technological limitations. In contrast, the polarization-resolved SEM-CL performed by *Attolight* ambiguously proved both effects. The previous CL data, conducted by ourselves, validated the second effects but somehow contradicted the first effect, leading to our decision for its removal. Despite these issues, we agree with the referee's judgment that the current CL data is still less straightforward and definitive, justifying the degradation of the previous claim to a sub-claim of #4-2 when combining data from three state-of-the-art characterization tools together: STEM-CL, polarization-resolved SEM-CL, and terahertz time-domain spectroscopy. For future experiment, we think that the possible invention of a STEM-CL with polarization-resolved emission capability would be the ideal solution to simultaneously confirm both effects under investigation, thus providing a clear and definitive evidence.

Main corresponding changes in the manuscript:

Refined the Main text from over 5000 words to 2500 words.

In addition to this main point, I would also strongly urge the authors to address the following points in the revised manuscript, which will strengthen the paper-

1. Mg is extremely prone to oxidation when exposed to air. Therefore, the authors must outline how they protected the Mg from oxidation between depositing on GaN and annealing at high temperatures under atmospheric pressure. Similarly, generally, when Mg temperature exceeds more than 350-400°C, it starts to vaporize heavily due to its high vapour pressure inside vacuum. For example, in MBE deposition of Mg, the K-cells are usually heated to 350-400°C to incorporate Mg as dopants in III-nitrides. I think the authors have mentioned that they performed the Mg annealing at atmospheric pressure. However, not much discussion is included in the paper. The authors must discuss this issue and provide a clear recipe for the experimental process.

Response:

We understand the referee's concerns regarding the handling of the Mg thin film. Generally, the oxidation resistance of Mg is considered good in dry air up to approximately 400 °C and up to about 350°C in moist air, with humidity playing a significant role in Mg corrosion as it forms Mg(OH)₂ [Ref. R7]. From our experience, at room temperature, the oxidation resistance of Mg is robust, the as-deposited Mg thin film (50 nm) retains its metallic luster when kept in air for weeks. We did not take special measures to prevent its oxidation, and the thermal annealing of the Mg film was typically conducted within a few days after deposition. Furthermore, we utilized pure nitrogen or argon ambient during annealing at elevated temperature to minimize oxygen or moisture effects. However, we also acknowledge that the Mg film surface is likely oxidized and covered by a thin oxide layer due to its exposure to air. This could also explain the additional oxygen incorporation into the MiGs structure in the SIMS profiles (Supplementary Fig. 3)

Another point worth mentioning concerns the lift-off process of photoresist and Mg where a liquid solution is necessary. We avoided using DI water, discovering that Mg nanofilm tends to dissolve in water (losing its metallic color) after just a few minutes. Instead, we employed pure isopropyl alcohol (IPA) instead of DI water, finding that Mg remains stable in IPA for an extended period (at least an hour), which is significantly sufficient for the lift-off process. In fact, IPA is commonly used in lieu of DI water in semiconductor processing practices. Additionally, we used N-Methyl-2-pyrrolidone (NMP) (or acetone), followed by methanol for the lift-off process, before employing IPA.

Fig. R6. Thermodynamic equilibrium between temperature degree (in units of degree centigrade for upper X-axis label and of degree Kelvin for lower X-axis label) and saturation vapor pressure (in units of mm Hg for left Y-axis label and of ATM for right Y-axis label) of pure metal Mg (highlighted in red color) Figure source: 1957 Radio Corporation of American.

Fig. R7. Phase diagram of Mg [Ref. R8].

Regarding the evaporation of Mg under vacuum condition, it is true, as seen in the Figs. R1 and R2. Under the high vacuum condition in a typical MBE growth environment (10^{-8} to 10^{-12} torr, roughly 10^{-6} to 10^{-8} Pa), the boiling point of Mg is below 200 °C, whereas under the 1 atmosphere, the boiling point of Mg increases to over 1000 °C, much higher than the melting point of 650 °C.

Based on our experience, we attempted to anneal a 50 nm thick Mg film in an RTA (furnace at 700 °C under a vacuum of 10 Pa, corresponding to a boiling point of approximately 500 °C for Mg. We observed that the Mg disappeared, likely having sublimated during the temperature ramp-up phase, and consequently, there was no improvement in the ohmic contact to p-type GaN as determined by our *I-V* testing. This led us to conclude that using atmosphere pressure is recommended to prevent the early sublimation of Mg before reacting with GaN.

Finally, in addition to directly answering this question, we would like to communicate additional insights to the referees and possibly the editors: pure metal Mg, as admitted in numerous studies, is not fully studied until now. Most research today related to Mg, is on the Mg alloys due to their superior mechanical properties, where pure Mg itself receive little study. Due to its infrequent use in semiconductor applications, metallic Mg is also underexplored and less familiar to researchers and practitioners. This might underline the reason for the lack of exploration of the reaction of metallic Mg on GaN for a long time. In fact, when we introduced our approach of electron beam evaporation of Mg to some veteran device processing engineers and colleagues worldwide, many of them were previously unaware that metal Mg is a readily commercially available source for physical vapor deposition, as they often assume Mg to be highly reactive and even prone to explosion in the chamber, which is not the case. Therefore, we hope that this study can appeal to more interdisciplinary efforts across both semiconductor and metallurgy fields to stimulate interest in exploring the fundamental properties of metal Mg.

Corresponding changes in the manuscript:

Added more details on the handling of Mg film in the sub-section of “Thermal annealing of metallic Mg on GaN” in Methods section (page 21, line 16-23 and page 22, line7-page 23, line 2).

[Ref. R7] Hu, Henry, Xueyuan Nie, and Yueyu Ma. "Corrosion and surface treatment of magnesium alloys." *Magnesium Alloys-Properties in Solid and Liquid States* (2014): 67-108. Available online at: <https://www.intechopen.com/chapters/47427>.

[Ref. R8] Gieseke, Matthias, et al. "Selective laser melting of magnesium and magnesium alloys." *Magnesium Technology 2013* (2016): 65-68.

2. Although the authors present 2D-Mg intercalated GaN as a metal/semiconductor superlattice in many places in the manuscript, they have not cited any previous papers on the metal/semiconductor superlattices. The authors should cite some references on the metal/semiconductor superlattices and could reinstate their previous citations.

Response:

We greatly thank the kinder reminder on this note. We have added more references in our third claim and relevant discussion of metal/semiconductor superlattice in the Top Summary paragraph (Abstract) and the Main section.

Corresponding changes in the manuscript:

Added sentences and relevant sentences in the Top Summary paragraph (page 2, line 19) and the Main section (page 11, line 6-10).

3. Typically, CMOS techniques require the process temperatures to be below 500°C. Whereas the authors' process temperature is higher. The authors should nuance their statements (para. 15) and add some comments.

Response:

We understand the concern of the referee for more clarifications on the compatibility to CMOS techniques. We think that the process highlighted in this work is fully CMOS compatible in the front-end-of-line procedure. As CMOS techniques are further divided into front-end-of-line and back-end-of-line, while the former requires temperature over 400 °C and latter requires less than 400 °C. Since most of the CMOS processes are carried out at relatively lower temperatures (below 500 °C); the MiGs formation is recommended as the process initiation step, the front-end-of-line steps. Once formed, the stable MiGs incorporated GaN surface will subsequently support both gate-first and gate-last CMOS processes.

4. In many instances, the authors present results and move from the description of n-type GaN to p-type GaN, and vice versa, which makes it quite difficult to follow the essential message that the authors wanted to present. The authors should address this issue for a clearer description of the physical picture.

Response:

We apologize for the confusion to the referee by our frequent transition between the notion of n-type GaN and p-type GaN in multiple sections throughout the manuscript. Additionally, it is special that n-type GaN in this context is, by nature, unintentionally doped GaN. Given the limitations of current growth technologies, GaN cannot be produced with the high level of purity with silicon, leading to it to inherently exhibit properties similar to those of low-doped n-type material.

We have found that in terms of MiGs formation and morphology, there is no significant difference between n-type GaN and p-type GaN. Hence, including studies on both n-type and p-type GaN samples in our research can help demonstrate the general applicability of our thermal annealing

method with metallic Mg on GaN. This approach is effective across a wide variety of GaN samples, irrespective of their conductivity type and film quality. In essence, the primary distinction between n-type GaN and p-type GaN lies in the initial Mg content within these samples. Specifically, n-type GaN and undoped GaN do not contain initially present Mg, whereas p-type GaN already has a significant amount of Mg. These pre-existing Mg atoms could also contribute to the constituent Mg in the formation of Mg sheets. Consequently, the presence of Mg in p-type GaN may slightly reduce the diffusion temperature or time required to form the MiGs structure.

Considering that we are submitting our manuscript to a journal orientated for general science audience with an interdisciplinary readership, we recognized the importance of avoiding highly specific details to ensure conciseness and a smoother reading experience. Based on this understanding, we agree that it is not wise to extensively detail the properties of n-type GaN and p-type GaN samples, except when discussing the *I-V* characterizations and hole transport, where p-type matrix may be necessary and more directly relevant.

With this perspective, we have been more mindful in our discussions and have accordingly reorganized more detailed technical discussions to the Supplementary Notes. This approach is intended to make the main text more accessible to a broader audience while still providing the necessary depth for those interested in the specific nuances of our work.

Corresponding changes in the manuscript:

Refined the schematic figures (Fig. 4a, Extended Data Fig. 4a, and Extended Data Fig. 6) for less specific texts.

Added sentences to the Methods section (page 23, line 22-page 24, line 5).

Reviewer Reports on the Second Revision:

Referees' comments:

Referee #2 (Remarks to the Author):

The authors have mostly addressed my comments with great efforts, particularly the strain distribution and fine crystal structure.

Just two remaining optional questions: Can the authors show or at least discuss the in-plane lattice structure of the intercalated Mg, as most of the existing experimental data originated from cross-sectional atomic imaging? Can the authors actually control the resulted sizes of Mg intercalant sheet as well as GaN layers between Mg sheets in MiGs by changing experiment conditions?

Referee #3 (Remarks to the Author):

I have gone through the response letter and the revised manuscript. The authors have addressed my comments satisfactorily and revised the manuscript accordingly. Therefore, I recommend the acceptance and publication of this manuscript in Nature.

Bivas Saha, Ph.D.

Associate Professor

International Center for Materials Science (ICMS)

Chemistry and Physics of Materials Unit (CPMU)

Jawaharlal Nehru Centre for Advanced Scientific Research (JNCASR)

Office: ICMS-109, Jakkur, Bangalore 560064, Karnataka, India

Email: bsaha@jncasr.ac.in, bivas.mat@gmail.com

Phone: +91-80-2208-2619 (O), +91-6360126595 (M)

Heterogeneous Integration Research Group (HIRG)

<http://old.jncasr.ac.in/bsaha/>

Author Rebuttals to Second Revision:

Referees' comments:

Referee #2 (Remarks to the Author):

The authors have mostly addressed my comments with great efforts, particularly the strain distribution and fine crystal structure.

Just two remaining optional questions: Can the authors show or at least discuss the in-plane lattice structure of the intercalated Mg, as most of the existing experimental data originated from cross-sectional atomic imaging? Can the authors actually control the resulted sizes of Mg intercalant sheet as well as GaN layers between Mg sheets in MiGs by changing experiment conditions?

Response:

We are pleased to learn that the referee is satisfied with our efforts to address the comments. We are also thankful for the additional questions, which hold particular relevance to our follow-up study on this work.

1. Regarding the in-plane lattice structure of the intercalated Mg: Although our presentation is limited to cross-sectional atomic imaging, the orientation of these images includes both *m*- and *a*-directions, which are orthogonal. These images reveal no significant differences, confirming the in-plane symmetry of both hexagonal GaN and hexagonal Mg. Hence, we infer that the in-plane 2D-Mg sheets exhibit in-plane symmetry. Furthermore, given the single crystallinity of the Mg and GaN atomic layers and the close lattice registry between them, we postulate that the in-plane 2D-Mg sheets possess well-defined crystallographic shapes, specifically hexagonal rather than circular. We acknowledge that the most conclusive and straightforward evidence would be a plan-view atomic resolution STEM image. However, acquiring such images may pose technical challenges, given the single-atomic layer thickness of the Mg sheets. Despite this, acquiring these images is listed as an immediate goal in our future characterization plans.
2. Since these 2D-Mg intercalant sheets are vertically aligned into a giant inverted pyramidal domain, the size control of these MiGs domains incorporating constituent Mg intercalant sheets is one of the most important topics for future study. We believe that the formation and development of these domains largely resemble the nucleation and growth of precipitate-like phases from a parental phase, which is heavily influenced by the kinetic factor of thermal diffusion of Mg interstitials (0D-Mg). The density and distribution of initial Mg interstitials in GaN, analogous to the degree of supercooling in precipitate formation from solid solution, serve as a critical thermodynamic factor. Two levels are considered in influencing the size of these MiGs domains:
 - 2.1 Median size of the MiGs domains: Increasing the median size of the MiGs domains requires a continuous supply of Mg interstitials diffusing to the existing domains. Currently, when the Mg thin film on GaN is annealed, the upper layer of the Mg thin film tends to react with oxygen or nitrogen under high temperatures, preventing further Mg supply. A potential solution could involve multiple annealing sessions.
 - 2.2 Size distribution of the MiGs domains: At this level, efforts to suppress the nucleation of these domains increase size variation and cause a non-homogenous distribution of the MiGs

domains. Conversely, promoting nucleation should result in a reduced size variation of the MiGs domains.

Our current basic approach, involving annealing an amorphous Mg thin film onto a GaN thin film through a simple and singular rapid thermal annealing process without optimization of any conditions, is rudimentary. To improve the controllability of the size (both median size and size variation) of the MiGs domains, focused and systematic experiments targeting all stages of the process are essential. These stages range from the reaction of Mg and GaN, driving Mg interstitials into the GaN lattice, to the assembly of Mg interstitial sheets into domains. The crystallinity of the Mg thin film, along with the times of annealing processes and the temperature in each process, are primarily targeted for future investigation.

Corresponding changes in the manuscript:

Added a few lines to the Supplementary Information.

Referee #3 (Remarks to the Author):

I have gone through the response letter and the revised manuscript. The authors have addressed my comments satisfactorily and revised the manuscript accordingly. Therefore, I recommend the acceptance and publication of this manuscript in Nature.

Bivas Saha, Ph.D.

Associate Professor

International Center for Materials Science (ICMS)

Chemistry and Physics of Materials Unit (CPMU)

Jawaharlal Nehru Centre for Advanced Scientific Research (JNCASR)

Office: ICMS-109, Jakkur, Bangalore 560064, Karnataka, India

Email: bsaha@jncasr.ac.in, divas.mat@gmail.com

Phone: +91-80-2208-2619 (O), +91-6360126595 (M)

Heterogeneous Integration Research Group (HIRG)

Response:

We sincerely appreciate the positive feedback on our revised manuscript.